# The postmonsoon carbon biogeochemistry of the Hooghly-Sundarbans estuarine system under different levels of anthropogenic impacts

**Manab Kumar Dutta[1], Sanjeev Kumar[1]\*, Rupa Mukherjee[1], Prasun Sanyal[2], Sandip Kumar Mukhopadhyay[2]**

[1]Geosciences Division, Physical Research Laboratory, Ahmedabad - 380009, Gujarat, India
[2]Department of Marine Science, University of Calcutta, Kolkata - 700019, West Bengal, India

\***Correspondence**: Sanjeev Kumar (sanjeev@prl.res.in)

**Abstract**

The present study focused on understanding differences in postmonsoon carbon (C) biogeochemistry of two adjacent estuaries undergoing different levels of anthropogenic stresses by investigating anthropogenically influenced Hooghly estuary and mangrove-dominated estuaries of the Sundarbans in the north-eastern India. The salinity of well oxygenated estuaries of the Sundarbans (DO: 91 - 104%) varied over a narrow range (12.74 - 16.69) relative to the Hooghly (0.04 - 10.37). A mixing model suggested a combination of processes including freshwater intrusion, carbonate precipitation, and carbonate dissolution to be major factor controlling DIC dynamics in the freshwater regime of the Hooghly, whereas phytoplankton productivity and $CO_2$ outgassing dominated in the mixing regime. In the Sundarbans, removal of DIC (via $CO_2$ outgassing, phytoplankton uptake, and export to adjoining continental shelf region) dominated over its addition through mineralization of mangrove derived organic C. The concentration of DOC in the Hooghly was ~ 40% higher than in the Sundarbans, which was largely due to cumulative effect of anthropogenic inputs, DOC-POC interconversion, and groundwater contribution rather than freshwater mediated input. The measured $\delta^{13}C_{POC}$ in the Hooghly suggested particulate organic matter contributions from different sources (freshwater runoff, terrestrial $C_3$ plants, and anthropogenic discharge), whereas the contribution from $C_3$ plants was dominant at the Sundarbans. The significant departure of $\delta^{13}C_{POC}$ from typical mangrove $\delta^{13}C$ in the mangrove-dominated Sundarbans suggested significant POC modification due to degradation by respiration. The average $p$CO$_2$ in the Hooghly was higher by ~ 1291 μatm compared to the Sundarbans with surface runoff and organic matter degradation by respiration as dominant factors controlling $p$CO$_2$ in the Hooghly and Sundarbans, respectively. The entire Hooghly-Sundarbans system acted as a source of $CO_2$ to the regional atmosphere with ~ 17 times higher emission from the Hooghly compared to the Sundarbans. Taken together, the cycling of C in estuaries having different levels of anthropogenic influences is evidently different with significantly higher $CO_2$ emission from the anthropogenically influenced estuary than the mangrove-dominated ones.

## 1 Introduction

Situated at the interface of land and sea, estuaries are highly susceptible to anthropogenic inputs and undergo intricate biogeochemical and hydrological processes. Estuaries play an important role in modulating global carbon (C) cycle and anthropogenic carbon dioxide ($CO_2$) budget (Bauer et al., 2013; Regnier et al., 2013; LeQuéré et al., 2016). Atmospheric $CO_2$ is sequestered into terrestrial systems through photosynthesis and weathering reactions and is transported to the ocean via rivers and estuaries. Tropical rivers, which constitute ~ 66% of global river water discharge, deliver ~ 0.53 Pg C to the estuaries annually (Huang et al., 2012). The majority of this exported C is in dissolved form [dissolved inorganic C (DIC): 0.21 Pg C $yr^{-1}$ and dissolved organic C (DOC): 0.14 Pg C $yr^{-1}$] with some contribution as particulate [particulate organic C (POC): 0.13 Pg C $yr^{-1}$ and particulate inorganic C (PIC): 0.05 Pg C $yr^{-1}$] (Huang et al., 2012). Although estuaries are only ~ 4% of the continental shelf regions, $CO_2$ emission flux from estuarine surface waters is as high as $CO_2$ uptake in continental shelf regions of the world, albeit with large uncertainty (Borges et al., 2005; Chen and Borges, 2009; Cai et al., 2006; Cai, 2011). This suggests estuaries to be not only active pathway for transport of C (Ittekkot and Laane, 1991) but also a hotspot for biogeochemical modification of labile organic matter (OM) (Frankignoulle et al., 1998).

Mangroves covering 137,760 $km^2$ along tropical and sub-tropical estuaries and coastlines (Giri et al., 2011) are among the most productive natural ecosystems in the world with net primary productivity of $218 \pm 72$ Tg C $yr^{-1}$ (Bouillon et al., 2008). Fine root production coupled with litter fall and wood production are primary sources of mangrove derived C to intertidal forest sediment (Bouillon et al., 2008). The fate of this mangrove derived C remains poorly understood. Despite taking C burial and $CO_2$ emission flux across mangrove sediment-atmosphere interface into account, estimates of global mangrove C budget showed a significant imbalance between mangrove net primary productivity and its sinks (Bouillon et al., 2008). Earlier studies reported mangroves to be responsible for ~ 10% of the global terrestrial derived POC and DOC exports to the coastal zones (Jennerjahn and Ittekkot, 2002; Dittmar et al. 2006). However, recent studies proposed DIC exchange as major C export pathway from mangrove forests, which was ~ 70% of the total mineralized C transport from mangrove forests to coastal waters (Maher et al., 2013; Alongi, 2014; Alongi and Mukhopadhyay, 2014). Another study reported groundwater advection from mangroves to be responsible for 93 - 99% and 89 - 92% of total DIC and DOC exports to the coastal ocean, respectively (Maher et al., 2013). Upon extrapolating these C exports to the global mangrove area, it was found that the calculated C

exports were similar to the missing mangrove C sink (Sippo et al., 2016). The remaining C that
escapes export gets buried in sub-surface sediment layers and participates either in complex
anaerobic processes (linked to production of biogenic trace gases like $CH_4$) or undergoes long-
term sequestration (Jennerjhan and Ittekkot, 2002; Barnes et al., 2006; Kristensen and Alongi,
2006; Donato et al., 2011; Linto et al., 2014).

Apart from lateral transport of dissolved and particulate C, biogeochemical processes
such as primary production, OM mineralization, carbonate precipitation / dissolution and
water-atmosphere $CO_2$ exchange occurring in the estuary also regulate inorganic and organic
C biogeochemistry of a mangrove-dominated estuary. These processes largely depend upon
pH, nutrient availability, euphotic depth variability as well as planktonic and bacterial
biodiversity and community compositions. The biogeochemical cycling of bioavailable
elements, such as C and N, in a mangrove-dominated estuary is largely different from
anthropogenically polluted estuary, where much of the OM is derived from domestic,
agricultural, and industrial wastes. In anthropogenically affected estuarine systems,
heterotrophy generally dominates over autotrophy (Heip et al., 1995; Gattuso et al., 1998) and
a substantial fraction of biologically reactive OM gets mineralized within the system (Servais
et al., 1987; Ittekkot, 1988; Hopkinson et al., 1997; Moran et al., 1999). However, this is not
always the case as observed in the Guanabara Bay, Brazil, which acts as a strong $CO_2$ sink
enhanced by eutrophication (Cotovicz Jr. et al., 2015). Lack of ample rate measurements of
above-mentioned biogeochemical processes in many regions of the world restrains
biogeochemists from an in-depth understanding of these processes in different ecological
settings. It also leads to uncertainty in estimation of coastal C budget on global scale.
In India, research related to C biogeochemistry of estuarine ecosystems have been in
focus since last two decades with emphasis on estuaries located in the southern India (e.g.,
Bouillon et al., 2003; Sarma et al., 2012; Sarma et al., 2014; Bhavya et al., 2017; Bhavya et al.
2018). The estuaries located in the northern part of India have received limited attention,
including adjacently located Hooghly estuary and the estuaries of Sundarbans, which are part
of the Ganga-Brahmaputra river system (Fig. 1). Characteristically, the Hooghly and the
estuaries of Sundarbans are different from each other. The Hooghly estuary experiences
significantly higher anthropogenic influence compared to the mangrove-dominated
Sundarbans as evidenced by high nutrient and freshwater inputs (Table 1). The anthropogenic
influences largely include supply of the industrial effluents and domestic sewage on daily basis
from industries and major cities (Kolkata and Howrah) located upstream (Table 1). The

industries along the Hooghly are principally *jute* (*Corchorus olitorius*) based, which produce fabrics for packaging a wide range of agricultural and industrial commodities.

Earlier, the major focus of biogeochemical studies in the Hooghly and the estuaries of Sundarbans had been on biogeochemistry of trace gases (Mukhopadhyay et al., 2002; Biswas et al., 2004, 2007; Ganguly et al., 2008, 2009; Dutta et al., 2013, 2015, 2017) with exception of one comprehensive study on nutrient budget at the Hooghly estuary (Mukhopadhyay et al., 2006). Recently, attempts have been made to understand different aspects of C cycling in these two estuaries (Samanta et al., 2015; Ray et al., 2015, 2018; Akhand et al., 2016). Samanta et al. (2015) comprehensively studied DIC dynamics in the Hooghly estuary, whereas Akhand et al. (2016) focused on DIC and $pCO_2$ at the Hooghly-Matla estuary. Different aspects of C cycling in the Hooghly-Sundarbans system have been reported by Ray et al. (2015, 2018). Barring Samanta et al. (2015), which has wider spatial and temporal coverages with respect to DIC in the Hooghly, other studies are severely limited in spatial coverage with focus on mid to lower parts of the Hooghly estuary and a few locations in the Sundarbans (one location by Ray et al., 2015, 2018; three locations by Akhand et al., 2016). Given the vast expanse of these estuaries, extrapolation of data from these studies for the entire ecosystem may lead to overestimation/underestimation.

The primary objective of the present study was to understand differences in varied aspects of C cycle (DIC, DOC, POC, and $CO_2$) of the Hooghly and the estuaries of Sundarbans during postmonsoon with relatively better spatial coverage compared to previous studies. The postmonsoon sampling was chosen because of relatively stable estuarine condition for wider spatial coverage and peak mangrove leaf litter fall during this season (Ray et al., 2011), which may have influence on estuarine C dynamics. Considering different nature and quantity of supplied OM within these two contrasting systems, we hypothesized C metabolism in these two estuaries to be very different with higher $CO_2$ exchange flux from anthropogenically influenced estuary compared to the mangrove-dominated one. Specifically, the major aims of the present study were to investigate: (a) factors controlling DIC and DOC dynamics in the region, (b) sources and fate of POC in these two contrasting systems, and (c) partial pressure of $CO_2$ ($pCO_2$) and its controlling mechanisms along with exchange across water-atmosphere interface at the Hooghly-Sundarbans during postmonsoon period.

## 2 Materials and methods

### 2.1 Study area

The present study was carried out in the mangrove dominated estuaries of Indian Sundarbans and anthropogenically dominated Hooghly estuary in the northeastern India. The Sundarbans (21°32' and 22°40'N: 88°05' and 89°E, Fig. 1a), inscribed as a UNESCO world heritage site, is the largest mangrove forest in the world situated at the land-ocean boundary of the Ganges - Brahmaputra delta and the Bay of Bengal (BOB). Out of 10,200 km$^2$ area of the Sundarbans, 41% is in India and the rest is in Bangladesh. The Indian part of Sundarbans (or Sundarbans Biosphere Reserve) contains 4200 km$^2$ of mangrove reserve forest and 1800 km$^2$ of estuarine waterways along with reclaimed areas. The Sundarbans is crisscrossed by several rivers, such as Muriganga, Saptamukhi, Thakuran, Matla, Bidya, Gosaba and Haribhanga, forming a sprawling archipelago of 102 islands covered with thick mangroves mostly composed of *Avicennia alba*, *Avicennia marina* and *Avicennia officinalis*. Semidiurnal tide with mean depth ~ 6 m is general characteristic of the estuary (Dutta et al., 2015).

The second study site, the Hooghly estuary (21°31'-23°20'N and 87°45'- 88°45'E), is the first deltaic offshoot of the Ganges which ultimately mixes with the northern BOB. Like the estuaries of Sundarbans, tides are semidiurnal in nature in the Hooghly as well with variable depth along the channel (~ 21 m at Diamond Harbor (H6) to ~ 8 m at the mouth of the estuary; Fig. 1b) (CIFRI, 2012). Before mixing with the BOB, the lower estuarine part of the Hooghly divides into two channels, one being main estuarine stream which directly mixes with the BOB and another smaller channel known as Muriganga (mean depth ~ 6 m; Sadhuram et al., 2005). The width of the river at the mouth of the estuary is ~ 25 km (Mukhopadhyay et al., 2006). Both estuarine systems experience typical tropical climate having three distinct seasons: premonsoon (February - May), monsoon (June - September) and postmonsoon (October - January) with ~ 80% rainfall during monsoon.

Covering upper, middle, and lower estuarine regions, the present study was carried out during low tide condition in three major estuaries of the Indian Sundarbans [Saptamukhi (S1-S3), Thakuran (T1-T3), and Matla (M1-M3); Fig. 1a] along with its related waterways (S4 & M4). The low-tide postmonsoon sampling was preferred as it was ideal time to evaluate the effect of mangroves on the adjoining estuary due to peak mangrove leaf litter fall (Ray et al., 2011) and groundwater (or pore-water) discharge. To compare and bring out the contrast in different components of the C cycle between mangrove-dominated and anthropogenically

influenced estuaries, low-tide sampling was also performed at 13 locations (H1 - H13, Fig. 1b)
in the Hooghly estuary (stretch: ~150 km).
For the purpose of discussion, henceforth, both the estuarine systems will be discussed
as 'Hooghly-Sundarbans system' and the estuaries of Sundarbans will be called 'Sundarbans'
unless discussed individually.
**2.2 Sampling and experimental techniques**
During postmonsoon (November, 2016), estuarine surface water samples were collected in
duplicate at different locations of the Hooghly-Sundarbans system using Niskin bottle
(Oceantest equipment; capacity: 5 L). A brief description of the on and off field sampling and
experimental techniques used during the present study are described below.
*2.2.1 Sample collection and on board measurements*
Water temperature and pH of the collected samples were measured onboard using thermometer
($\pm$ 0.1 $^{o}$C) and portable pH meter (Orion Star A211) fitted with a Ross type combination
electrode calibrated (as described by Frankignoulle and Borges, 2001) on the NBS scale
(reproducibility: $\pm$ 0.005 pH units). Salinity ($\pm$ 0.1) and dissolved oxygen (DO: $\pm$ 0.1 mg L$^{-1}$)
concentrations were measured onboard following the Mohr-Knudsen and Winkler titration
methods, respectively (Grasshoff et al., 1983). For total alkalinity (TAlk), 50 ml of filtered
(Whatman GF/F filter) estuarine water was titrated onboard in a closed cell using 0.1N HCl
following potentiometric titration method (Bouillon et al., 2003). Uncertainty in TAlk
measurements was $\pm$ 1 $\mu$mol kg$^{-1}$ as estimated using certified reference material (Dickson
standard: CRM-131-0215).
For DIC and $\delta^{13}C_{DIC}$ measurements, estuarine surface waters were collected by gently
overfilling glass vials fitted with teflon septa (Fig. 1). Pore-water was also collected from lower
littoral zone of the Lothian Island (one of the virgin island of the Indian Sundarbans, Fig. 1a)
by digging a hole (~ 30 cm below the water table). It was not possible to collect pore-water
samples from the mid and upper littoral zones of the island due to logistic problems. After
purging water at least twice in the bore, sample was collected from the bottom of the bore
through syringe and transferred to the glass vial (Maher et al., 2013). Twelve groundwater
samples were collected from the nearby locations of the Hooghly-Sundarbans system via tube
pump. After collection, all samples for DIC and $\delta^{13}C_{DIC}$ were preserved immediately by adding
saturated HgCl$_2$ solution to arrest the microbial activity.

222   For both DOC and SPM (suspended particulate matter) measurements, surface water

223 samples were filtered on board through pre-weighted and pre-combusted (500 $^o$C for 6 hours)

224 Whatman GF/F filters (pore size: 0.7 μm). Filtrates were kept for DOC analysis in brown

225 bottles followed by immediate preservation via addition of $H_3PO_4$ (50 μL/15 mL sample)

226 (Bouillon et al., 2003), whereas the residues were kept for particulate matter analysis. Collected

227 DIC, DOC and SPM samples were properly preserved at 4 $^o$C during transportation to the

228 laboratory. Additionally, micrometeorological parameters associated with water-atmosphere

229 $CO_2$ exchange flux computation continuously monitored at 10 m height over the estuary using

230 a portable weather monitor (DAVIS - Vintage Pro2 Plus).

### 2.2.2 Laboratory measurements

233 The DIC concentrations were measured using Coulometer (Model: UIC. Inc. CM - 5130) with

234 analytical uncertainty of ± 0.8%. The $\delta^{13}C_{DIC}$ were measured using Gas Bench attached to a

235 continuous flow mass spectrometer (Thermo Scientific MAT 253) with precision better than

236 0.10‰. The DOC were measured using high-temperature catalytic oxidation analyzer

237 (Shimadzu TOC 5000), which was calibrated using potassium hydrogen phthalate (KHP)

238 solution containing 1, 2, 5, 10, 20 mg $L^{-1}$ of DOC (Ray et al., 2018). The analytical error for

239 DOC measurement was < 2%. For SPM measurement, filter papers containing SPM were dried

240 in hot air oven at 60 $^o$C and final weights were noted. The SPM were calculated based on

241 differences between final and initial weights of the filter paper and volumes of water filtered.

242 For measurements of POC and $\delta^{13}C_{POC}$, filter papers containing SPM were de-carbonated (by

243 HCl fumes) and analyzed using Elemental Analyzer (Flash 2000) attached to the continuous

244 flow mass spectrometer (Thermo Scientific MAT 253) via conflo. The $\delta^{13}C_{POC}$ values are

245 reported relative to V-PDB with reproducibility better than ± 0.10‰, whereas uncertainty for

246 POC was < 10%.

### 2.2.3 Computation of air - water CO₂ flux and %DO

249 The $pCO_2$ were calculated based on surface water temperature, salinity, TAlk, pH and

250 dissociation constants calculated following Millero (2013). The uncertainty for estimated $pCO_2$

251 was ± 1%. The $CO_2$ exchange fluxes (FCO$_2$ in μmol $m^{-2}$ $hr^{-1}$) across water-atmosphere

252 boundary of the estuary were calculated as follows:

$$FCO_2 = k \times K_H^{CO2} \times [pCO_{2\ (water)} - pCO_{2\ (atmosphere)}]$$

Where, $K_H^{CO2}$ = $CO_2$ solubility. 'k' is the gas transfer velocity, which is highly variable and
remains a matter of debate (Raymond and Cole, 2001). The 'k' during the present study was
computed as a function of wind velocity following Liss and Merlivat (1986) parametrization.
For the same wind velocity, the parametrization of Liss and Merlivat (1986) provides least 'k'
value over other parametrization (Wanninkhof, 1992; Raymond and Cole, 2001; Borges et al.,
2004) and therefore, the $FCO_2$ presented during this study may be considered as the
conservative estimates. The wind velocity based 'k' estimation for the Hooghly-Sundarbans
system has been applied in earlier studies as well (Mukhopadhyay et al., 2002; Biswas et al.,
2004). Mean global atmospheric $CO_2$ mixing ratio in dry air during 2016 (data source:
ftp://aftp.cmdl.noaa.gov/products/trends/co2/co2_annmean_gl.txt) was corrected for water
vapor partial pressure to calculate $pCO_{2\ (atmosphere)}$. The fraction, "$K_H^{CO2}$ x [$pCO_{2\ (water)}$ − $pCO_2$
$_{(atmosphere)}$]" is the departure of free dissolved $CO_2$ from atmospheric equilibrium that may be
termed as "excess $CO_2$ ($ECO_2$)" (Zhai et al., 2005).
The % saturation of DO and apparent oxygen utilization (AOU, departure of dissolved $O_2$ from
atmospheric equilibrium) were calculated as follows:
$$\% \text{ saturation of DO} = ([O_2]_{Measured} \text{ x } 100 / [O_2]_{Equilibrium})$$
$$AOU = ([O_2]_{Equilibrium} - [O_2]_{Measured})$$
Where, $[O_2]_{Equilibrium}$ is the equilibrium DO concentration calculated at *in situ* temperature and
salinity (Weiss, 1970) and $[O_2]_{Measured}$ is the measured DO concentration of surface water.
*2.2.4 Mixing model calculation*
Considering salinity as a conservative tracer and an ideal indicator for estuarine mixing
mechanism (Fry, 2002), conservative mixing model was applied to the Hooghly estuary to
understand additions/removals of dissolved and particulate C by *in situ* biogeochemical
processes. Concentrations and stable isotopic compositions of dissolved or particulate C
(presented as C) during conservative mixing ($C_{CM}$ and $\delta^{13}C_{CM}$) were computed as follows
(Carpenter et al., 1975; Mook and Tan, 1991):
$$C_{CM} = C_F F_F + C_M F_M$$
$$S_S [C_F \delta^{13}C_F - C_M \delta^{13}C_M] + S_F C_M \delta^{13}C_M - S_M C_F \delta^{13}C_F$$
$\delta^{13}C_{CM} =$ -----------------------------------------------------------------------------
$$S_S (C_F - C_M) + S_F C_M - S_M C_F$$
Here, 'S' denotes salinity, the suffixes 'CM', 'F', 'M' and 'S' denote conservative mixing,
freshwater end member, marine end member and sample, respectively. $F_F$ = freshwater fraction
= $1 - (S_S / S_M)$ and $F_M$ = marine water fraction = $(1 - F_F)$. $C_{Sample} > C_{CM}$ indicates C addition,
whereas reverse indicates removal. For model calculation, means of salinities, C
concentrations, and $\delta^{13}C$ of samples collected at salinity $\leq 0.3$ at the Hooghly estuary were
considered as end member values for freshwater, whereas respective values for marine end
member were taken from Dutta et al. (2010) and Akhand et al. (2012). Quantitative deviations
($\Delta C$ and $\Delta\delta^{13}C$) of measured C concentrations and $\delta^{13}C$ from the respective conservative
mixing values were estimated as follows (Alling et al., 2012):

$$\Delta C = (C_{Sample} - C_{CM}) / C_{CM}$$

$$\Delta\delta^{13}C = \delta^{13}C_{Sample} - \delta^{13}C_{CM}$$

Plots between $\Delta C$ and $\Delta\delta^{13}C$ for DIC and POC have been used to understand processes
influencing DIC and POC in the Hooghly-Sundarbans system. However, the above model
could not be applied to DOC due to unavailability of $\delta^{13}C_{DOC}$ during the present study.
Unlike the Hooghly, direct application of above-mentioned conservative mixing model
was not justified for the mangrove-dominated Sundarbans due to narrow salinity gradient (see
later). However, assuming that apart from conservative mixing only mangrove derived C
($\Delta C_{Mangrove}$) contributes to estuarine C pool, an approach can be taken to quantify $\Delta C_{Mangrove}$.
Two different mass balance equations as used by Miyajima et al. (2009) for estimating
$\Delta DIC_{Mangrove}$ was extended to calculate $\Delta C_{Mangrove}$ during the present study:

$$\Delta C_{Mangrove} (\Delta C_{M1}) = C_{Sample} - C_{CM}$$

$$\Delta C_{Mangrove} (\Delta C_{M2}) = \frac{C_{Sample} \times [\delta^{13}C_{CM} - \delta^{13}C_{Sample}]}{\delta^{13}C_{CM} - \delta^{13}C_{Mangrove}}$$

For model calculation, $\delta^{13}C_{Mangrove}$ was taken as $-28.4‰$ for Sundarbans (Ray et al., 2015)
and end members were taken as same as the Hooghly as the estuaries of Sundarbans are
offshoot of lower Hooghly estuary.

*2.2.5 Computation of advective DIC input from mangrove forest to estuary*
A first-time baseline value for advective DIC input from mangrove forest sediment to the
adjoining estuary ($F_{DIC}$) via pore-water exchange was calculated following Reay et al. (1995):
$F_{DIC}$ = Sediment porosity x Mean linear velocity x Mean pore water DIC conc.
Mean linear velocity = Pore water specific discharge / Sediment porosity

## 3 Results

### *3.1 Environmental parameters*

During the present study, water temperature did not show any distinct spatial trend and varied from 28 to 29 °C and 30.5 to 33 °C for the Sundarbans (Table 2) and Hooghly (Table 3), respectively. Salinity of the estuaries of Sundarbans varied over a narrow range (12.74 to 16.69; Table 2) with minimum at the upper estuarine locations throughout. A relatively sharp salinity gradient was noticed at the Hooghly estuary (0.04 to 10.37; Table 3). Based on the observed salinity gradient, the Hooghly estuary can be divided into two major salinity regimes: (a) freshwater regime (H1 - H6) and (b) mixing regime (H7 - H13; Fig. 1b). However, due to narrow salinity range, no such classification was possible for the estuaries of Sundarbans. Estuaries of Sundarbans were relatively well-oxygenated (DO = 91 to 104%) compared to the Hooghly (DO = 71 to 104%; Fig. 2). Both pH and TAlk in the Hooghly estuary (pH: 7.31 to 8.29, TAlk: 1797 to 2862 $\mu$eq L$^{-1}$, Table 3) showed relatively wider variation compared to the estuaries of Sundarbans (pH: 8.01 to 8.13, TAlk: 2009 to 2289 $\mu$eq L$^{-1}$; Table 2).

### *3.2 Variability in DIC, $\delta^{13}C_{DIC}$ and DOC*

In the Sundarbans, both DIC and $\delta^{13}C_{DIC}$ varied over a relatively narrow range (DIC = 1683 to 1920 $\mu$M, mean: $1756 \pm 73$ $\mu$M; $\delta^{13}C_{DIC} = -5.93$ to $-4.29$‰, mean: $-5.04 \pm 0.58$‰, Table 2) compared to the Hooghly estuary (DIC = 1678 to 2700 $\mu$M, mean: $2083 \pm 320$ $\mu$M; $\delta^{13}C_{DIC} = -8.61$ to $-5.57$‰, mean: $-6.95 \pm 0.90$‰; Table 3). In the Hooghly, DIC was relatively higher in the freshwater regime compared to the mixing regime, whereas reverse was observed for $\delta^{13}C_{DIC}$. Different estuaries of the Sundarbans showed different trends with Saptamukhi and Thakuran showing maximum and minimum DIC at the upper and lower estuarine regions, respectively, with reverse trend for $\delta^{13}C_{DIC}$. However, for the Matla, no distinct spatial trend was noticed for both DIC and $\delta^{13}C_{DIC}$. In comparison to the estuarine surface waters, markedly higher DIC and lower $\delta^{13}C_{DIC}$ were observed for the groundwater (Hooghly: DIC = 5655 to 11756 $\mu$M, $\delta^{13}C_{DIC} = -12.66$ to $-6.67$‰; Sundarbans: DIC = 7524 to 13599 $\mu$M, $\delta^{13}C_{DIC} = -10.56$ to $-6.69$‰; Table 4) and pore-water samples (Sundarbans: DIC = 13425 $\mu$M; $\delta^{13}C_{DIC} = -18.05$‰; Table 4) collected from the Hooghly-Sundarbans system. The DOC in the Sundarbans varied from 154 to 315 $\mu$M (mean: $235 \pm 49$ $\mu$M; Table 2) with no distinct spatial variability. In comparison, ~ 40% higher DOC was noticed in the Hooghly (235 to 662 $\mu$M; Table 3) reaching peak in the mixing regime.

*3.3 Variability in particulate matter and $\delta^{13}C_{POC}$*
In the Sundarbans, both SPM and POC varied over a wide range (SPM = 80 to 741 mg L$^{-1}$,
mean: 241 ± 197 mg L$^{-1}$; POC = 80 to 436 µM, mean: 173 ± 111 µM; Table 2) with no distinct
spatial variability. Compared to that, SPM and POC in the Hooghly were relatively lower and
varied from 38 to 289 mg L$^{-1}$ and 95 to 313 µM (Table 3), respectively; reaching maximum at
the freshwater regime. The $\delta^{13}C_{POC}$ of the Sundarbans varied from – 23.82 to – 22.85‰ (mean:
– 23.36 ± 0.32‰), whereas in the Hooghly it varied from – 26.28 to – 23.47‰ (mean: – 24.87
± 0.89‰).

*3.4 Variability in pCO$_2$ and FCO$_2$*
In the Sundarbans, surface water $p$CO$_2$ varied from 376 to 561 µatm (mean: 464 ± 66 µatm;
Table 2) with no spatial pattern. Compared to the Sundarbans, ~ 3.8 times higher $p$CO$_2$ was
estimated in the Hooghly estuary (267 to 4678 µatm; Table 3) reaching its peak in the
freshwater regime. Except one location at the Sundarbans (M2: – 42 µM) and two locations in
the mixing regime at the Hooghly (H12: – 3.26 µM; H13: – 3.43 µM), ECO$_2$ values were
always positive in the Hooghly-Sundarbans system. The calculated FCO$_2$ at the Hooghly
estuary (– 19.8 to 717.5 µmol m$^{-2}$ hr$^{-1}$; mean: 231 µmol m$^{-2}$ hr$^{-1}$; Table 3) was ~ 17 times higher
than the mangrove dominated estuaries of the Indian Sundarbans (FCO$_2$: – 2.6 to 30.3 µmol m$^{-2}$
$^{-2}$ hr$^{-1}$; Table 2). Spatially, in the Hooghly, higher FCO$_2$ was noticed in the freshwater regime
(285.2 to 717.5 µmol m$^{-2}$ hr$^{-1}$) compared to the mixing regime, while no such distinct spatial
trend was observed at the Sundarbans.

**4 Discussion**
Based on the results obtained during the present study, below we discuss different aspects of
C cycle within the Hooghly-Sundarbans system.
*4.1 Major drivers of DIC dynamics*
DIC concentrations observed in this study for the Hooghly were higher than that reported by
Samanta et al. (2015) for the same season (DIC: 1700 to 2250 µM), whereas observed $\delta^{13}C_{DIC}$
were within their reported range ($\delta^{13}C_{DIC}$: – 11.4 to – 4.0‰). Statistically significant
correlations between DIC - salinity (r$^2$ = 0.43, p = 0.015) and $\delta^{13}C_{DIC}$ - salinity (r$^2$ = 0.58, p =
0.003) in the Hooghly suggested potential influence of marine and freshwater mixing on DIC
and $\delta^{13}C_{DIC}$ in the estuary (Fig. 3a & 3b), rationalizing the application of two end member
mixing model. Application of two end member mixing model to decipher processes influencing
DIC chemistry has been done earlier in the Hooghly estuary (Samanta et al., 2015).

Based on the methodology discussed earlier, calculated $\Delta C$ for DIC ($\Delta DIC \sim -0.27$ to

0.17) predicted dominance of DIC addition (n = 4) over removal (n = 2) in the freshwater
regime of the Hooghly, whereas only removal was evident in the mixing regime. In case of
$\Delta \delta^{13}C$ for DIC ($\Delta \delta^{13}C_{DIC}$), values were mostly positive (n = 9), i.e., measured $\delta^{13}C_{DIC}$ was
higher compared to estimated $\delta^{13}C_{DIC}$ due to conservative mixing. Deviation plot ($\Delta DIC$ vs.
$\Delta \delta^{13}C_{DIC}$; Fig. 3c) for samples of the Hooghly showed following patterns: (a) decrease in $\Delta DIC$
with increasing $\Delta \delta^{13}C_{DIC}$ (n = 5) indicating phytoplankton productivity and/or outgassing of
$CO_2$ across water-atmosphere interface, (b) decrease in $\Delta DIC$ with decreasing $\Delta \delta^{13}C_{DIC}$ (n = 4)
indicating carbonate precipitation, and (c) increase of $\Delta DIC$ with increasing $\Delta \delta^{13}C_{DIC}$ (n = 4)
representing carbonate dissolution within the system.

Based on these calculations, both organic and inorganic processes (productivity,

carbonate precipitation and dissolution) along with physical processes ($CO_2$ outgassing across
water-atmosphere interface) appeared to regulate DIC chemistry in the Hooghly estuary.
Spatially, phytoplankton productivity and/or outgassing of $CO_2$ appeared to regulate DIC in the
mixing regime (n = 5 out of 7) of the Hooghly. Earlier studies have advocated for high
phytoplankton productivity in non-limiting nutrient condition during postmonsoon in the
Hooghly (Mukhopadhyay et al., 2002; Mukhopadhyay et al., 2006). However, based on the
present data, particularly due to lack of direct primary productivity measurements, it was
difficult to spatially decouple individual contributions of primary productivity and $CO_2$
outgassing in the mixing regime. In contrast to the mixing regime, carbonate precipitation and
dissolution appeared to be dominant processes affecting DIC chemistry in the freshwater
regime of the Hooghly.

In mangrove-dominated estuaries of the Sundarbans, observed $\delta^{13}C_{DIC}$ during this study

were within the range ($\delta^{13}C_{DIC}: -4.7 \pm 0.7‰$) reported by Ray et al. (2018), whereas observed
DIC concentrations were lower than their estimates (DIC: $2130 \pm 100$ µmol kg$^{-1}$). Our data
also showed similarity with Khura and Trang river, two mangrove-dominated rivers of
peninsular Thailand flowing towards Andaman sea, although from hydrological prospective
these two systems are contrasting in nature [Sundarbans: narrow salinity gradient (12.74 to
16.69) vs. Khura and Trang river: sharp salinity gradient (~ 0 to 35); Miyajima et al., 2009].
Like Hooghly, $\delta^{13}C_{DIC}$ - salinity relationship was statistically significant ($r^2 = 0.55$, $p = 0.009$)
for the Sundarbans, but DIC - salinity relationship remained insignificant (p = 0.18) (Fig. 3d &
3e).

Given the dominance of mangroves in the Sundarbans, the role of mangrove derived
organic carbon (OC) mineralization may be important in regulating DIC chemistry in this
ecosystem. Theoretically, $\Delta C_{Mangrove}$ for DIC ($\Delta DIC_{Mangrove}$) estimated based on DIC ($\Delta DIC_{M1}$)
and $\delta^{13}C_{DIC}$ ($\Delta DIC_{M2}$) should be equal. The negative and unequal values of $\Delta DIC_{M2}$ (– 41 to
62 μM) and $\Delta DIC_{M1}$ (– 186 to 11 μM) indicate large DIC out-flux over influx through
mineralization of mangrove derived OC in this tropical mangrove system. The removal
mechanisms of DIC include $CO_2$ outgassing across estuarine water-atmosphere boundary,
phytoplankton uptake and export to the adjacent continental shelf region (northern BOB, Ray
et al., 2018). The evidence for $CO_2$ outgassing was found at almost all locations covered during
the present study (10 out of 11 locations covered; see section 4.4). Also, a recent study by Ray
et al. (2018) estimated DIC export (~ 3.69 Tg C yr$^{-1}$) from the estuaries of Sundarbans as the
dominant form of C export. Although data for primary productivity is not available for the
study period, earlier studies have reported postmonsoon as peak season for phytoplankton
productivity (Biswas et al., 2007; Dutta et al., 2015). Given the evidences for presence of DIC
removal processes in the Sundarbans, a comprehensive study that measures rates of these
processes with higher spatial and temporal coverages is desirable to understand the balance
between influx and out-flux of DIC in the Sundarbans.

Other than biogeochemical processes, factors such as groundwater and pore-water
exchange to the estuary might also play a significant role in estuarine DIC chemistry (Tait et
al., 2016). High $p$CO$_2$ and DIC along with low pH and TAlk/DIC are general characteristics of
groundwater, specially within carbonate aquifer region (Cai et al., 2003). Although all the
parameters of groundwater inorganic C system (like pH, TAlk and $p$CO$_2$) were not measured
during the present study, groundwater DIC were ~ 5.57 and ~ 3.61 times higher compared to
mean surface water DIC in the Sundarbans and Hooghly, respectively. The markedly higher
DIC in groundwater as well as similarity in its isotopic composition with estuarine DIC may
stand as a signal for influence of groundwater on estuarine DIC, with possibly higher influence
at the Sundarbans than Hooghly as evident from the slope of the TAlk - DIC relationships
(Hooghly: 0.98, Sundarbans: 0.03). In the Sundarbans, to the best of our knowledge, no report
exists regarding groundwater discharge. Contradictory reports exist for the Hooghly, where
Samanta et al. (2015) indicated groundwater contribution at low salinity regime (salinity < 10,
same as our salinity range) based on 'Ca' measurement, which was not observed based on 'Ra'

isotope measurement in an earlier study (Somayajulu et al., 2002). Pore-water DIC in the Sundarbans was ~ 7.63 times higher than the estuarine water, indicating possibility of DIC input from the adjoining mangrove system to the estuary through pore-water exchange depending upon changes in hypsometric gradient during tidal fluctuation (i.e., tidal pumping). Using pore-water specific discharge and porosity as 0.008 cm min$^{-1}$ and 0.58 (Dutta et al., 2013, Dutta et al., 2015), respectively during postmonsoon and extrapolating the flux value over daily basis (i.e., for 12 hours as tides are semidiurnal in nature), mean $F_{DIC}$ during postmonsoon was calculated as ~ 770.4 mmol m$^{-2}$ d$^{-1}$. However, significant impact of pore-water on DIC may be limited only in mangrove creek water (samples not collected) as evident from narrow variability of estuarine TAlk and DIC as well as no significant correlation between them (p = 0.93). A comprehensive investigation that measures rates of ground and pore waters mediated DIC additions is needed to thoroughly understand their importance in controlling DIC chemistry of the Hooghly-Sundarbans system.

From the above discussion, it appears that higher DIC in the Hooghly compared to the Sundarbans may be due to cumulative interactions between freshwater content to the individual estuaries as well as degree of biogeochemical and hydrological processes. Relatively higher freshwater contribution in the Hooghly compared to the Sundarbans (as evident from salinity) as well as significant negative relationship between DIC - salinity proved significant impact of freshwater on DIC pool in the Hooghly. However, quantifications of other biogeochemical and hydrological processes are needed to decipher dominant processes affecting DIC dynamics in the Hooghly-Sundarbans system.

*4.2 DOC in the Hooghly-Sundarbans*

In the Hooghly, DOC concentrations observed during this study were higher than the range (226.9 ± 26.2 to 324 ± 27 µM) reported by Ray et al. (2018), whereas observed DOC in the Sundarbans were comparable with their estimates (262.5 ± 48.2 µM). The marine and fresh water mixing did not appear to exert major control over DOC in the Hooghly-Sundarbans system as evident from lack of significant correlations between DOC and salinity (Hooghly freshwater regime: $r^2 = 0.33$, p = 0.23; Hooghly mixing regime: $r^2 = 0.10$, p = 0.50; Sundarbans: $r^2 = 0.27$, p = 0.10, Fig. 4a). Our observation showed similarity with other Indian estuaries (Bouillon et al., 2003) with opposite reports from elsewhere (Raymond and Bauer, 2001, Abril et al., 2002). This indicates that DOC in this sub-tropical estuarine system is principally controlled by processes other than mixing of two water masses.

Although it is difficult to accurately decipher processes influencing DOC without $\delta^{13}C_{DOC}$ data, some insights may be obtained from estimated $\Delta C$ of DOC ($\Delta DOC$). The estimated $\Delta DOC$ in the Hooghly indicated both net addition (n = 3) and removal (n = 3) of DOC in the freshwater regime ($\Delta DOC = -0.16$ to $0.11$); whereas, only net addition was evident throughout the mixing regime ($\Delta DOC = 0.08$ to $1.74$). In the Sundarbans, except lower Thakuran (St. T3, $\Delta DOC_{M1} = -20\ \mu M$), net addition of mangrove derived DOC was estimated throughout ($\Delta DOC_{M1} = 2$ to $134\ \mu M$).

In an estuary, DOC can be added through *in situ* production (by benthic and pelagic primary producers), lysis of halophobic freshwater phytoplankton cells and POC dissolution. DOC can be removed through bacterial mineralization, flocculation as POC, and photo-oxidation (Bouillon et al., 2006). At the Hooghly - Sundarbans system, no evidence for freshwater phytoplankton ($\delta^{13}C$: $-33$ to $-40$‰; Freitas et al., 2001) was found from $\delta^{13}C_{POC}$, ruling out its potential effect on DOC. Although an indirect signal for phytoplankton productivity was observed in the freshwater regime from $\delta^{13}C_{DIC}$ and POC relationship ($r^2 = 0.68$, p = 0.05), further evaluation of its impact on DOC was not possible due to lack of direct measurement. Contradictory results exist regarding influence of phytoplankton productivity on DOC. Some studies did not find direct link between DOC and primary productivity (Boto and Wellington, 1988), whereas significant contribution of phytoplankton production to build DOC pool ($\sim 8$ to $40\%$) has been reported by others (Dittmar and Lara, 2001; Kristensen and Suraswadi, 2002).

In a nutrient rich estuary like Hooghly, lack of significant relationship between DOC - $p$CO$_2$ (freshwater regime: p = 0.69, mixing regime: p = 0.67, Fig. 4b) suggested either inefficient bacterial DOC mineralization or significant DOC mineralization compensated by phytoplankton CO$_2$ uptake. However, significant positive relationship between these two in the Sundarbans ($r^2 = 0.45$, p = 0.02, Fig. 4c) indicated increase in aerobic bacterial activity with increasing DOC. In mangrove ecosystems, leaching of mangrove leaf litter as DOC is fast as $\sim 30\%$ of mangrove leaf litter leaching as DOC is reported within initial 9 days of degradation (Camilleri and Ribi, 1986). In the Sundarbans, mangrove leaf litter fall peaks during postmonsoon (Ray et al. 2011) and its subsequent significant leaching as DOC was evident during the present study from relatively higher DOC compared to POC (DOC:POC = 0.50 to 3.39, mean: $1.79 \pm 0.94\%$). Our interpretation for Sundarbans corroborated with that reported by Ray et al. (2018) for the same system as well as Bouillon et al. (2003) for the Godavari estuary, South India.

Despite high water residence time in the Hooghly (~ 40 days during postmonsoon; Samanta et al., 2015) and in mangrove ecosystem like the Sundarbans (Alongi et al., 2005, Singh et al., 2016), DOC photo-oxidation may not be so potent due to unstable estuarine condition in the Hooghly-Sundarbans system (Richardson number < 0.14) having intensive vertical mixing with longitudinal dispersion coefficients of 784 $m^2 s^{-1}$ (Goutam et al., 2015, Sadhuram et al., 2005). The unstable condition may not favor DOC - POC interconversion as well but mediated by charged complexes and repulsion-attraction interactions, the interconversion partly depends upon variation in salinity. More specifically, the interconversion is efficient during initial mixing of fresh (river) and seawater and the coagulation mostly completes within salinity range 2 - 3. This appeared to be the case in the Hooghly, where DOC and POC were negatively correlated in the freshwater regime ($r^2 = 0.86$, $p = 0.007$, Fig. 4d) but not in the mixing regime ($p = 0.43$) or in the Sundarbans ($p = 0.84$).

Although estimated $\Delta$DOC indicated largely net DOC addition to the Hooghly-Sundarbans system, except leaf litter leaching in the Sundarbans, no significant evidence for other internal sources was found. This suggested potential contribution from external sources that may include industrial effluents and municipal wastewater discharge (i.e., surface runoff) in the freshwater regime of the Hooghly (Table 1). However, there is no direct DOC influx data available to corroborate the same. Relatively higher DOC compared to POC (DOC/POC > 1) at some locations (H2, H5, H6) of the freshwater regime may stand as a signal for higher DOC contribution at those locations but it is not prudent to pinpoint its sources due to lack of isotopic data. Considering significantly high DOC levels in wastewater effluent (Katsoyiannis and Samara, 2006, 2007) along with fast degradation of biodegradable DOC (~ 80% within 24 hours; Seidl et al., 1998) and residence time of Hooghly water (mentioned earlier), Samanta et al. (2015) suggested possibility of anthropogenic DOC biodegradation during its transport in the estuary. Although anthropogenic inputs were mostly confined to the freshwater regime, relatively higher DOC in the mixing regime of the Hooghly compared to the freshwater regime suggested DOC input via some additional pathway, possibly groundwater discharge. The contribution of groundwater to the Hooghly estuary within the salinity range observed during the present study has been reported (Samanta et al., 2015). However, there is no report of groundwater mediated DOC influx to the estuary. For mangrove-dominated ecosystems like the Sundarbans, a recent study by Maher et al. (2013) estimated ~ 89 - 92% of the total DOC export to be driven by groundwater advection. To understand spatial variability of DOC chemistry in the Hooghly-Sundarbans system, a thorough investigation that measures rates of groundwater and surface runoff mediated DOC additions is warranted.

Overall, on an average, the concentration of DOC in the Hooghly was ~ 40% higher

than in the Sundarbans, which appeared to be due to cumulative effect of contributions from
freshwater and groundwater, higher anthropogenic inputs, and DOC - POC interconversion.
However, DOC inputs via other pathways may be dominant over freshwater mediated input as
evident from insignificant DOC - salinity relationship during the present study. To
quantitatively understand the relative control of the above-mentioned contributors to the DOC
pool in the Hooghly-Sundarbans system, the individual components need to be studied in detail.

*4.3 Major drivers of particulate organic matter*
The average POC during this study was relatively higher than the range (Hooghly: $40.3 \pm 1.1$
to $129.7 \pm 6.7$ µM, Sundarbans: $45.4 \pm 7.5$ µM) reported by Ray et al. (2018) for the Hooghly-
Sundarbans system. However, it was within the range (51 to 750 µM; Sarma et al., 2014)
reported for a large set of Indian estuaries. No significant SPM - salinity or POC - salinity
relationships were observed during the present study (Fig. 5a & 5b), except for a moderate
negative correlation between POC and salinity ($r^2 = 0.62$, $p = 0.06$) in the freshwater regime of
the Hooghly. This inverse relationship may be linked to freshwater mediated POC addition.
Also, as described earlier, contribution of POC via surface runoff is also a possibility in this
regime due to presence of several industries and large urban population (St: H2: Megacity
Kolkata) that discharge industrial effluents and municipal wastewater to the estuary on regular
basis (Table 1). A signal for surface runoff mediated POC addition was evident in the
freshwater regime where ~ 61% and ~ 43% higher POC were observed at 'H3' and 'H4',
respectively compared to an upstream location (St. H2). However, based on the present data, it
was not possible to decouple freshwater and surface runoff mediated POC inputs to the
Hooghly estuary. Relatively lower contribution of POC to the SPM pool of the Sundarbans
(0.66 to 1.23%) compared to the Hooghly (0.96 to 4.22%; Fig. 5c) may be due to low primary
production owing to high SPM load (Ittekkot and Laane, 1991) as observed in the mangrove-
dominated Godavari estuary in the southern India (Bouillon et al., 2003).

In general, wide ranges for $\delta^{13}C$ (rivers ~ – 28 to – 25‰; marine plankton ~ – 22 to –

18‰; $C_3$ plant ~ – 32 to – 24‰; $C_4$ plants ~ – 13 to –10‰; freshwater algae and their detritus
~ – 30 to – 40‰) have been reported in ecosystem (Smith and Epstein, 1971; Cerling et al.,
1997; Bouillon et al., 2003; Bontes et al., 2006; Kohn, 2010; Marwick et al., 2015). In the
Hooghly, our measured $\delta^{13}C_{POC}$ suggested influx of POC via freshwater runoff as well as
terrestrial $C_3$ plants. Additionally, the estuary was also anthropogenically stressed during
postmonsoon with measured $\delta^{13}C_{POC}$ within the range reported for sewage ($\delta^{13}C_{POC} \sim -28$ to
$-14‰$, Andrews et al., 1998; $\delta^{13}C_{DOC} \sim -26‰$, Jin et al., 2018). In the mixing regime of the
Hooghly, significantly lower $\delta^{13}C_{POC}$ at 'H11' and 'H12' compared to other sampling locations
may be linked to localized $^{13}C$ depleted organic C influx to the estuary from adjacent
mangroves and anthropogenic discharge, respectively.

In the estuaries of Sundarbans, isotopic signatures of POC showed similarity with

terrestrial $C_3$ plants. Interestingly, despite being mangrove-dominated estuary (salinity: 12.74
to 16.55) no clear signature of either freshwater or mangrove ($\delta^{13}C$: mangrove leaf $\sim -28.4‰$,
soil $\sim -24.3‰$, Ray et al., 2015, 2018) borne POC was evident from $\delta^{13}C_{POC}$ values, suggesting
towards the possibility of significant POC modification within the system. Modification of
POC within the estuaries of Indian sub-continent has been reported earlier (Sarma et al., 2014).
Inter-estuary comparison revealed relatively lower average $\delta^{13}C_{POC}$ at the Hooghly (mean
$\delta^{13}C_{POC}$: $-24.87 \pm 0.89‰$) compared to the Sundarbans (mean $\delta^{13}C_{POC}$: $-23.36 \pm 0.32‰$),
which appeared to be due to differences in degree of freshwater contribution, anthropogenic
inputs (high in Hooghly vs. little/no in Sundarbans), nature of terrestrial $C_3$ plant material
(mangrove in the Sundarbans vs. others in Hooghly) as well as responsible processes for POC
modification within the system.

To decipher processes involved in POC modification, estimated $\Delta C$ for POC ($\Delta POC$)

in the Hooghly indicated both net addition (n = 3) and removal (n = 3) of POC in the freshwater
regime ($\Delta POC = -0.45$ to 0.48), whereas removal (n = 6) dominated over addition (n = 1) in
the mixing regime ($\Delta POC = -0.39$ to 0.07). In an estuary, POC may be added through
freshwater and surface runoff mediated inputs, phytoplankton productivity, and DOC
flocculation. The removal of POC is likely due to settling at subtidal sediment, export to the
adjacent continental shelf region, modification via conversion to DOC and degradation by
respiration in case of oxygenated estuary.

The plot between $\Delta\delta^{13}C$ for POC ($\Delta\delta^{13}C_{POC}$) and $\Delta POC$ (Fig. 5d) indicated different

processes to be active in different regimes of the Hooghly estuary. Decrease in $\Delta POC$ with
increase in $\Delta\delta^{13}C_{POC}$ (n = 4 for the mixing regime and n = 1 for the freshwater regime)
suggested degradation of POC by respiration. This process did not appear to significantly
impact estuarine $CO_2$ pool as evident from the POC - $pCO_2$ relationship (freshwater regime: p
= 0.29, mixing regime: p = 0.50; Fig. 5e). Decrease in both $\Delta POC$ and $\Delta\delta^{13}C_{POC}$ (n = 2 for
mixing regime and n = 2 for freshwater regime) supported settling of POC to sub-tidal
sediment. Despite high water residence time, this process may not be effective in the Hooghly
due to unstable estuarine condition (described earlier). Increase in $\Delta POC$ with decrease in
$\Delta\delta^{13}C_{POC}$ (n = 2 for the freshwater regime) indicated POC inputs via surface and freshwater
runoffs as well as phytoplankton productivity. Increase in both $\Delta POC$ and $\Delta\delta^{13}C_{POC}$ (n = 1 for
the mixing regime and n = 1 for the freshwater regime) may be linked to DOC to POC
conversion by flocculation.

In the Sundarbans, negative and lower $\Delta POC_{M2}$ (– 209 to – 28 µM) compared to

$\Delta POC_{M1}$ (– 35 to 327 µM) suggested DIC like behavior, i.e., simultaneous removal or
modification along with addition of mangrove derived POC. No evidence for *in situ* POC -
DOC exchange was obvious based on POC - DOC relationship; however, signal for
degradation of POC by respiration was evident in the Sundarbans from POC - $pCO_2$
relationship ($r^2 = 0.37$, p = 0.05, Fig. 5f). Similar to the Hooghly, despite high water residence
time in mangroves (Alongi et al., 2005; Singh et al., 2016), unstable estuarine condition may
not favor efficient settlement of POC at sub-tidal sediment. The export of POC from the
Hooghly-Sundarbans system to the northern BOB, without significant *in situ* modification, is
also a possibility. This export has been estimated to be ~ 0.02 to 0.07 Tg and ~ 0.58 Tg annually
for the Hooghly and Sundarbans, respectively (Ray et al., 2018).

*4.4 pCO$_2$ and FCO$_2$ in the Hooghly-Sundarbans*
The estimated $pCO_2$ for the Hooghly-Sundarbans system during this study were in the range
(Cochin estuary: 150 to 3800 µatm, Gupta et al., 2009; Mandovi - Zuari estuary: 500 to 3500
µatm, Sarma et al., 2001) reported for other tidal estuaries of India. In the Sundarbans, barring
three locations (S3, T3 and M2), a significant negative correlation between $pCO_2$ and %
saturation of DO ($r^2 = 0.76$, p = 0.005; Figure not given) suggested presence of processes, such
as degradation of OM by respiration, responsible for controlling both $CO_2$ production and $O_2$
consumption in the surface estuarine water. Furthermore, significant positive correlation
between $ECO_2$ and AOU ($ECO_2 = 0.057AOU + 1.22$, $r^2 = 0.76$, p = 0.005, n = 8; Fig. 6a)
confirmed the effect of OM degradation by respiration on $CO_2$ distribution, particularly in the
upper region of the Sundarbans. Our observations were in agreement with a previous study in
the Sundarbans (Akhand et al., 2016) as well as another sub-tropical estuary, Pearl River
estuary, China (Zhai et al., 2005). However, relatively lower slope for $ECO_2$ - AOU
relationship (0.057) compared to the slope for Redfield respiration in $HCO_3^-$ rich environment
$[(CH_2O)_{106}(NH_3)_{16}H_3PO_4 + 138O_2 + 18HCO_3^{2-} \rightarrow 124CO_2 + 140H_2O + 16NO_3^- + HPO_4^{2-};$
$\Delta CO_2: (-\Delta O_2) = 124/138 = 0.90$, Zhai et al., 2005] suggested lower production of $CO_2$ than
expected from Redfield respiration. This may be linked to formation of low molecular weight
OM instead of the final product ($CO_2$) during aerobic OM respiration (Zhai et al., 2005).
Moreover, $pCO_2$ - salinity relationship (p = 0.18, Fig. 6b) confirmed no significant effect of
fresh and marine water contribution on variability of $pCO_2$ in the Sundarbans. Other potential
source of $CO_2$ to the mangrove-dominated Sundarbans could be groundwater (or pore water)
exchange across intertidal mangrove sediment-water interface. Although based on our own
dataset, it is not possible to confirm the same. However, relatively higher $pCO_2$ levels during
low-tide compared to high-tide at Matla estuary in the Sundarbans (Akhand et al. 2016) as well
as in other estuarine mangrove systems worldwide (Bouillon et al., 2007, Call et al., 2015,
Rosentreter et al., 2018) suggested groundwater (or pore water) exchange to be a potential $CO_2$
source in such systems.
Unlike the Sundarbans, $ECO_2$ - AOU relationship did not confirm significant impact of
OM degradation by respiration on $CO_2$ in either freshwater (p = 0.50) or mixing regimes (p =
0.75) of the Hooghly (Fig. 6c). Overall, $pCO_2$ in the freshwater regime of the Hooghly was
significantly higher compared to the mixing regime (Table 3), which may be linked to
additional $CO_2$ supply in the freshwater regime via freshwater or surface runoffs from adjoining
areas (Table 1). Inter-estuary comparison of $pCO_2$ also revealed higher average $pCO_2$ in the
Hooghly by ~ 1291 μatm compared to the Sundarbans, which was largely due to significantly
higher $pCO_2$ in freshwater regime of the Hooghly (Table 2 & 3). Lack of negative correlation
between $pCO_2$ - salinity in freshwater regime (Fig. 6d) of the Hooghly suggested limited
contribution of $CO_2$ due to freshwater input. Therefore, $CO_2$ supply via surface runoff may be
primary reason for higher $pCO_2$ in the Hooghly estuary.
Positive mean $FCO_2$ clearly suggested the Hooghly-Sundarbans system to be a net
source of $CO_2$ to the regional atmosphere during postmonsoon (Fig. 6e & 6f). Specifically,
from regional climate and environmental change perspectives, anthropogenically influenced
Hooghly estuary was a relatively greater source of $CO_2$ to the regional atmosphere compared
to the mangrove-dominated Sundarbans ([$FCO_2$] $_{Hooghly}$: [$FCO_2$] $_{Sundarbans}$ = 17). However,
despite being a $CO_2$ source, $FCO_2$ measured for the estuaries of Sundarbans were considerably
lower compared to global mean $FCO_2$ reported for the mangrove-dominated estuaries (~ 43 to
59 mmol C m$^{-2}$ d$^{-1}$; Call et al., 2015). Similarly, $FCO_2$ measured for the Hooghly estuary were
relatively lower compared to some Chinese estuarine systems (Pearl River inner estuary: 46
mmol m$^{-2}$ d$^{-1}$, Guo et al., 2009; Yangtze River estuary: 41 mmol m$^{-2}$ d$^{-1}$, Zhai et al., 2007).
The difference in $FCO_2$ between the Hooghly and Sundarbans may be due to variability
in $pCO_2$ level as well as micrometeorological and physicochemical parameters controlling gas
transfer velocity across water-atmosphere interface. Quantitatively, the difference in 'k' values
for the Hoogly and Sundarbans were not large (k $_{Sundarbans}$ – k $_{Hooghly}$ ~ 0.031 cm hr$^{-1}$). Therefore,
large difference in FCO$_2$ between these two estuarine systems may be due to difference in
$p$CO$_2$. Taken together, supporting our hypothesis, it appears that differences in land use and
degrees of anthropogenic influence have the potential to alter the C biogeochemistry of aquatic
ecosystems with anthropogenically stressed aquatic systems acting as a relatively greater
source of CO$_2$ to the regional atmosphere than mangrove-dominated ones.

**5. Conclusions**
The present study focused on investigating different aspects of C biogeochemistry of the
anthropogenically affected Hooghly estuary and mangrove dominated estuaries of the
Sundarbans during postmonsoon. Considering different nature and quantity of supplied organic
matter within these two contrasting systems, it was hypothesized in this study that C
metabolism in these two estuaries was different with higher CO$_2$ exchange flux from the
anthropogenically influenced estuary compared to the mangrove-dominated one. The results
obtained during the study supported this hypothesis with significant differences in
physicochemical parameters and active biogeochemical processes in these two estuaries. While
freshwater intrusion along with inorganic and organic C metabolisms appeared to shape DIC
dynamics in the Hooghly, significant DIC removal (via CO$_2$ outgassing, phytoplankton uptake
as well as export to adjoining continental shelf region) and influence of groundwater were
noticed in the Sundarbans. Relatively higher DOC concentration in the Hooghly compared to
the Sundarbans was due to cumulative interactions among anthropogenic inputs, DOC-POC
interconversion, and groundwater contribution. Freshwater runoff, terrestrial C$_3$ plants, and
anthropogenic inputs contributed to POC pool in the Hooghly, whereas contribution from C$_3$
plants was dominant at the Sundarbans. Surface runoff from adjoining areas in the Hooghly
and degradation of OM by respiration in the Sundarbans largely controlled $p$CO$_2$ in the system.
Overall, the entire Hooghly-Sundarbans system acted as source of CO$_2$ to the regional
atmosphere with ~ 17 times higher emission from the Hooghly compared to the Sundarbans,
suggesting significant role played by anthropogenically stressed estuarine system from regional
climate change perspective.

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

**Data availability**

Data used in the manuscript is presented in tables (Table 2, Table 3, and Table 4) of the manuscript.

**Author contributions**

MKD and SK designed the study. MKD with RM and PS collected and analyzed samples. MKD and SK interpreted the data and drafted the manuscript. SKM provided facility to measure basic physicochemical parameters and DOC.

**Competing interest**

The author declares no conflict of interest.

**Acknowledgment**

MKD is thankful to Physical Research Fellowship (PRL) postdoctoral fellowship program for providing fellowship. Authors are thankful to ISRO-GBP for financial support and Sundarbans Biosphere Reserve for their permission to carry out the sampling. Thanks to Ms. R. Mukherjee and Ms. A. Acharya for their help during field observations. We also thank two anonymous reviewers and the associate editor for valuable comments, which significantly improved the quality of the manuscript.

Table - 1: General characteristics of the Hooghly estuary and the estuaries of Sundarbans.

| Parameters | Hooghly | Sundarbans |
|---|---|---|
| Nutrients (postmonsoon) | DIN: $14.72 \pm 1.77$ to $27.20 \pm 2.05$ µM<br>DIP: $1.64 \pm 0.23$ to $2.11 \pm 0.46$ µM<br>DSi: $77.75 \pm 6.57$ to $117.38 \pm 11.54$ µM<br>(Mukhopadhyay et al., 2006) | DIN: $11.70 \pm 7.65$ µM<br>DIP: $1.01 \pm 0.52$ µM<br>DSi: $75.9 \pm 36.9$ µM<br>(Biswas et al., 2004) |
| Chl *a* (postmonsoon) | $2.35 - 2.79$ mg m$^{-3}$<br>(Mukhopadhyay et al., 2006) | $7.88 \pm 1.90$ mg m$^{-3}$<br>(Dutta et al., 2015) |
| Population density | North 24 Parganas and Hooghly: 2500 km$^{-2}$,  Kolkata: 22000 km$^{-2}$, Howrah: 3300 km$^{-2}$, South 24 Parganas: 820 km$^{-2}$ | No major Cities and town |
| Freshwater discharge (postmonsoon) | $3070 - 7301$ million m$^3$<br>(Rudra et al., 2014) | No information available |
| Catchment area | $6 \times 10^4$ km$^2$<br>(Sarkar et al., 2017) | No information available |
| Industrial and municipal wastewater discharge | 1153.8 million L d$^{-1}$<br>(Ghosh, 1973; Khan, 1995) | No information available |
| Dissolved metal flux | Increased from $230 - 1770\%$ annually<br>(Samanta and Dalai, 2018) | No information available |










Table - 2: Physicochemical parameters, inorganic and organic C related parameters, and $CO_2$
exchange flux across water-atmosphere interface at the estuaries of Sundarbans. Here, $W_T$ =
water temperature, DO = dissolved oxygen.

| Station | $W_T$ (ºC) | Salinity | DO (mgL$^{-1}$) | pH | DIC (μM) | $\delta^{13}C_{DIC}$ (‰) | DOC (μM) | POC (μM) | $\delta^{13}C_{POC}$ (‰) | $p$CO$_2$ (μatm) | FCO$_2$ (μmol m$^{-2}$ hr$^{-1}$) |
|---|---|---|---|---|---|---|---|---|---|---|---|
| S1 | 28.50 | 12.74 | 6.65 | 8.02 | 1780 | − 5.59 | 278 | 154 | − 22.85 | 536 | 26.5 |
| S2 | 28.00 | 16.02 | 6.65 | 8.02 | 1703 | − 4.33 | 267 | 124 | − 23.54 | 561 | 30.3 |
| S3 | 28.00 | 16.69 | 6.61 | 8.12 | 1700 | − 4.29 | 197 | 114 | − 23.43 | 395 | 0.9 |
| S4 | 29.00 | 15.25 | 6.46 | 8.01 | 1861 | − 5.27 | 315 | 93 | − 23.68 | 543 | 27.6 |
| T1 | 29.00 | 14.30 | 6.56 | 8.05 | 1757 | − 5.57 | 259 | 80 | − 23.62 | 490 | 18.1 |
| T2 | 29.00 | 15.51 | 6.74 | 8.07 | 1727 | − 4.79 | 182 | 106 | − 23.21 | 456 | 11.9 |
| T3 | 28.50 | 16.55 | 6.46 | 8.11 | 1683 | − 4.39 | 154 | 154 | − 22.97 | 403 | 2.4 |
| M1 | 28.00 | 15.14 | 6.99 | 8.07 | 1711 | − 5.93 | 282 | 264 | − 23.07 | 443 | 9.4 |
| M2 | 28.00 | 15.14 | 6.91 | 8.12 | 1735 | − 4.63 | 219 | 436 | − 23.15 | 376 | –2.6 |
| M3 | 28.00 | 15.23 | 7.46 | 8.13 | 1736 | − 5.30 | 222 | 287 | − 23.62 | 401 | 1.9 |
| M4 | 28.50 | 14.78 | 6.84 | 8.04 | 1920 | − 5.38 | 215 | 96 | − 23.82 | 503 | 20.3 |

Table - 3: Physicochemical parameters, inorganic and organic C related parameters, and $CO_2$ exchange flux across water-atmosphere interface at the Hooghly estuary. Here, $W_T$ = water temperature, DO = dissolved oxygen.


| Station | $W_T$ (°C) | Salinity | DO (mgL$^{-1}$) | pH | DIC (μM) | $\delta^{13}C_{DIC}$ (‰) | DOC (μM) | POC (μM) | $\delta^{13}C_{POC}$ (‰) | $pCO_2$ (μatm) | $FCO_2$ (μmol m$^{-2}$ hr$^{-1}$) |
|---|---|---|---|---|---|---|---|---|---|---|---|
| H1 | 32.0 | 0.04 | 6.29 | 7.92 | 2700 | − 6.98 | 244 | 313 | − 25.34 | 2036 | 285.2 |
| H2 | 33.0 | 0.07 | 6.11 | 7.71 | 1678 | − 8.38 | 304 | 177 | − 25.19 | 2316 | 343.8 |
| H3 | 31.0 | 0.08 | 6.45 | 7.83 | 2498 | − 6.70 | 235 | 286 | − 25.95 | 2490 | 355.4 |
| H4 | 31.0 | 0.13 | 5.24 | 7.73 | 2446 | − 7.38 | 243 | 254 | − 25.40 | 2691 | 389.2 |
| H5 | 31.0 | 0.19 | 5.38 | 7.77 | 2355 | − 7.56 | 340 | 130 | − 25.67 | 2123 | 293.1 |
| H6 | 30.5 | 0.32 | 5.66 | 7.31 | 2157 | − 8.61 | 308 | 116 | − 24.07 | 4678 | 717.5 |
| H7 | 31.5 | 5.83 | 6.71 | 7.68 | 1829 | − 6.79 | 662 | 145 | − 24.70 | 1184 | 132.0 |
| H8 | 31.0 | 5.19 | 7.14 | 7.31 | 2023 | − 6.78 | 354 | 139 | − 23.47 | 3153 | 455.8 |
| H9 | 31.5 | 9.08 | 6.62 | 7.90 | 1915 | − 6.08 | 332 | 161 | − 23.53 | 665 | 44.9 |
| H10 | 31.5 | 9.72 | 6.17 | 8.08 | 1787 | − 5.78 | 249 | 95 | − 24.06 | 452 | 10.1 |
| H11 | 31.0 | 8.43 | 6.37 | 8.07 | 1977 | − 7.21 | 358 | 95 | − 25.94 | 486 | 15.6 |
| H12 | 31.5 | 5.83 | 7.40 | 8.29 | 1871 | − 6.60 | 260 | 133 | − 26.28 | 274 | −19.3 |
| H13 | 31.0 | 10.37 | 7.00 | 8.24 | 1843 | − 5.57 | 394 | 129 | − 24.72 | 267 | −19.8 |










Table - 4:  The DIC concentrations and $\delta^{13}C_{DIC}$ of groundwater (GW) and pore-water (PW)
samples collected around the Hooghly-Sundarbans system.

| Ecosystems | Station | DIC (μM) | $\delta^{13}C_{DIC}$ (‰) |
|---|---|---|---|
| **Hooghly** | H3GW | 11756 | − 12.66 |
| | H4GW | 6230 | − 7.85 |
| | H5GW | 6327 | − 8.96 |
| | H6GW | 7026 | − 11.27 |
| | H7GW | 5655 | − 6.91 |
| | H11GW | 9115 | − 7.67 |
| | H12GW | 6858 | − 7.49 |
| | H13GW | 7258 | − 7.21 |
| | Gangasagar GW | 7246 | − 6.67 |
| **Sundarbans** | Lothian GW | 7524 | − 6.84 |
| | Lothian PW | 13425 | − 18.05 |
| | Kalash GW | 13599 | − 6.69 |
| | Virat Bazar GW | 8300 | − 10.56 |















 **Figure Captions:**

 **Fig. 1**: Sampling locations at the (a) estuaries of Sundarbans, and (b) Hooghly estuary.

 **Fig. 2**: % saturation of DO - salinity relationship in the Hooghly-Sundarbans system.

 **Fig. 3**: (a) DIC - salinity in the Hooghly, (b) $\delta^{13}C_{DIC}$ - salinity in the Hooghly, (c) $\Delta$DIC - $\Delta$
 $\delta^{13}C_{DIC}$ in the Hooghly, (d) DIC - salinity in the Sundarbans, and (e) $\delta^{13}C_{DIC}$ - salinity in the
 Sundarbans.

 **Fig. 4**: (a) DOC - salinity in the Hooghly-Sundarbans system, (b) DOC - $p$CO$_2$ in the Hooghly,
 (c) DOC - $p$CO$_2$ in the Sundarbans, and (d) DOC - POC in the Hooghly-Sundarbans system.

 **Fig. 5**: (a) SPM - salinity in the Hooghly-Sundarbans system, (b) POC - salinity in the Hooghly-
 Sundarbans system, (c) %POC/SPM - salinity in the Hooghly-Sundarbans system, (d) $\Delta$POC -
 $\Delta$ $\delta^{13}C_{POC}$ in the Hooghly, (e) POC - $p$CO$_2$ in the Hooghly, and (f) POC - $p$CO$_2$ in the
 Sundarbans.

 **Fig. 6:** (a) ECO$_2$ - AOU in the Sundarbans, (b) $p$CO$_2$ - salinity in the Sundarbans, (c) ECO$_2$ -
 AOU in the Hooghly, (d) $p$CO$_2$ - salinity in the Hooghly, (e) FCO$_2$ - salinity in the Hooghly,
 and (f) FCO$_2$ - salinity in the Sundarbans.


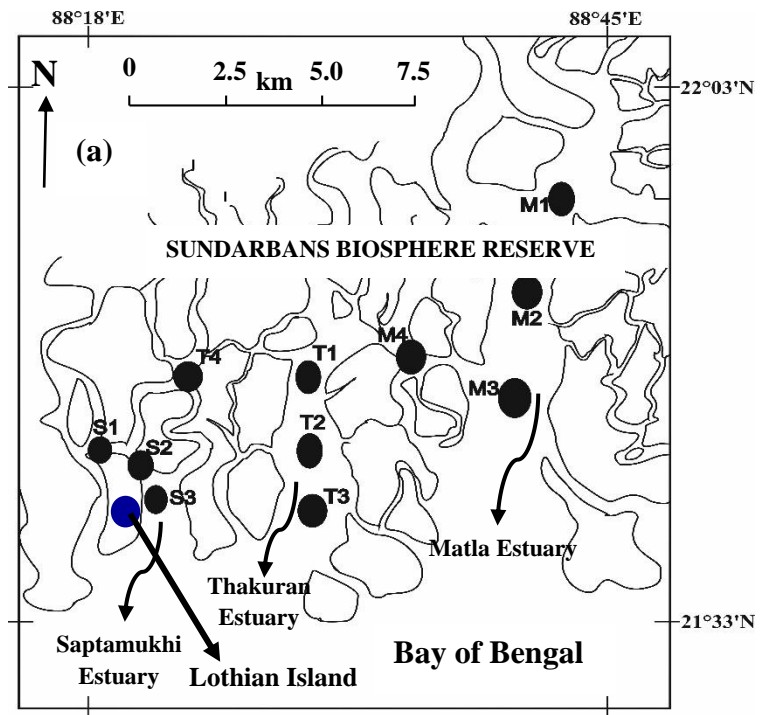


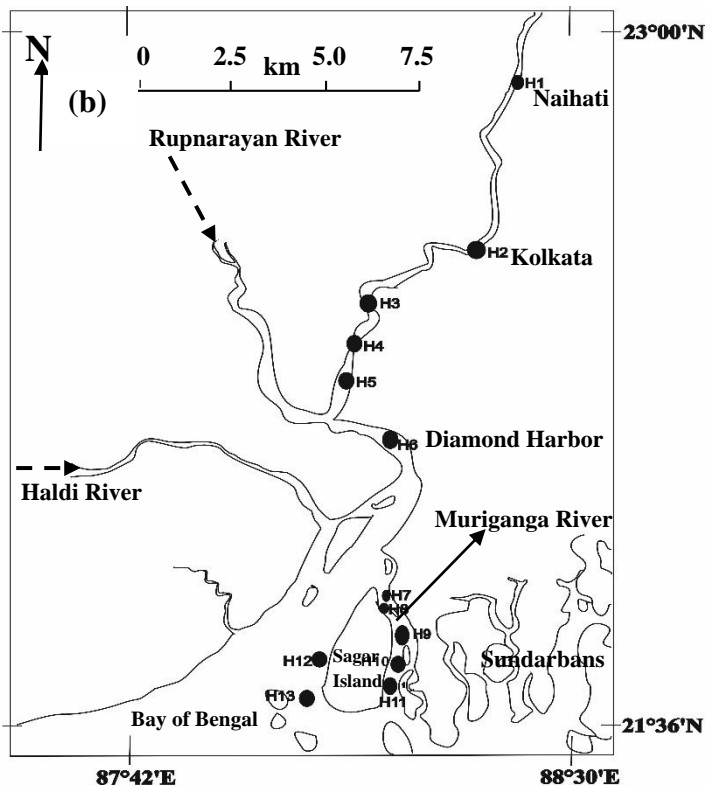


**Fig. 1**







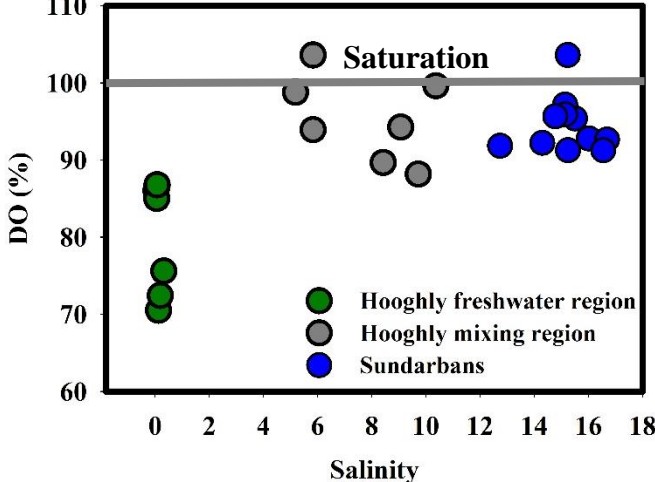

**Fig. 2**

















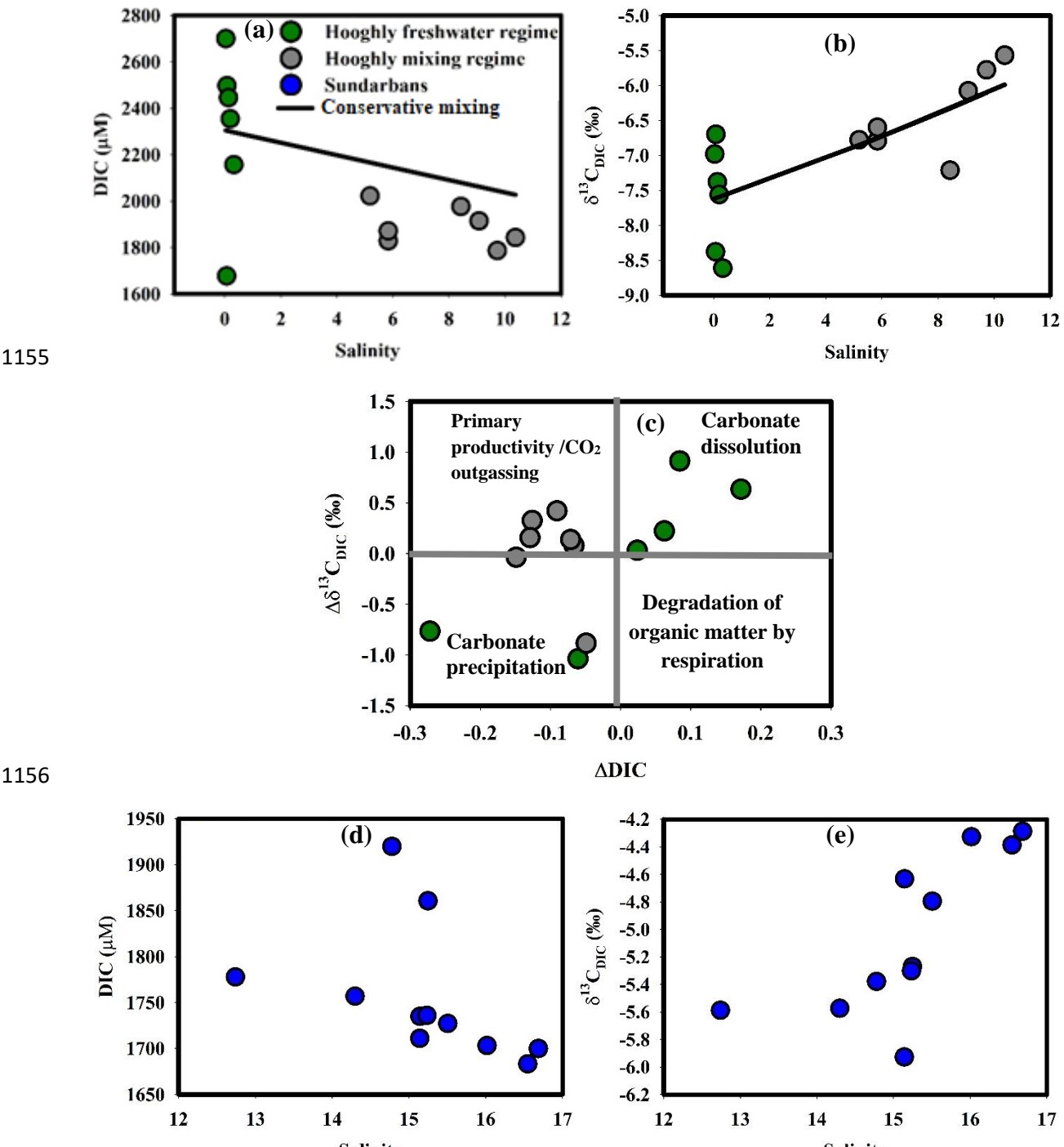





Fig. 3

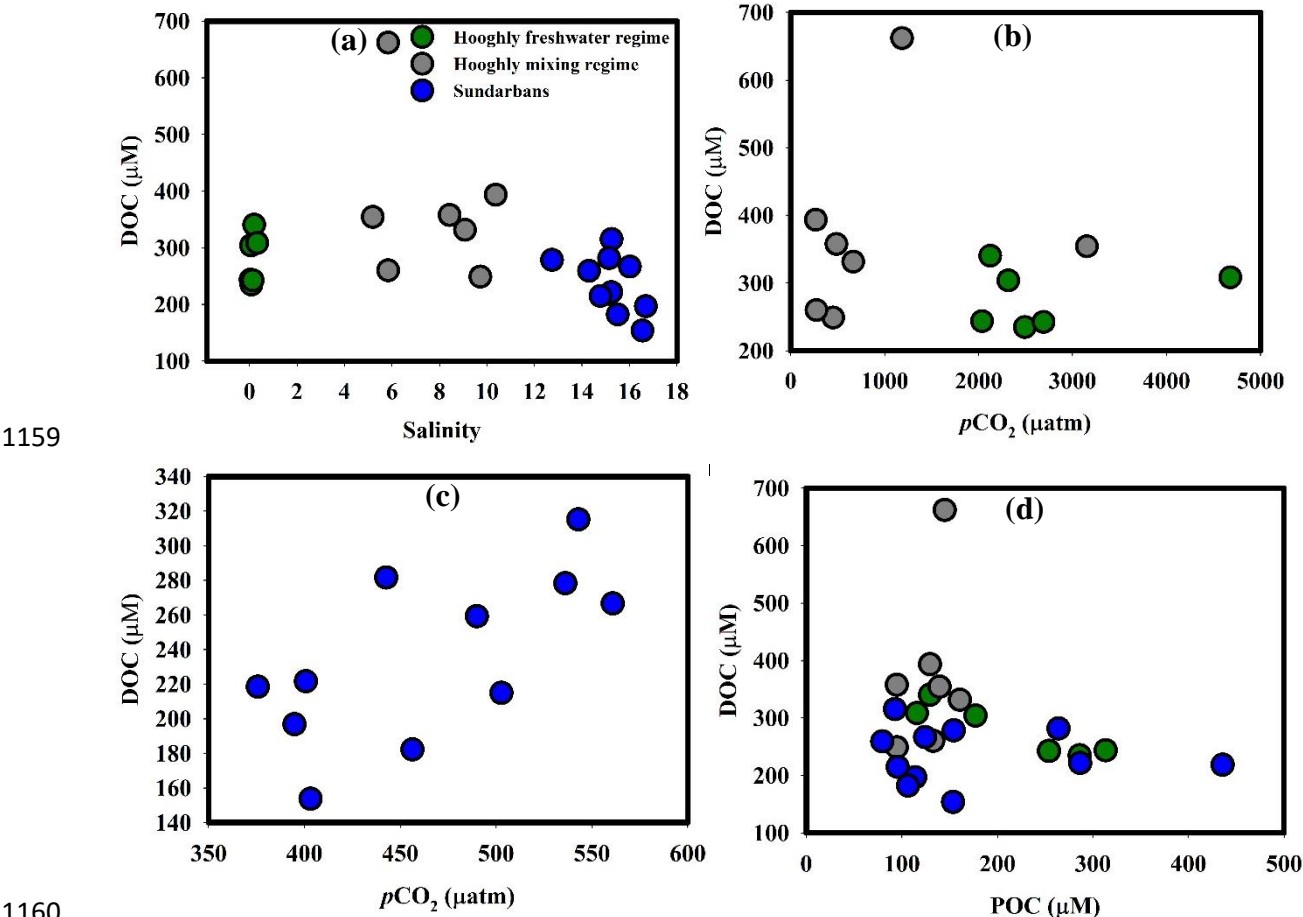



**Fig. 4**










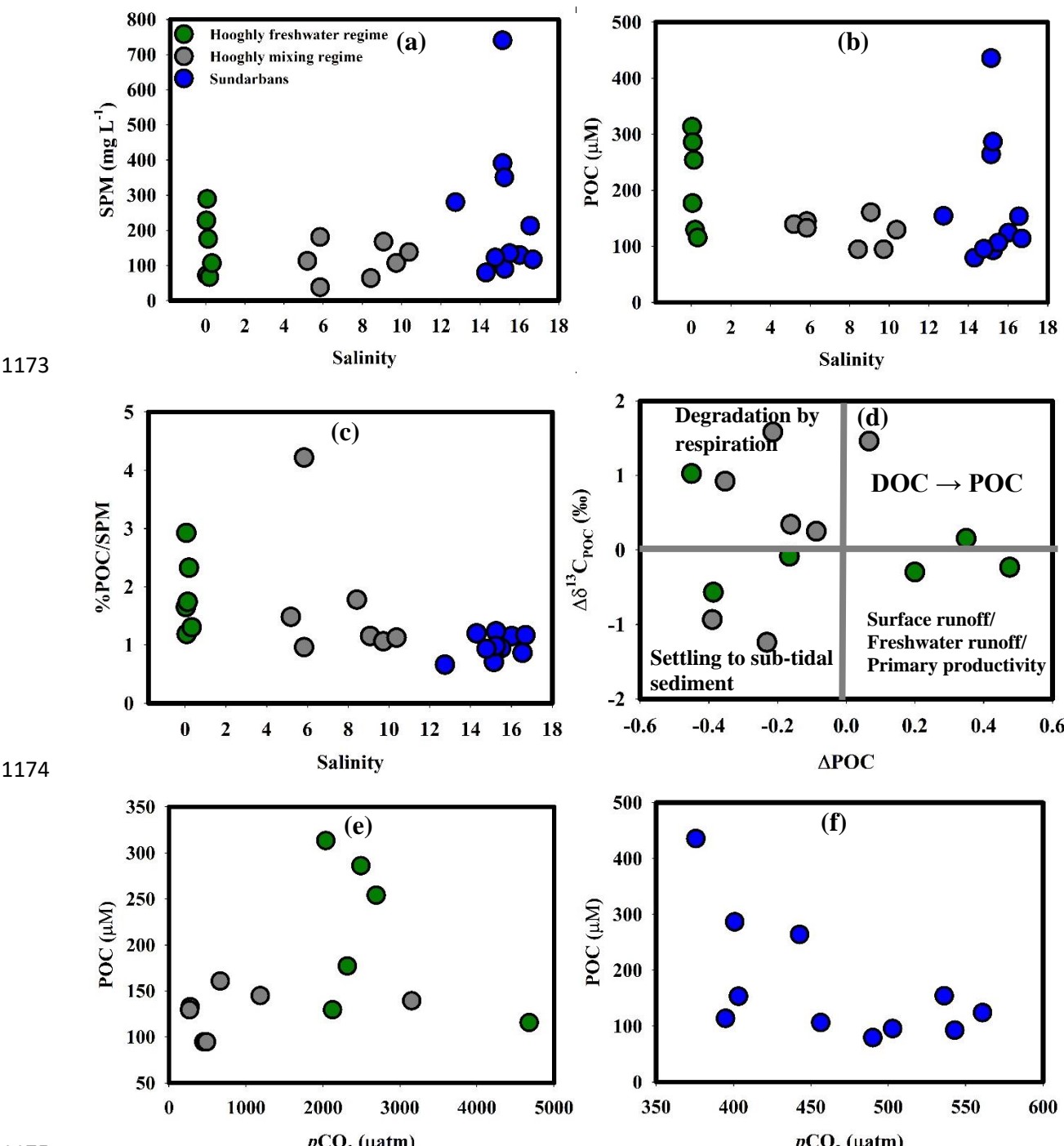



**Fig. 5**









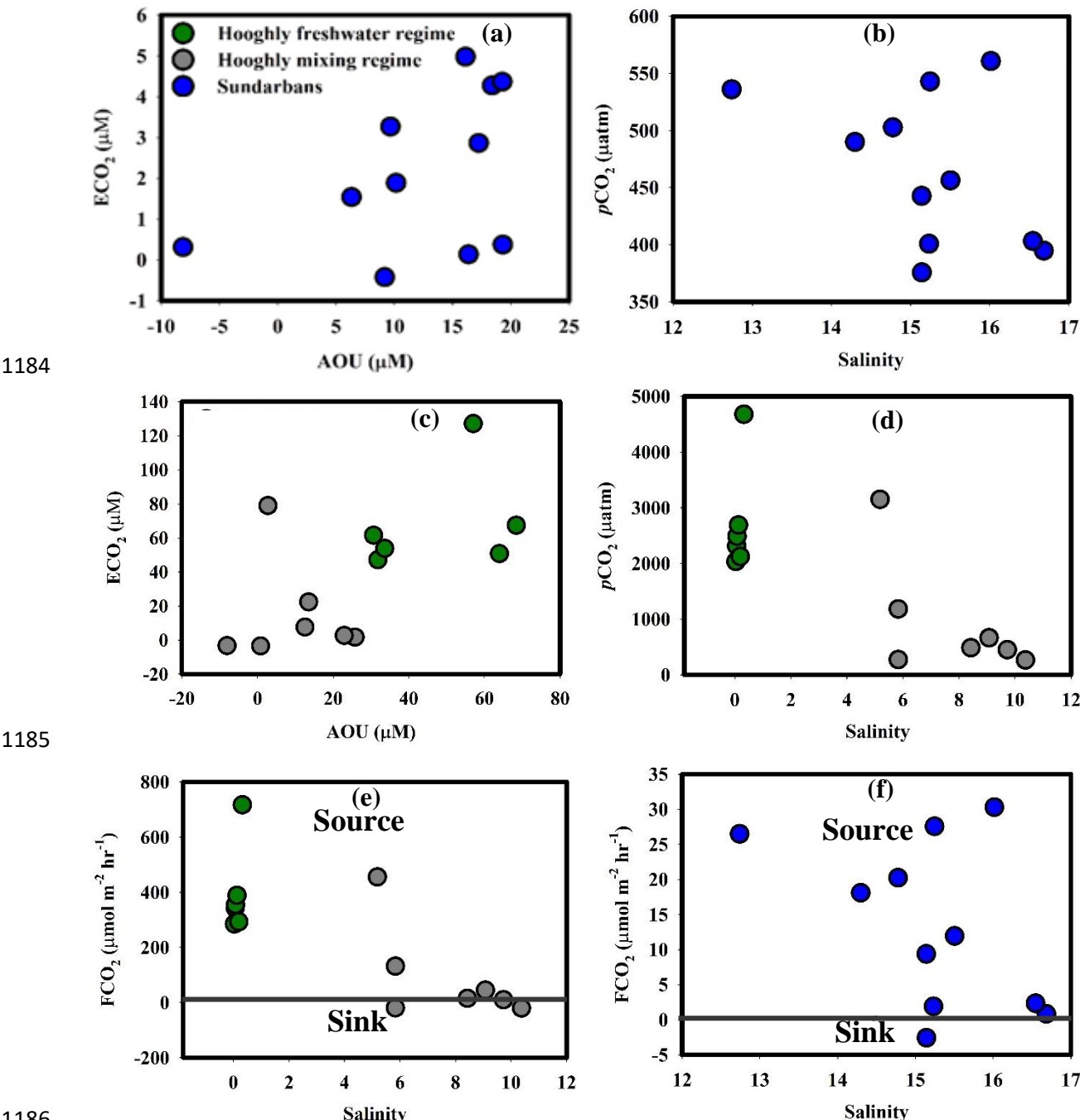



**Fig. 6**


