# Peer review of "The postmonsoon carbon biogeochemistry of the Hooghly-system under different levels Sundarbans estuarine of anthropogenic impacts Manab Kumar Dutta1, Sanjeev Kumar1\*, Rupa Mukherjee1, Prasun Sanyal2, Sandip Kumar Mukhopadhyay2 1G"

_Biogeosciences, 2018_

## Referee Comment (RC1) · Anonymous Referee #1 · 23 Jul 2018

Review of Dutta et al., The authors made measurements of organic and inorganic carbon parameters, along with isotopes and other ancillary measurements in an attempt to determine the sources and distribution of DIC, DOC, and POC in an estuary in the Hooghly-Sundarbans system (shortly written as C biogeochemistry by the authors). Although the ms falls within the scopes of BG and covers a good data range from various sites of the estuarine system but finally it ends up in a disappointment because of poor writing and hesitations of choosing a concrete aim. Unfortunately, the manuscript reads like a data dump, with incomplete descriptions of the methods, presentation of the data, and some speculation about processes but with major processes left out; nothing seems conclusive. The manuscript is still in quite a rough stage, as detailed

with a non-exhaustive list of examples below, and does not seem ready for publication. Specific comments: The problem lies within the title. It seems the authors are in serous dilemma to show the data what actual basis: on C dynamics in polluted vs non-polluted system or only focus on mangroves and compare with sidechain Hooghly in a specific season or discuss on DIC mainly and less focus on DOC and POC or avoid already published articles on the same systems on same parameters on same season ! (e.g. Samanta 2015, Ray 2018, 2015) Unfortunately nothing was clear due to poor writing and unclear intention.

Other major comments I would suggest authors to give details of the sampling stations e.g. how or what type of anthropogenic input is there in the Hooghly? From where it is more coming from (upstream?). Its better to segment the study sites of Hooghly as upper/mid/lower stretch and Sundarbans as west/central and east. I anticipate the upper and mid stretches are human or industrial impacted compared to lower, so one of ideas in designing the story would be to explain variations of results within Hooghly first between e.g. H1-6 and H6-11 and then compare with S,T,M series. That would read the paper interesting otherwise its just mimicking the findings already shown by Samanta 2015, Ray 2018. Authors argued on C- data limitation of previous reports but it is found that Samanata'15 covered even much higher sites from Hooghly than the present report (c.a 35 vs 13 surface water and 8 vs 8 ground water) and Ray '18 was also not far (>10 in S series vs 10 S,T,M). So this argument on data imitation does not hold true ! Result section is only meant for results and it should be avoded to define data set and add citations in Results that fully present in the paper. It is proposed to move those parts of the Result section to discussion (LN 229-234, 248-49, 257-59, 267-71) This is over-speculative to argue on contributions of pore water on the overlying DIC concentrations based on only one measurement (Tab 3, Lothian PW). LN342-345: This is unclear why $\Delta DICM2$ is shown as micromole instead of permil. Authors should better calculate the amount of DOC and POC added or subtracted from the system applying conservative mixing (same way they did for DIC) and explain in-depth details of their mixing pattern (same applies to DIC). LN349 Are the ground and pore

water discharge not being considered as 'biogeochemical' process? Section 4.3. This part is weakly written and over-speculative without supporting any evidence .e.g. the argument of DOC photooxidation or conversion of DOC to POC as removal process. While it requires suitable ambient condition for DOC photooxidation such as high water residence time, stable environmental condition (not expected in mangroves), the same applies to adsorption/desorption of DOC-POC. Part of that exchange is mediated by charged complexes, repulsion - attraction interactions, and therefore subject to salinity effects. So, when river water rich in DOC first mixes with saline water, at least a portion of DOC is lost from solution (removed) and incorporated into POC (Fe-oxide colloids usually are extracted at the same time). Once the salinity exceeds 2 - 3, however, the effect of salinity on coagulation behavior is largely complete. Another point is no detailed explanation on distribution pattern with salinity was given, authors should highlight the reasons of the mild upward gradient along Hooghly and steep downward trend along the Sundarban. Section 4.4 LN410 only freshwater runoff, no surface run off that adds POC too in upstream? LN440-446 this part is totally redundant as there was not an iota of signal of CH4 from the observed d13 POC (13CH4 is $\sim$ 55-60 permil) Does the author have Chl-a or nutrient data (even from literature) to support higher marine input in POC in Sundarban and 13C values of mangrove leaf, and soil from Hooghly to denote higher terrigenous contribution to the POC pool? Authors are suggested to read carefully the works of Samanta'15 and Ray'18 and use their values to support some of the arguments.

points of concerns

terminology > I counted 'biogeochemistry' was used over 25 times in the 16 pages ms ! too much. Additionally, this is not clear to me what does it actually mean by C biogeochemistry? Is it C-components distributions in different phases (solid suspended and dissolved) under varying biogeochemical processes? If so please specify at least once > d13C values are not 'depleted' or 'enriched' (LN256, 428..). When referring to d13C values, they can be described as higher or lower when comparing different

samples, or one could describe differences as e.g. a certain C pool is enriched or depleted in 13C versus another C pool or sample. > r2 not R2 Inconsistent use of [POC] in the discussion, if the bracket is used for POC then it should also appear for DIC and DOC

unit Random use of units : DOC in mg/L, DIC in mM, POC in uM. These should be harmonized. Use DOC in uM for better compare with other studies

Sampling Define sampling strategy neatly, Its written postmosoon was chosen due to high litterfall, but there is no account of litter source identified for DOC or POC or any impact positive or negative on estuarine C biogeochemistry authors assumed. That is to be addressed in the discussion. Mention the H, S, T, M series in the text Mention general tidal nature while sampling (height, HT/LT, depth)

methods specify> pore size of filters used for DOC, SPM relative uncertainty in POC methods; technique of pore water collection; ground water (from tube pump?)

Figs Again weak representation: font sizes of x, y axis digits (and titles) to be increased much (too much stress to eyes now !); use box to cover legends, its confusing with data points and legends, remove break in y axis in Fug 3e and 4a), black star coding was used both for sundarban and observed d13DIC and grey round coding was used for Hooghly and observed DIC, these symbols must be changed to give separate identity of them in all figs <overall IMPROVE CLARITY of ALL FIGURES>

Data use a consistent number of decimals (1) to report d13C data, and Salinity considering the analytical error on the measurements.

Minor comments First sentence of abstract is redundant LN65 Use current reference for the riverine export flux (works of Pete Raymond, Huang) Many references are out of place e.g. the comparison of present data with Khura (LN 231, 249 Miyajima paper) was unlikely as two environments are totally different even if compared authors should mention conservative data like S in Khura estuary for better comparison. LN234: Provide values of Samanta et al 2015

Finally, I think it is necessary to stand back and consider how to best weave the entire story together in the discussion more efficiently and succinctly

---

## Referee Comment (RC2) · Anonymous Referee #2 · 27 Jul 2018

Review of Dutta et al "The postmonsoon carbon biogeochemistry of estuaries under different levels of anthropogenic impact". Submitted to Biogeosciences. This study presents data from a single cruise in 2 Indian estuaries to try and decipher differences in carbon cycling between a 2 Indian estuaries with differing levels of anthropogenic influence. After reading and rereading this paper several times, it is unclear what the purpose of this study is. There is no defined hypothesis to be tested, and while the title suggests there will be some kind of comparative analysis to look at anthropogenic impact on carbon cycling (an interesting and important topic), I am left a little under-whelmed with the analysis undertaken. The entire manuscript is based single sampling campaigns, which while not ideal is not the main issue. The main area of concern is the

lack of any direction in the paper, and the somewhat descriptive and qualitative nature. I suggest that the authors define their hypothesis more clearly, and use the data to test this hypothesis.

I have a series of comments below, some minor some major that may help.

Abstract I am not convinced that the data as presented can be used to draw such strong conclusions as to the drivers of carbon dynamics in the studied estuaries. For example Ln 35-38 The evidence supporting these processes is weak at best – no measurements of production, carbonate dissolution nor porewater exchange were measured, and the spatial trends in concentrations and isotopes (and relationships between carbon variables and DO etc.) were not strong enough to draw any distinct conclusions on the importance of these mechanisms.

Same goes for lines 45-47.

Line 49 – 52 I am unconvinced that the observed trends are shown to be directly linked to anthropogenic influence. Yes the estuaries appear to differ, but what else might be driving this. For example looking at salinity and pCO2 in the 2 different estuaries – the highest salinity in the "anthropogenically" impacted estuary is lower than the lowest salinity in the "undisturbed" estuaries. Could the observed differences simply be related to freshwater input? What are the nutrient concentrations in the 2 estuaries? How different are they in hydrodynamics (looks like the geomorphology is distinctly different between the 2 estuary types from Fig 1). These are just a few of the alternative reasons to look at for explaining the differences observed

Introduction

Ln 59 – 60 What is meant by "record biogeochemical and hydrological processes"?

Ln 67 – Richey is not correct ref for this statement (Richey paper is on Amazon)

Ln 68 – 70 – Still large uncertainties on estuarine $CO_2$ flux – look at error bars on Cai 2011 estimate

Ln 76 What is meant by "biogeochemical characteristic"?

Ln 78 – 79 Not always – see Cotovicz Jr, L. C., Knoppers, B. A., Brandini, N., Costa Santos, S. J., & Abril, G. (2015). A strong $CO_2$ sink enhanced by eutrophication in a tropical coastal embayment (guanabara bay, rio de janeiro, brazil). Biogeosciences, 12(20), 6125-6146. doi:10.5194/bg-12-6125-2015

Among others

Ln 81 – 84 There has been a lot of work on mangrove carbon cycling work done since Dittmar and Larra's work in the early 2000's. Might be worth looking at more recent papers to see how far our understanding has come since then.

Ln 104- 106 Give some quantitative data to support your "anthropogenically influenced" argument. What are nutrient concentrations like? Population density? Land use? Freshwater inflow? Etc etc. A table compiling this data would give the reader an instant understanding of the differences.

Line 117 What is meant by positive and negative feedback here? These terms are not really applicable to biogeochemistry as a whole, but may be related to specific mechanisms/cycles.

Ln 137-140 Clearly there is freshwater input – the salinities are very low. In fact my thoughts are that these freshwater inputs are a main driver of the observed differences.

Ln 159 Assume the filters were GFF filters – add these details.

Ln 161 Accuracy of TAlk measurements. Were CRMs measured (hope so!). Also add accuracy/precision etc of all other parameters.

Ln 196 – 198 What were the input parameters for measuring pCO2? What disassociation constants were used etc?

Ln 205 – 208 Why use L&M relationship? Need some kind of justification here other than saying it is conservative.

Results

Do not compare and contrast your data with previous studies in the results. Just report your data.

Discussion

Ln 289 – 293 What are the implications for these findings? Need to dig deeper or remove.

Ln 306-311 (and Fig 3b) How was the conservative d13C-DIC mixing line calculated? Looks like you have simply added a linear relationship between the 2 endmembers, the relationship is generally not linear (See Fry, B. (2002). Conservative mixing of stable isotopes across estuarine salinity gradients: A conceptual framework for monitoring watershed influences on downstream fisheries production. Estuaries, 25(2), 264-271. Also as you do not have any mineralogy of carbonates – I would avoid using the term "calcite" precipitation, change to "carbonate" precipitation

Ln 323-325 – What does DO tell you about primary production? Looks like DO is generally undersaturated?

Ln 335-338 Describe all the terms in this equation in the following text

Ln 359 Where do the TAlk/DIC numbers come from? The stoichiometric relationship should be based on the slope of the line over the whole estuary, rather than individual data points – therefore not sure how you have a range here.

Ln 364 – 368 Give details on this calculation. Just using the discharge rate and pore-water DIC concentration I get a different value.

Ln 383-390 – Not sure that looking at pCO2 VS DOC gives any indication as to the importance of porewater exchange! Could also simply be freshwater input from upstream, surface water runoff, or simply leaching/respiration.

Ln 412 Give details about the "jute" industry.

Ln 424-426 The POC isotopes could simply be related to the relative amount of fresh-water inputs in each system (this can also be applied to most of the other differences observed)

Ln 431-446 I am unsure why anaerobic respiration (which is energetically les favourable than aerobic respiration) would be more important in a well oxygenated estuary. The authors should expand this to explain things more clearly or remove.

Ln 447-451 What is the importance/implications of this – expand or remove.

Ln 455 – 460 These sections seem to contradict each other. Initially it is stated man-grove inputs are insignificant – then porewater exchange of mangrove derived $CO_2$ is highlighted as important?

Ln 463 – Cai ref – there are plenty of mangrove references for this process, might be more appropriate to use some of those here

Ln 463 – 466 How about plotting $ECO_2$ vs AOU (in molar units). Look at the slope of the line. This will give a better indication of the importance or aerobic vs anaerobic R.

Ln 470 – 473 How was gas exchange and the differences between $CO_2$ and $O_2$ coupled into this calculation? Also how does this value compare to your air-water $CO_2$ fluxes (you will need to normalize your volumetric rates to surface area for comparison)

Ln 480 – I think your global value for mangrove systems (63 umol/m2/d) should be 63 mmol/m2/d – which is much higher than the fluxes measured in this study.

Conclusions:

Point 1 – this variability is likely simply linked to the variability in salinity (and therefore freshwater inputs) between the studied estuaries.

Point 2 – Unconvinced that primary production has been shown to be the main control-ling factor on DIC. Without any measurements of PP or some more thorough analysis of other potential mechanisms, this statement is far too strong

Point 3 – I see no strong conclusive evidence of either of these points. Again statement is too strong without measurements of DOC flocculation or porewater exchange of DOC

Point 4 Assume this is based on isotopes? Again this could simply be related to the marked differences in freshwater content within each of the estuaries.

---

## Short Comment (SC1) · 13 Aug 2018

In the present study, large spatial extent has been covered which includes Hooghly River and other rivers of Indian part of Sundarban. My comments regarding the present study are as follows: 1. From the sampling strategy (line no. 150 to 153), it is apparent that only one-time discrete sampling has been done in all the sites in duplicate, whereas from the third objective of the study it is clear that the authors had the aim to quantify and characterise the air-water $CO_2$ flux for the post-monsoon season. The authors concluded "During post monsoon, the entire Hooghly-Sundarbans system acted as a source of $CO_2$ to the regional atmosphere." How can it be concluded (even qualitatively) from such discrete data without performing at least one complete diurnal sampling at each site within post-monsoon season, while four months (October, November, December and January) are generally considered as post monsoon season in this region? 2. The study area and sampling locations are quite similar with the recent work of Akhand et al. (2016). Moreover, the third objective and one of the conclusions of the present study is also very similar to the Akhand et al. (2016). For example, the authors stated, "The entire Hooghly-Sundarbans system acted as source of CO2 to the regional atmosphere with ∼17 times higher emission from the Hooghly compared to Sundarbans", whereas one of the key findings of Akhand et al. (2016) is "River-dominated Hugli Estuary emits 14 times more CO2 than the marine-dominated Matla Estuary". Surprisingly, despite of such degree of similarity between two studies, there is no comparison of data with Akhand et al. (2016) and not even mentioning of Akhand et al. (2016) in the present work. 3. Reviewer 2 already mentioned that line no. 455 to 460 are self-contradictory. I want to add that I agree with the authors statement that in the estuarine water of Sundarban, an important source of CO2 is mangrove sediment pore-water exchange during tidal pumping. This fact is also well established from the diurnal dataset of Akhand et al. (2013) and Akhand et al. (2016) in Sundarban. But, it is not clear to me, how this phenomenon can prove the exogenous origin of CO2? Moreover, except Hooghly and its distributary Muriganga, all other rivers (Saptamukhi, Thakuran, Matla, Gosaba and Bidya) in the Indian part of Sundarban have lost their original connections with the Ganga because of siltation and their estuarine character is now maintained by the monsoonal runoff only (Cole and Vaidyaraman, 1966). So, the central part of Sundarban (which comprises a major part of Indian Sundarban) experiences lack of freshwater (Chakrabarti1998; Mitra et al. 2009). Hence, the source of the exogenous nature of CO2 input in the Indian part of Sundarban needs more clarifications. 4. In line no. 479 to 481 authors stated "FCO2 measured for the estuaries of Sundarbans was markedly higher than global mean FCO2 (∼63 $\mu$ mol m-2 d-1) observed in mangrove creek and other similar estuaries (Call et al., 2015)". Reviewer 2 already correctly identified that the value should be ∼63 m mol m-2d-1. It might be

a typo by the authors, but it may convey wrong message to the global audience about Sundarban's mangrove surrounding water. Because, one of the key findings of Akhand et al. (2016) is that the fCO2 (water) value of the Matla, a mangrove dominated estuary of Sundarban, is at the lower end of the reported data from other mangrove ecosystems of the world. Biswas et al. (2004) also found that the Sundarban's mangrove dominated water is acting as a sink for atmospheric CO2 for all the four post monsoon months, while sampling in the three river-mouths. Also see Rosentreter et al. (2018), where they estimated world average flux of ∼57.5 mmol m-2 d-1 of CO2 from the mangrove surrounding water, and also commented that the CO2 efflux from the estuarine water of Sundarban is much lower side than the world average even sinks for atmospheric CO2 in some cases.

References: Akhand, A., Chanda, A., Dutta, S., Manna, S., Sanyal, P., Hazra, S., Rao, K.H. and Dadhwal, V.K., 2013. Dual character of Sundarban estuary as a source and sink of CO2 during summer: an investigation of spatial dynamics. Environmental Monitoring and Assessment, 185(8), pp.6505-6515. Akhand, A., Chanda, A., Manna, S., Das, S., Hazra, S., Roy, R., Choudhury, S.B., Rao, K.H., Dadhwal, V.K., Chakraborty, K. and Mostofa, K.M.G., 2016. A comparison of CO2 dynamics and air‐water fluxes in a river‐dominated estuary and a mangrove‐dominated marine estuary. Geophysical Research Letters, 43(22). Biswas, H., Mukhopadhyay, S.K., De, T.K., Sen, S. and Jana, T.K., 2004. Biogenic controls on the air—water carbon dioxide exchange in the Sundarban mangrove environment, northeast coast of Bay of Bengal, India. Limnology and Oceanography, 49(1), pp.95-101. Chakrabarti, P.S., 1998. Changing courses of Ganga, Ganga–Padma river system, West Bengal, India–RS data usage in user orientation, river behavior and control. Journal of River Research Institute, 25, pp.19-40. Cole, C. V., and P. P. Vaidyaraman. "Salinity distribution and effect of freshwater flows in the Hooghly River." In Proceedings of Tenth Conference on Coastal Engineering, Tokyo, Japan, September, pp. 1312-1434. 1966. Mitra, A., Banerjee, K., Sengupta, K. and Gangopadhyay, A., 2009. Pulse of Climate Change in Indian Suindarbans: A Myth or Reality?. National Academy Science Letters (India), 32(1), p.19.

Rosentreter, J.A., Maher, D.T., Erler, D.V., Murray, R. and Eyre, B.D., 2018. Seasonal and temporal CO2 dynamics in three tropical mangrove creeks–A revision of global mangrove CO2 emissions. Geochimica et Cosmochimica Acta, 222, pp.729-745.
* * *

---

## Author Comment (AC1) · 10 Sep 2018

Comment: Review of Dutta et al., The authors made measurements of organic and inorganic carbon parameters, along with isotopes and other ancillary measurements in an attempt to determine the sources and distribution of DIC, DOC, and POC in an estuary in the Hooghly-Sundarbans system (shortly written as C biogeochemistry by the authors). Although the ms falls within the scopes of BG and covers a good data range from various sites of the estuarine system but finally it ends up in a disappointment because of poor writing and hesitations of choosing a concrete aim. Unfortunately, the manuscript reads like a data dump, with incomplete descriptions of the methods, pre-

sentation of the data, and some speculation about processes but with major processes left out; nothing seems conclusive. The manuscript is still in quite a rough stage, as detailed with a non-exhaustive list of examples below, and does not seem ready for publication.

Response: We are thankful to the reviewer for his constructive criticism and comments. We are open to include his suggestions in the revised manuscript at suitable places and come back with a better version of the manuscript.

Specific comments:

Comment: The problem lies within the title. It seems the authors are in serous dilemma to show the data what actual basis: on C dynamics in polluted vs non-polluted system or only focus on mangroves and compare with sidechain Hooghly in a specific season or discuss on DIC mainly and less focus on DOC and POC or avoid already published articles on the same systems on same parameters on same season! (e.g. Samanta 2015, Ray 2018, 2015) Unfortunately nothing was clear due to poor writing and unclear intention.

Response: The main objective of the present study is to bring out contrast in different components of the carbon cycle of anthropogenically affected Hooghly estuary and mangrove-dominated estuaries of Sundarbans during postmonsoon. We have tried to focus on each component depending on the variabilities and scope of our data. We would respectfully disagree with the reviewer that we have tried to avoid the earlier works by Samanta et al., (2015) and Ray et al. (2015, 2018). We have cited their works and used the findings of these authors in our manuscript to interpret our data. We would also like to submit that whereas Samanta et al. (2015) is a nice study with comprehensive focus on only DIC in the Hooghly estuary; Ray et al. (2015, 2018) covers more number of parameters with limited spatial coverage (please see Figure 1 attached). In a vast mangrove ecosystem as Sundarbans, Ray et al. (2015, 2018) have covered just one location during both studies. We have tried our best to cover
the Hooghly-Sundarbans system on wider scale with multiple parameters to comprehensively study C dynamics in this system. So far as writing is concerned, we believe that there is always a scope for improvement and we will be happy to incorporate the suggestions by the reviewer.

Other major comments

Comment: I would suggest authors to give details of the sampling stations e.g. how or what type of anthropogenic input is there in the Hooghly? From where it is more coming from (upstream?).

Response: We would include sentences to that effect in the revised version. For example, surface runoff at freshwater region, like waste water discharge from the City of Kolkata (St. H2) and jute industry (located in between Stns. H1 to H3) is a major source of anthropogenic inputs to the Hooghly. We would also include previously published nutrients concentration as an evidence for higher anthropogenic input in the Hooghly.

Comment: Its better to segment the study sites of Hooghly as upper/mid/lower stretch and Sundarbans as west/central and east. I anticipate the upper and mid stretches are human or industrial impacted compared to lower, so one of ideas in designing the story would be to explain variations of results within Hooghly first between e.g. H1-6 and H6-11 and then compare with S,T,M series. That would read the paper interesting otherwise its just mimicking the findings already shown by Samanta 2015, Ray 2018.

Response: During this postmonsoonal study, based on the present salinity range and gradient, it is difficult to divide the Hooghly estuary into upper/mid/lower stretch like other estuaries with sharp salinity gradient from fresh to marine zone. Although reviewer has suggested human and industrial influence along with lower estuarine region as a basis for such demarcation, we believe that such demarcation would be qualitative as no quantitative information are available to us to support such demarcation. Therefore, we are inclined to divide Hooghly estuary as freshwater zone (H1-H6) and mixing

zone (H6 – H11) based on our salinity data. For the Sundarbans, the spatial extent is not wide enough (less than 100 km2) to divide them into west, central and east zones. If we apply this criterion, we would be left with 3 data points from each region (upper/middle/lower), which is not enough for data analysis and further interpretation to understand the characteristics of individual estuaries (S, T and M). Therefore, in the revised manuscript, we wish to discuss first the freshwater and mixing region in the Hooghly estuary and then compare it with the Sundarbans. We hope that the reviewer will agree to our suggestion.

Comment: Authors argued on C- data limitation of previous reports but it is found that Samanata'15 covered even much higher sites from Hooghly than the present report (c.a 35 vs 13 surface water and 8 vs 8 ground water) and Ray '18 was also not far (>10 in S series vs 10 S,T,M). So this argument on data imitation does not hold true!

Response: We agree with the reviewer that DIC is extensively discussed by Samanta et al. (2015) for the Hooghly estuary with much better spatial and seasonal coverage compared to our study. The author have also reported $\delta$13CPOC at some locations (n = 26). DIC and pCO2 for the Hooghly and Matla estuaries have also been reported by Akhand et al. (2016). The first report for the Hooghly-Sundarbans system with different components of C cycle with their isotopic compositions were reported by Ray et al. (2015). However, this study is limited by spatial coverage (3 stations from Hooghly and one from Sundarbans). Unless reviewer is referring to paper other than Ray et al. (2018) published in The Science of Total Environment, his argument about Ray et al. (2018) having large sampling locations (>10 in S series vs 10 S,T,M as pointed out by the reviewer) appears to be not correct. The map of the sampling location of Ray et al. (2018) is shown in Fig.1. In the light of the above, we would like to argue that the present study has much larger spatial coverage with multiple parameters and is better equipped to decipher the differences in C biogeochemistry of the contrasting systems such as Hooghly and the estuaries of Sundarbans.

Comment: Result section is only meant for results and it should be avoided to define

data set and add citations in Results that fully present in the paper. It is proposed to move those parts of the Result section to discussion (LN 229-234, 248-49, 257-59, 267-71).

Response: Thanks for the suggestion. Agreed.

Comment: This is over-speculative to argue on contributions of pore water on the overlying DIC concentrations based on only one measurement (Tab 3, Lothian PW).

Response: We agree with the reviewer that it is not enough to quantify contribution of pore water on adjoining estuarine water DIC pool based on a single measurement in this large mangrove ecosystem (Sundarbans). We are sure reviewer will appreciate that it is a logistics challenge to perform sampling in the Sundarbans. To perform sampling, permission is needed from the forest department. Also, very few islands are open for scientific investigations and some of them are tiger infested. During the present sampling, we had planned to cover at least all littoral zones of the Lothian Island. However, we were not permitted by the forest security service as conditions were not conducive to carry out investigation at mid and upper littoral zones. Therefore, we had to restrict our measurement in lower littoral zone only. Our advective DIC flux across mangrove sediment-estuary interface can be considered as first-time baseline value. The same caveat we have put in the manuscript as well.

Comment: LN342- 345: This is unclear why ïĄĎDICM2 is shown as micromole instead of permil.

Response: As you can see from the formula, the units of numerator and denominator is ($\mu$M x ‰ and ‰ respectively. The ‰ gets cancelled keeping $\Delta$DICM2 unit as '$\mu$M'.

Comment: Authors should better calculate the amount of DOC and POC added or subtracted from the system applying conservative mixing (same way they did for DIC) and explain in-depth details of their mixing pattern (same applies to DIC).

Response: Thanks to the reviewer for this suggestion. Using similarly calculated end

members values or taken from the same references as DIC, added or removed POC and DOC in the Hooghly were calculated for the revised manuscript. For Sundarbans, mangrove derived POC and DOC addition/removal was calculated using the same expressions as DIC. Additionally, very similar to DIC, a mixing plot between $\Delta$POC and $\Delta\delta$13CPOC was plotted to explore influencing processes. However, for DOC it was not possible to perform this analysis due to unavailability of $\delta$13CDOC data during the present study. We have used interrelationships between various parameters to justify removal or addition. We will include the above information in the revised manuscript.

Based on the above, additions to the POC section may be as following: Estimated $\Delta$POC in the Hooghly indicated both net addition (n = 3) and removal (n = 3) of POC in the freshwater region ($\Delta$POC = − 95 to 101$\mu$M). The removal (n = 6) dominated over addition (n = 1) in the mixing region ($\Delta$POC = − 60 to 10$\mu$M). In an estuary, POC may be added through freshwater and surface runoff mediated inputs, phytoplankton productivity, and DOC flocculation. The removal of POC is likely due to settling at sub-tidal sediment, export to adjacent continental shelf region, modification via conversion to DOC and aerobic respiration in case of oxygenated estuary. The $\Delta\delta$13CPOC and $\Delta$POC were plotted (Figure 2 attached) that showed: [a] decrease in $\Delta$POC with increase in $\Delta\delta$13CPOC (RR) at four locations in the mixing region and one location in the freshwater region suggesting modification of POC via aerobic respiration, supporting our earlier argument. However, this process may not have great impact on estuarine pelagic metabolism as evident from the POC - pCO2 relationship (freshwater region: p = 0.29, mixing region: p = 0.50; Figure will be included in the revised version). [b] Decrease in both $\Delta$POC and $\Delta\delta$13CPOC (SD; n = 2 for mixing region vs. n = 2 for freshwater region) supports settling of POC to sub-tidal sediment. However, due to unstable estuarine condition this process may not be very effective despite high estuarine water residence time ($\sim$ 40 days during postmonsoon, Samanta et al., 2015). [c] Increase in $\Delta$POC with decrease in $\Delta\delta$13CPOC (SR, FR & PP; n = 2 for freshwater region) supports increase of POC via surface runoff or phytoplankton productivity. In the freshwater region, further evidence for surface runoff and primary productivity

influenced POC addition was found based on spatial POC variability and $\delta$13CDIC - POC relationship (freshwater region: r2 = 0.68, p = 0.05) [d] Increase in both $\Delta$POC and $\Delta\delta$13CPOC (n = 1 for mixing region vs. n = 1 for freshwater region) may be linked to DOC to POC conversion by flocculation (more details in DOC section). In freshwater region, direct signal for DOC - POC conversion was evident from the DOC study but not in the mixing zone. In the Sundarbans, negative and lower $\Delta$POCM2 (–209 to –28$\mu$M) compared to $\Delta$POCM1 (–35 to 327$\mu$M) suggested DIC like behaviour, i.e., evidence for simultaneous removal or modifications along with addition of mangrove derived POC. No evidence for in situ POC-DOC exchange was found based on POC - DOC relationship; however, signal for POC mineralization was evident in the Sundarbans from POC - pCO2 relationship (r2 = 0.37, p = 0.05, Figure will be included in the revised version). Similar to the Hooghly, despite having high water residence time in mangroves (Alongi et al., 2005, Singh et al., 2016), unstable estuarine condition may not favour significant settlement of POC at sub-tidal sediment. The export of POC from the Hooghly-Sundarbans system to the Bay of Bengal, without any in situ modification, is also a possibility, which has been estimated to be $\sim$ 0.02 - 0.07Tg and $\sim$ 0.58Tg annually for the Hooghly and Sundarbans, respectively (Ray et al. 2018). Please look later for explanation related to DOC.

Comment: LN349 Are the ground and pore water discharge not being considered as 'biogeochemical' process?

Response: We believe it is better to leave ground and pore water discharge from the realm of biogeochemical processes, as no biogeochemical processes are associated with them. It may be described as hydrological processes. We found "The driving forces of pore-water and groundwater flow in permeable coastal sediments: A review" published by Santos et al. (2012) in the Estuarine Coastal and Shelf Science a nice review work in this field.

Comment: Section 4.3. This part is weakly written and over-speculative without supporting any evidence e.g. the argument of DOC photo-oxidation or conversion of DOC

to POC as removal process. While it requires suitable ambient condition for DOC photo-oxidation such as high water residence time, stable environmental condition (not expected in mangroves), the same applies to adsorption/desorption of DOC-POC. Part of that exchange is mediated by charged complexes, repulsion - attraction interactions, and therefore subject to salinity effects. So, when river water rich in DOC first mixes with saline water, at least a portion of DOC is lost from solution (removed) and incorporated into POC (Fe-oxide colloids usually are extracted at the same time). Once the salinity exceeds 2 - 3, however, the effect of salinity on coagulation behaviour is largely complete. Another point is no detailed explanation on distribution pattern with salinity was given, authors should highlight the reasons of the mild upward gradient along Hooghly and steep downward trend along the Sundarban.

Response: We are thankful to the reviewer for insightful comment on the DOC study. We will try to include these points in the revised manuscript.

The section on DOC may be evolved as following in the revised version: "DOC - salinity relationship did not confirm significant influence of estuarine mixing on DOC at both Hooghly (freshwater: $r2 = 0.33$, $p = 0.23$; mixing region: $r2 = 0.10$, $p = 0.50$) and Sundarbans ($r2 = 0.27$, $p = 0.10$). Our observations showed similarity with other Indian estuaries (Bouillon et al., 2003) with opposite reports from elsewhere (Raymond and Bauer, 2001a, Abril et al., 2002). Although it is difficult to accurately decipher influencing processes on DOC without $\delta13CDOC$, some insights may be obtained from estimated $\Delta DOC$. The estimated $\Delta DOC$ in the Hooghly indicated both net addition (n = 3) and removal (n = 3) of DOC in the freshwater zone ($\Delta DOC = -44$ to $63\mu M$); whereas, only net addition was evident throughout the mixing zone ($\Delta DOC = 18$ to $420\mu M$). In the Sundarbans, except lower Thakuran (St. T3, $\Delta DOC = -20\mu M$), net addition of DOC was estimated throughout ($\Delta DOC = 2 - 134\mu M$). In an estuary, DOC can be added through in situ production (by benthic and pelagic primary producers), lysis of halophobic freshwater phytoplankton cells and POC dissolution. DOC can be removed through bacterial mineralization, flocculation as POC, and photo-oxidation

(Morris et al., 1978; Alvarez-Salgado et al., 1998; Bouillon et al., 2006). At the Hooghly - Sundarbans system, no evidence for freshwater phytoplankton ($\delta$13C: $-$ 33 to $-$ 40‰ $Freitas et al., 2001$) was found from $\delta$13C POC, ruling out its potential effectiveness on DOC. Based on $\delta$13CDIC and POC study (r2 = 0.68, p = 0.05), an indirect evidence for phytoplankton productivity was observed in the freshwater region of the Hooghly, but direct evaluation of its impact on DOC was not possible due to lack of data. Contradictory results exist regarding influence of phytoplankton productivity on DOC. Some studies did not find direct link between DOC and primary productivity (Boto and Wellington, 1988), whereas primary productivity mediated significant DOC formation ($\sim$ 8 - 40%) has been reported by others (Dittmar & Lara (2001a), Kristensen & Suraswadi (2002)). The DOC - pCO2 relationship suggested inefficient bacterial DOC mineralization in the Hooghly (freshwater zone: p = 0.69, mixing zone: p = 0.67). However, significant positive relationship between these two in the Sundarbans (r2 = 0.45, p = 0.02) indicated increase in aerobic bacterial activity with increasing DOC. In mangrove ecosystems, leaching of mangrove leaf litter as DOC is fast as $\sim$ 30% of mangrove leaf litter leaching as DOC is reported within initial 9 days of degradation (Camilleri and Ribi, 1986). In the Sundarbans, mangrove litter fall peaks during postmonsoon (Ray et al. 2011) and its subsequent significant leaching as DOC was evident during the present study from comparatively higher DOC compared to POC (DOC:POC = 0.50 $-$ 3.39, mean: 1.79 $\pm$ 0.94%). Our interpretation corroborated with that reported by Ray et al. (2018) for the same system as well as Bouillon et al. (2003) for the Godavari estuary, South India. Despite having high water residence time in the Hooghly ($\sim$ 40 days during postmonsoon, Samanta et al., 2015) and in mangroves (Alongi et al., 2005, Singh et al., 2016), degree of DOC photo-oxidation may not be so potent due to unstable estuarine condition in the Hooghly-Sundarbans system (Richardson number < 0.14) with intensive vertical mixing with longitudinal dispersion coefficients of 784 m2 s$-$1 (Goutam et al., 2015, Sadhuram et al., 2005). The unstable condition may not favour DOC - POC interconversion as well but mediated by charged complexes and repulsion - attraction interactions, the interconversion partly depends upon variation in salinity. More

specifically, the interconversion is efficient when fresh (river) water mixes with seawater and the coagulation is mostly complete within salinity range 2 – 3. This appeared to be the case in the Hooghly where DOC and POC was negatively correlated (r2 = 0.86, p = 0.007) in the freshwater region, which was missing in the mixing region (p = 0.43) and in the Sundarbans (p = 0.84). Although estimated $\Delta$DOC indicated mostly net DOC addition in the Hooghly-Sundarbans system, except leaf litter leaching in the Sundarbans, no significant evidence for the same was found. This suggested potential influence of exogenous (with respect to estuary) DOC sources to the estuary. Although there is no quantitative data available to justify this argument, DOC influx via surface water runoff is expected to be much higher in the freshwater region of the Hooghly due to presence of several jute industries and major cities including Kolkata (St. H2; population density: 22000 per km2, 7th highest in India), a hotspot for waste water disposal in this region. Relatively higher DOC level in the mixing zone compared to freshwater region suggested potential role of some other processes, possibly groundwater discharge. Contradictory results exist regarding contribution of groundwater discharge in the Hooghly. Based on dissolved Ca, groundwater contribution to the Hooghly estuary has been suggested by Samanta et al. (2015) at low salinity regime (S < 10, our salinity range); however, no signal for the same was found based on 'Ra' isotopic study by Somayajulu et al. (2002). Based on the present data, it is not possible to justify groundwater mediated DOC addition to the Sundarbans. Maher et al. (2013) estimated $\sim$89 - 92% of the total DOC export from mangrove driven by groundwater advection. To understand spatial variability of DOC chemistry in the Hooghly-Sundarbans system, a thorough investigation related to groundwater discharge and surface runoff mediated DOC flux is warranted.

Comment: Section 4.4 LN410 only freshwater runoff, no surface run off that adds POC too in upstream?

Response: We will include possibility of surface runoff mediated POC addition in the revised manuscript. The section may look like: In general, the salinity-SPM and salinity-

POC relationships (Figure will be included) were not significant, except freshwater region of the Hooghly, where a moderate negative correlation between POC and salinity was observed ($r2 = 0.62$, $p = 0.06$). The inverse relationship may be linked to freshwater mediated POC influx with additional contribution from surface runoff from adjoining areas. In the freshwater zone, contribution from surface runoff was more evident as $\sim$ 61% and $\sim$ 43% higher POC at 'H3' and 'H4' were observed compared to the upstream location 'H2'. However, based on the present data, decoupling between freshwater and surface runoff mediated POC inputs was not possible.

Comment: LN440-446 this part is totally redundant as there was not an iota of signal of CH4 from the observed d13 POC (13CH4 is 55-60 permil)

Response: We have removed the section from the revised manuscript.

Comment: Does the author have Chl-a or nutrient data (even from literature) to support higher marine input in POC in Sundarban and 13C values of mangrove leaf, and soil from Hooghly to denote higher terrigenous contribution to the POC pool? Authors are suggested to read carefully the works of Samanta'15 and Ray'18 and use their values to support some of the arguments.

Response: We are thankful to the reviewer for this suggestion. We have supported higher terrestrial contribution in the Hooghly based on previously reported nutrient and chlorophyll concentrations in this region compared to the Sundarbans. 13C values of mangrove leaf and soil from Sundarbans as reported by Ray et al. (2018) also included in the revised manuscript as an evidence to establish higher marine influence or modification of POC within the estuaries of Sundarbans. The addition regarding this in the section may look like as following: Our interpretation regarding higher terrestrial contribution in the Hooghly was also corroborated by previously reported relatively higher dissolved inorganic nutrients and Chl-a concentrations in the Hooghly (DIN: $14.72 \pm 1.77$ to $27.20 \pm 2.05 \mu M$, DIP: $1.64 \pm 0.23$ to $2.11 \pm 0.46 \mu M$, DSi: $77.75 \pm 6.57$ to $117.38 \pm 11.54 \mu M$, Chl-a: $2.35 - 2.79$ mgm-3) compared to the Sundarbans (DIN:

11.70 ± 7.65μM, DIP: 1.01 ± 0.52μM, DSi: 75.9 ± 36.9μM, Chl-a: 7.88 ± 1.90 mgm-3) based on seasonal and spatial investigations (Biswas et al., 2004, Mukhopadhyay et al., 2006, Dutta et al., 2015). In the Sundarbans, possibility of POC modification was evident as $\delta$13CPOC during the present study was comparatively higher than previously reported values of $\delta$13C of mangrove leaf ($\delta$13C $\sim$ −28.4‰ and soil ($\delta$13C $\sim$ −24.3‰ for the same system (Ray et al., 2015, 2018). Also, despite large sewage ($\delta$13C $\sim$ −28.56 to −22.14 ‰ Andrews et al., 1998) discharge from Kolkata (St: H2) relatively higher $\delta$13CPOC in the Hooghly also adds to the possibility of POC modification in Hooghly. Modification of POC within the estuaries of Indian sub-continent has also been reported by Sarma et al. (2014).

points of concerns

Comment: terminology > I counted 'biogeochemistry' was used over 25 times in the 16 pages ms! too much. Additionally, this is not clear to me what does it actually mean by C biogeochemistry?

Response: We will take care of it in the revised manuscript.

Comment: Is it C-components distributions in different phases (solid suspended and dissolved) under varying biogeochemical processes? If so please specify at least once

Response: Agreed. We will include it in the abstract of the revised manuscript.

Comment: > d13C values are not 'depleted' or 'enriched' (LN256, 428..). When referring to d13C values, they can be described as higher or lower when comparing different samples, or one could describe differences as e.g. a certain C pool is enriched or depleted in 13C versus another C pool or sample.

Response: Agreed. We will take care of it in the revised manuscript.

Comment: > r2 not R2

Response: We will change in the revised version.

Comment: Inconsistent use of [POC] in the discussion, if the bracket is used for POC then it should also appear for DIC and DOC

Response: Brackets will be removed for all cases in the revised manuscript.

Comment: unit Random use of units: DOC in mg/L, DIC in mM, POC in uM. These should be harmonized. Use DOC in uM for better compare with other studies

Response: To maintain uniformity we will change the units to '$\mu$M' in the revised version.

Comment: Sampling Define sampling strategy neatly, Its written postmonsoon was chosen due to high litterfall, but there is no account of litter source identified for DOC or POC or any impact positive or negative on estuarine C biogeochemistry authors assumed. That is to be addressed in the discussion. Mention the H, S, T, M series in the text Mention general tidal nature while sampling (height, HT/LT, depth).

Response: The leaf litter fall is the main source of organic carbon in mangrove sediment, which peaks during postmonsson (Ray et al., 2011). It is expected that high litter fall might influence C components in the Sundarbans. The signal for influence of litter fall on DOC was evident from the DOC:POC ratio (as leaching) in the Sundarbans, but no direct signature for mangrove leaf litter on POC was found (modification is also a possibility, see POC section for more details). We are ready to include these points in the revised manuscript. Details on 'H, S, T and M' will be included in the revised manuscript. All samples were collected during the low tide phase as intertidal mangrove sediment - water interaction through groundwater discharge is maximum during low tide phase. Therefore, low tide is ideal sampling time to understand impact of mangroves on adjoining estuarine systems. To assess contrasting features between the Sundarbans and Hooghly, sampling was also conducted during low tide in the Hooghly estuary. We will include these points in the revised manuscript.

Methods

Comment: specify> pore size of filters used for DOC, SPM relative uncertainty in POC methods;

Response: Pre-combusted (500oC for 6 hours) Whatman GF/F (pore size: 0.7$\mu$m) was used for DOC filtration and SPM collection. Uncertainty for POC was < 10%. This information will be included in the revised manuscript.

Comment: technique of pore water collection; ground water (from tube pump?)

Response: We will add collection techniques for pore-water and groundwater in the revised manuscript on following lines: During low tide condition, pore-water from mangrove forest floor was collected from Lothian Island (one of the virgin island of Sundarbans) by digging a hole of $\sim$ 30 cm below the water table. After purging water at least twice in the bore, sample was collected from the bottom of the bore through syringe and transferred to the glass vial (Maher et al., 2013). Groundwater samples were collected from the nearby locations of the Hooghly-Sundarbans system via tube pump.

Figures

Comments: Again weak representation: font sizes of x, y axis digits (and titles) to be increased much (too much stress to eyes now!); use box to cover legends, its confusing with data points and legends, remove break in y axis in Fug 3e and 4a), black star coding was used both for sundarban and observed d13DIC and grey round coding was used for Hooghly and observed DIC, these symbols must be changed to give separate identity of them in all figs <overall IMPROVE CLARITY of ALL FIGURES>

Response: We will improve the figures as suggested.

Comment: Data use a consistent number of decimals (1) to report d13C data, and Salinity considering the analytical error on the measurements.

Response: Ok. We will take care of it in the revised manuscript and both salinity and $\delta$13C data will be presented up to two decimals.

Minor comments

Comment: First sentence of abstract is redundant

Response: We will remove it from the revised manuscript.

Comment: LN65 Use current reference for the riverine export flux (works of Pete Raymond, Huang)

Response: We are thankful to the reviewer for suggestion. We will include Huang et al. (2012) in the revised manuscript at appropriate place.

Comment: Many references are out of place e.g. the comparison of present data with Khura (LN 231, 249 Miyajima paper) was unlikely as two environments are totally different even if compared authors should mention conservative data like S in Khura estuary for better comparison.

Response: We will present better comparison in the revised manuscript.

Comment: LN234: Pro-vide values of Samanta et al 2015

Response: We will provide postmonsoon DIC $(1.70 – 2.25mM)$ and $\delta13CDIC$ $(–11.4$ to $– 4.0‰$ values of Hooghly estuary as reported by Samanta et al. (2015) in the revised manuscript.

Comment: Finally, I think it is necessary to stand back and consider how to best weave the entire story together in the discussion more efficiently and succinctly

Response: Thanks for the valuable suggestions.

———————————————————

[Figure]

[Figure]

*Ray et al. (2015)*          *Ray et al. (2018)*

Figure 1. Spatial coverage in the Hooghly-Sundarbans by Ray et al. (2015, 2018)

**Fig. 1.**

Fig 2 Plot between ΔPOC and $\Delta\delta^{13}C_{POC}$ in the Hooghly. Green and grey circles indicate location in the freshwater and mixing zones. SR: surface runoff; FR: freshwater runoff, PP: primary productivity; SD: Settling to sediments; RR: respiration.

**Fig. 2.**

---

## Author Comment (AC2) · 10 Sep 2018

Comment: Review of Dutta et al "The postmonsoon carbon biogeochemistry of estuaries under different levels of anthropogenic impact". Submitted to Biogeosciences. This study presents data from a single cruise in 2 Indian estuaries to try and decipher differences in carbon cycling between a 2 Indian estuaries with differing levels of anthropogenic influence. After reading and rereading this paper several times, it is unclear what the purpose of this study is. There is no defined hypothesis to be tested, and while the title suggests there will be some kind of comparative analysis to look at anthropogenic impact on carbon cycling (an interesting and important topic), I am left

a little underwhelmed with the analysis undertaken. The entire manuscript is based single sampling campaigns, which while not ideal is not the main issue. The main area of concern is the lack of any direction in the paper, and the somewhat descriptive and qualitative nature. I suggest that the authors define their hypothesis more clearly, and use the data to test this hypothesis.

Response: Thanks to reviewer for going through our manuscript and providing valuable suggestions which will help to improve the quality of the revised version. We understand the concern he has raised and we are trying to improve the manuscript accordingly. As we have said in the response to reviewer 1, the main objective of the present study is to bring out contrast in different components of the carbon cycle of anthropogenically affected Hooghly estuary and mangrove-dominated estuaries of the Sundarbans during postmonsoon. As suggested by the reviewer later in comments, we have introduced a table bringing out the differences in basic characteristics of these two systems, which will help the readers to appreciate the differences in anthropogenically affected and mangrove-dominated system. As suggested by the reviewer, in the revised version, given the contrasting nature of the estuaries, we also propose to bring out a central hypothesis. The central hypothesis of this study would be: exchange of $CO_2$ to be significantly higher in anthropogenically impacted estuary than the mangrove-dominated estuary during the postmonsoon. Given the larger spatial coverage of the mangrove-dominated estuary during the present (so far only one estuary in this system has been studied), there is a need for this hypothesis to be tested on wider spatial level.

I have a series of comments below, some minor some major that may help.

Comment: Abstract I am not convinced that the data as presented can be used to draw such strong conclusions as to the drivers of carbon dynamics in the studied estuaries. For example, Ln 35-38 The evidence supporting these processes is weak at best – no measurements of production, carbonate dissolution nor pore-water exchange were measured, and the spatial trends in concentrations and isotopes (and relationships

between carbon variables and DO etc.) were not strong enough to draw any distinct conclusions on the importance of these mechanisms. Same goes for lines 45-47.

Response: Based on the specific comments of the reviewers, we have re-analysed the data and reassessed the role of processes he is referring to in sentences mentioned above. In the response to comments below, he will find that we have either discarded the descriptive part or backed the processes active with reanalysis of the data during the present study.

Comment: Line 49 – 52. I am unconvinced that the observed trends are shown to be directly linked to anthropogenic influence. Yes, the estuaries appear to differ, but what else might be driving this. For example, looking at salinity and pCO2 in the 2 different estuaries – the highest salinity in the "anthropogenically" impacted estuary is lower than the lowest salinity in the "undisturbed" estuaries. Could the observed differences simply be related to freshwater input? What are the nutrient concentrations in the 2 estuaries? How different are they in hydrodynamics (looks like the geomorphology is distinctly different between the 2 estuary types from Fig 1). These are just a few of the alternative reasons to look at for explaining the differences observed

Response: Based on the comments from both reviewers, we have provided a table in the supplement (point - 1), which will help readers to understand the basic differences between the two estuaries. The present study was carried out during postmonsoon season, which brings significant amount of freshwater inputs to the region. Moreover, the Hooghly undergoes sever anthropogenic stress as it passes through industrial areas as well as one of the most densely populated region in India (included in table). We revisited the data in light of the comments from both reviewers and in responses we discuss the changes and processes active in the two estuaries, which led to observed difference.

Introduction

Comment: Ln 59 – 60 What is meant by "record biogeochemical and hydrological

processes"?

Response: We meant physical/hydrological processes such as mixing between marine and freshwater, tide and wave action, sediment transport etc. and biogeochemical processes such as primary productivity, organic matter decomposition etc. We believe the reviewer was concerned with 'record'. We will modify the sentence in the revised manuscript.

Comment: Ln 67 – Richey is not correct ref for this statement (Richey paper is on Amazon)

Response: Thanks to point this out. We will modify the statement with correct reference.

Comment: Ln 68 – 70 – Still large uncertainties on estuarine CO2 flux – look at error bars on Cai, 2011 estimate Response: We will point out this issue in the revised manuscript.

Comment: Ln 76 What is meant by "biogeochemical characteristic"?

Response: We meant with regards to cycling of bio-available elements, such as C, N and P. We will change this sentence to more specific.

Comment: Ln 78 – 79 Not always – see Cotovicz Jr, L. C., Knoppers, B. A., Brandini, N., Costa Santos, S. J., & Abril, G. (2015). A strong $co_2$ sink enhanced by eutrophication in a tropical coastal embayment (guanabara bay, rio de janeiro, brazil). Biogeosciences, 12(20), 6125-6146. doi:10.5194/bg-12-6125-2015

Response: Thanks for this reference. In the revised version, the sentence may look like: In anthropogenically affected estuarine systems, heterotrophy generally dominates over autotrophy (Heip et al., 1995; Gattuso et al., 1998) and a substantial fraction of biologically reactive organic matter gets mineralized within the system (Servais et al., 1987; Ittekkot, 1988; Hopkinson et al., 1997; Moran et al., 1999). However, this is not always the case as observed in Guanabara Bay, Brazil, which acts as a strong

CO2 sink enhanced by eutrophication (Cotovicz Jr. et al., 2015)."

Among others

Comment: Ln 81 – 84 There has been a lot of work on mangrove carbon cycling work done since Dittmar and Larra's work in the early 2000's. Might be worth looking at more recent papers to see how far our understanding has come since then.

Response: Sure. We will modify this section in the revised manuscript. Following information may be added: Around ~50% of mangrove net primary productivity (112-160 Tg C yr-1) is imbalanced by various sinks as estimated by global mangrove C budget (Bouillon et al., 2008, Alongi, 2009, Breithaupt et al., 2012). Litter fall is identified as a primary source of mangrove derived C input in mangrove sediment and fate of this C remains a topic of research. Earlier studies reported that mangroves were responsible for ~10% of the global terrestrial derived POC (Jennerjahn and Ittekkot 2002) and DOC (Dittmar et al. 2006) export to the coastal zones. However, recent studies proposed DIC exchange as major C export pathway from mangrove forests, which is responsible for ~70% of the total mineralized C transport in coastal waters (Maher et al., 2013 Alongi, 2014; Alongi and Mukhopadhyay, 2014). Another study reported groundwater advection to be responsible for 93–99% of total DIC export and 89–92% of total DOC export to the coastal ocean (Maher et al.,2013). Upon extrapolating these C exports to the global mangrove area, it was found that the calculated C exports were similar to the missing mangrove C sink (Sippo et al., 2016). After its outflux as DOC, DIC and POC from mangrove system to the adjoining aquatic system, the remaining mangrove C gets buried in sediment layers to participate in anaerobic reactions in subsurface deep sediment layers or undergoes long-term sequestration (Jennerjhan and Ittekkot 2002; Barnes et al., 2006; Kristensen and Alongi, 2006; Donato et al., 2011; Linto et al., 2014).

Comment: Ln 104- 106 Give some quantitative data to support your "anthropogenically influenced" argument. What are nutrient concentrations like? Population density? Land

use? Freshwater inflow? Etc etc. A table compiling this data would give the reader an instant understanding of the differences.

Response: Thanks to the reviewer for bringing this point. Reviewer 1 has also asked to include some information in this context from literature. Texts or a table comparing the Hooghly and Sundarbans during postmonsoon based on nutrients concentration, Chla, population density and freshwater inflow will be introduced in the revised manuscript. The information in tabular form is attached as Supplement file (point - 1).

Comment: Line 117 What is meant by positive and negative feedback here? These terms are not really applicable to biogeochemistry as a whole, but may be related to specific mechanisms/cycles.

Response: Ok. In the revised manuscript we will change this statement as follows: The postmonsoon sampling was chosen because of relatively stable estuarine condition for spatial sampling and reported peak mangrove leaf litter fall during this season (Ray et al., 2011), which may have influence on estuarine C dynamics.

Comment: Ln 137-140 Clearly there is freshwater input – the salinities are very low. In fact, my thoughts are that these freshwater inputs are a main driver of the observed differences.

Response: The freshwater input in the estuaries of Sundarbans is evident from the salinity values (12.64-16.69) during the study period. However, if you see the salinity values in the Hooghly estuary during the same season (0.04-10.37), the extent of freshwater input in Hooghly is far greater. Because of this reason, we stated 'no perennial source of freshwater and limited anthropogenic input during monsoon". We may change the sentence as: The present study was carried out in three major estuaries of the Indian Sundarbans (Saptamukhi, Thakuran and Matla) covering upper, middle and lower estuarine locations."

Comment: Ln 159 Assume the filters were GF/F filters – add these details.

Response: Yes, as reviewer stated it was Whatman GF/F filters. We will include it in the revised manuscript.

Comment: Ln 161 Accuracy of TAlk measurements. Were CRMs measured (hope so!). Also add accuracy/precision etc of all other parameters.

Response: Uncertainties were as follows: Water temperature: $\pm 0.1$oC, Salinity: $\pm 0.1$, DO: $\pm 0.1$ mgL-1, DIC: <1%, $\delta$13CDIC: < $\pm 0.10$‰ DOC: $\pm 52$ $\mu$gL-1, POC: <10%, $\delta$13CPOC: < $\pm 0.10$‰ pCO2: $\pm$ 1%. Yes, accuracy of TAlk was tested using Dickson standard (CRM: Batch – 131) and uncertainty was found to be $\pm 1 \mu$molkg-1.

Comment: Ln 196 – 198 What were the input parameters for measuring pCO2? What disassociation constants were used etc?

Response: The pCO2 was calculated using TAlk, pH, water temperature and salinity and the dissociation constants were calculated following Millero, (2013). We will include that information in the revised manuscript.

Comment: Ln 205 – 208 Why use L&M relationship? Need some kind of justification here other than saying it is conservative.

Response: Unfortunately, we don't have data on estuarine current velocity which along with wind speed is used for flux calculation as it is believed that turbulence of estuary might have an effect on air-water trace gas flux calculation. Based on only wind velocity, the L&M relationship is one of the most reliable and tested methods for flux calculations, which has been used in previous studies in the region as well (Biswas et al., 2004).

Results

Comment: Do not compare and contrast your data with previous studies in the results. Just report your data.

Response: We will remove the comparison part from the result.

Discussion

Comment: Ln 289 – 293 What are the implications for these findings? Need to dig deeper or remove.

Response: Our intension was to present influence of salinity on pH and provide the information at the beginning that this region is a bicarbonate dominated system. We will remove the sentences in the revised version.

Comment: Ln 306-311 (and Fig 3b) How was the conservative d13C-DIC mixing line calculated? Looks like you have simply added a linear relationship between the 2 endmembers, the relationship is generally not linear (See Fry, B. (2002). Conservative mixing of stable isotopes across estuarine salinity gradients: A conceptual framework for monitoring watershed influences on downstream fisheries production. Estuaries, 25(2), 264-271. Also as you do not have any mineralogy of carbonates – I would avoid using the term "calcite" precipitation, change to "carbonate" precipitation.

Response: Conservative $\delta$13CDIC mixing line was calculated using the expression given by Mook and Tan (1991) as given in the attached supplement file (point - 2). We will change 'calcite precipitation' as 'carbonate precipitation' in the revised manuscript.

Comment: Ln 323-325 – What does DO tell you about primary production? Looks like DO is generally under-saturated?

Response: The influence of primary productivity (PP) and/or CO2 outgassing on DIC at the mixing zone was evident from mixing plot between $\Delta$DIC and $\Delta\delta$13CDIC. We tried to go further and decouple these two processes based on TAlk - DIC relationship. However, as suggested by the reviewer, due to lack of PP measurements and level of DO indicate that it may not be a stretch. We will be remove this part from the manuscript.

Comment: Ln 335-338 Describe all the terms in this equation in the following text

Response: Here, DIC and DICCM indicate DIC concentration of the sample and DIC concentration due to conservative mixing, respectively. The $\delta$13CDIC, $\delta$13CDIC(CM)

and $\delta$13CMangroove indicate C isotopic compositions of DIC of the sample, C isotopic composition of DIC under conservative mixing, and C isotopic composition of mangroves, respectively. We will include it in the revised manuscript. Additionally, in the revised manuscript, $\delta$13CMangrove will be changed as - 28.4‰ as reported by Ray et al. (2015) for the Sundarbans system.

Comment: Ln 359 Where do the TAlk/DIC numbers come from? The stoichiometric relationship should be based on the slope of the line over the whole estuary, rather than individual data points – therefore not sure how you have a range here.

Response: Thanks to the reviewer for this suggestion. Based on his advice, we will make necessary changes in the text and improve this section on following lines: High pCO2 and DIC along with low pH and TAlk/DIC are general characteristics of groundwater, specially within carbonate aquifer region (Cai et al., 2003). Although all the parameters of groundwater inorganic C system (like pH, TAlk and pCO2) were not measured during the present study, groundwater DIC were $\sim$5.57 and $\sim$3.61 times higher compared to average surface water DIC in the Sundarbans and Hooghly, respectively. The markedly higher DIC in groundwater as well as similarity in its isotopic composition with estuarine DIC may stand as a signal for influence of groundwater on estuarine DIC, with possibly higher influence at the Sundarbans than Hooghly as evident from the slope of the TAlk - DIC relationships (Hooghly: 0.98, Sundarbans: 0.03). In the Sundarbans, to the best of our knowledge, no report exists regarding groundwater discharge. Contradictory reports exist for the Hooghly where Samanta et al. (2015) indicated groundwater contribution at low salinity regime (salinity < 10, same as our salinity range) based on 'Ca' measurement, which was not observed based on 'Ra' isotope measurement in an earlier study (Somayajulu et al., 2002).

Comment: Ln 364 – 368 Give details on this calculation. Just using the discharge rate and pore water DIC concentration I get a different value.

Response: Advective DIC flux from intertidal mangrove sediment to estuarine water

column (FISW) was computed using the relation (Reay et al., 1995); FISW = $\Phi.\nu.C$; where, $\Phi$ = porosity of sediment = 0.58 (Dutta et al., 2013), $\nu$ = average linear velocity = d$\Phi$-1 (d = specific discharge), C = DIC concentration in intertidal sediment pore water. So ultimately: FISW = d.C. During postmonsoon, d = 0.008 cm min-1 (Dutta et al., 2015a). Therefore, FISW = (0.008 cm min-1 x 13.43 mmolL-1) = 0.107mmol.cm.min-1/1000cm3= 0.000107 mmol cm-2 min-1 = 1.07 mmol m-2 min-1. In Sundarbans, tides are semidiurnal in nature, so depending upon changes in hypsometric gradient discharge of pore water will be effective during low period only (i.e. 12 hours). So, FISW =1.07 mmolm-2min-1 = (1.07 x 60 x 12 mmolm-2d-1) =770.4 mmolm-2d-1. There is a marginal difference in the manuscript, which will be corrected. We will put the formula for calculation in the revised manuscript. We hope calculation is clear to the reviewer.

Comment: Ln 383-390 – Not sure that looking at pCO2 VS DOC gives any indication as to the importance of pore-water exchange! Could also simply be freshwater input from upstream, surface water runoff, or simply leaching/respiration.

Response: We have suggested to modify the DOC section which does not include the above argument. Please see response to reviewer 1 which deals with DOC (section 4.3).

Comment: Ln 412 Give details about the "jute" industry.

Response: This is an industry based on fiber of Corchorus plants, which is used in fabrics for packaging a wide range of agricultural and industrial commodities that require bags, sacks, packs, and wrappings. Locally this is known as Jute industry. We will put the scientific name of the plant in the revised manuscript.

Comment: Ln 424-426 The POC isotopes could simply be related to the relative amount of freshwater inputs in each system (this can also be applied to most of the other differences observed)

Response: If relative amount of freshwater in each system had a major control on POC

isotopes, we would expect salinity dependent $\delta 13CPOC$ variability, which was not noticed. We can modify the referred sentence on following lines: The $\delta 13CPOC$ - salinity relationships in the Hooghly (freshwater region: p = 0.20, mixing region: p = 0.79) and in the Sundarbans (p = 0.65) did not confirm significant influence of freshwater on $\delta 13CPOC$. However, on an average, $\delta 13CPOC$ at the Hooghly (− 24.87 ± 0.89‰ was relatively lower compared to that of Sundarbans (—23.36 ± 0.32‰ suggesting relatively higher influence of terrestrial inputs in the Hooghly.

Comment: Ln 431-446 I am unsure why anaerobic respiration (which is energetically les favourable than aerobic respiration) would be more important in a well oxygenated estuary. The authors should expand this to explain things more clearly or remove.

Response: We will remove the anaerobic respiration part from the revised manuscript.

Comment: Ln 447-451 What is the importance/implications of this – expand or remove.

Response: The intension was to quantitatively explore dominant OC form (DOC or POC) in total OC pool and dominant dissolved C form (DIC or DOC) in total dissolved C pool in the estuary. We will remove the lines as reviewer suggested.

Comment: Ln 455 – 460 These sections seem to contradict each other. Initially it is stated mangrove inputs are insignificant – then pore-water exchange of mangrove derived CO2 is highlighted as important?

Response: For the revised manuscript, ECO2 - AOU relationship (as suggested by the reviewer) was investigated (please see response to a later comment). The significant positive relationship between the two (ECO2 = 0.057AOU + 1.22, r2 = 0.76, p = 0.005, n = 8) suggested influence of OM respiration on pCO2 in the Sundarbans. Although, the calculated slope (0.057) was markedly lower compared to the slope for Redfield respiration in HCO3- rich environment [$\Delta$CO2: (-$\Delta$O2) = 124/138 = 0.90, Zhai et al., 2005] indicating effect of OM mineralization in controlling pCO2 to be not so potent. Therefore, possibility of pore-water mediated CO2 influx cannot be totally neglected in

mangroves. Although based on present dataset (only low tide phase sampling) it is not possible to justify the argument, a signal for it was also observed from 24 hours pCO2 observation in the Matla estuary (Sundarbans) by Akhand et al. (2016). We will add these observations in the revised version.

Comment: Ln 463 – Cai ref – there are plenty of mangrove references for this process, might be more appropriate to use some of those here

Response: We agree. We will include some other mangrove references in the revised manuscript, such as Call et al. (2015), Bouillon et al. (2007).

Comment: Ln 463 – 466 How about plotting ECO2 vs AOU (in molar units). Look at the slope of the line. This will give a better indication of the importance or aerobic vs anaerobic R.

Response: In the Sundarbans, barring three locations ( S3, T3 and M2), a positive correlation between ECO2 and AOU was noticed (ECO2 = 0.057AOU + 1.22, r2 = 0.76, p = 0.005, n = 8) suggesting aerobic OM mineralization in the system, particularly in the upper region. Although, the calculated slope (0.057) was markedly lower compared to the slope for Redfield respiration in HCO3- rich environment [$\Delta$CO2: (-$\Delta$O2) = 124/138 = 0.90, Zhai et al., 2005] indicating effect of OM mineralization in controlling pCO2 to be not so potent. Also, the effectiveness of salinity on pCO2 was ruled out based on pCO2-salinity relationship in the Sundarbans. No significant relationship between ECO2 and AOU was observed in either freshwater or mixing zones of the Hooghly estuary suggesting limited role of organic matter respiration on CO2. However, significant positive and negative relationships between pCO2-salinity were noticed in the freshwater (r2 = 0.71, p = 0.04) and mixing zones (r2 = 0.72, p = 0.03, n = 6) of the Hooghly. The positive pCO2-salinity relationship coupled with no significant relationship between ECO2 and AOU suggest exogenous CO2 supply in freshwater region (probability from surface runoff), whereas freshwater mediated addition of pCO2 is evident in the mixing zone. In the revised manuscript, we will explain these observations.

[Figure]

Comment: Ln 470 – 473 How was gas exchange and the differences between CO2 and O2 coupled into this calculation? Also how does this value compare to your air-water CO2 fluxes (you will need to normalize your volumetric rates to surface area for comparison)

Response: In both freshwater and mixing zone of the Hooghly estuary, no evidence for significant impact of aerobic OM respiration on pCO2 was found. Therefore, we will remove this section from the revised version.

Comment: Ln 480 – I think your global value for mangrove systems (63 umol/m2/d) should be 63 mmol/m2/d – which is much higher than the fluxes measured in this study.

Response: We are thankful to the reviewer for pointing this out. We have rechecked the value from Call et al. (2015). The actual value (range) is $\sim$ 43-59 mmol C m-2d-1. We will correct it in the revised manuscript.

Conclusions:

Comment: Point 1 – this variability is likely simply linked to the variability in salinity (and therefore freshwater inputs) between the studied estuaries.

Response: Freshwater inputs definitely has a role to play in the variabilities observed. However, these variabilities are also linked to in situ processes in the estuaries as described in our responses to both the reviewers.

Comment: Point 2 – Unconvinced that primary production has been shown to be the main controlling factor on DIC. Without any measurements of PP or some more thorough analysis of other potential mechanisms, this statement is far too strong

Response: We agree with the reviewer. We will modify the DIC section accordingly. As we said in our response earlier, we were trying to decouple the CO2 outgassing and PP for locations which were falling into that quadrant of $\Delta$DIC and $\Delta\delta$13CDIC plot. However, due to lack of direct PP data, we have decided to not do that in the revised

manuscript.

Comment: Point 3 – I see no strong conclusive evidence of either of these points. Again statement is too strong without measurements of DOC flocculation or porewater exchange of DOC

Response: In light of the comments from both the reviewers on this section, we are proposing to significantly modify the DOC section. Please see our response to reviewer 1 in this regard (response for section 4.3). We have avoided emphasizing the processes for which we do not have direct data.

Comment: Point 4 Assume this is based on isotopes? Again this could simply be related to the marked differences in freshwater content within each of the estuaries.

Response: If relative amount of freshwater in each system had a major control on POC isotopes, we would expect salinity dependent $\delta$13CPOC variability, which was not noticed. We can modify the referred sentence on following lines: The $\delta$13CPOC - salinity relationships in the Hooghly (freshwater region: p = 0.20, mixing region: p = 0.79) and in the Sundarbans (p = 0.65) did not confirm significant influence of freshwater on $\delta$13CPOC. However, on an average, $\delta$13CPOC at the Hooghly (− 24.87 ± 0.89‰ was relatively lower compared to that of Sundarbans (—23.36 ± 0.32‰ suggesting relatively higher influence of terrestrial inputs in the Hooghly.

Please also note the supplement to this comment:
https://www.biogeosciences-discuss.net/bg-2018-310/bg-2018-310-AC2-supplement.pdf

**Supplement:**

**Point – 1:**

| Parameters | Hooghly | Sundarbans | Reference |
|---|---|---|---|
| **Nutrients (postmonsoon)** | DIN: 14.72 ± 1.77 to 27.20 ± 2.05µM
DIP: 1.64 ± 0.23 to 2.11 ± 0.46µM
DSi: 77.75 ± 6.57 to 117.38 ± 11.54µM | DIN: 11.70 ± 7.65µM
DIP: 1.01 ± 0.52µM
DSi: 75.9 ± 36.9µM | Biswas et al. (2004)
Mukhopadhyay et al. (2006) |
| **Chl*a* (postmonsoon)** | Chl-a: 2.35 – 2.79 mgm$^{-3}$ | Chl*a*: 7.88 ± 1.90 mgm$^{-3}$ | Mukhopadhyay et al. (2006), Dutta et al. (2015) |
| **Population density (districts located on banks of the River Hooghly)** | North 24 Parganas and Hooghly: 2500 km$^{-2}$
Kolkata: 22000 km$^{-2}$
Howrah: 3300km$^{-2}$
South 24 Parganas: 820km$^{-2}$ | No major Cities and town | |
| **Freshwater inflow (postmonsoon)** | 3070 - 7301 million m$^3$ | No information available | Rudra et al. (2014) |

**Point – 2:**

Conservative $\delta^{13}C_{DIC}$ mixing line was calculated using the expression given by Mook and Tan (1991) as given below:

$$\delta^{13}C_{DIC(CM)} = \frac{Sal_S\,[DIC_F\,\delta^{13}C_{DIC(F)} - DIC_M\,\delta^{13}C_{DIC(M)}] + Sal_F\,DIC_M\,\delta^{13}C_{DIC(M)} - Sal_M\,DIC_F\,\delta^{13}C_{DIC(F)}}{Sal_S\,(DIC_F - DIC_M) + Sal_F\,DIC_M - Sal_M\,DIC_F}$$

Here, 'Sal' denotes salinity, the suffixes CM, F, M and S denote conservative mixing, freshwater end member, marine end member and sample, respectively. $F_F$ = freshwater fraction $= 1 - (Sal_S / Sal_M)$ and $F_M$ = marine water fraction $= (1 - F_F)$. This is a commonly used expression for such studies and has been followed by many other workers (Samanta et al. (2015); Bouillon et al. (2003))

---

## Author Comment (AC3) · 10 Sep 2018

Comment : In the present study, large spatial extent has been covered which includes Hooghly River and other rivers of Indian part of Sundarban. My comments regarding the present study are as follows: 1. From the sampling strategy (line no. 150 to 153), it is apparent that only one-time discrete sampling has been done in all the sites in duplicate, whereas from the third objective of the study it is clear that the authors had the aim to quantify and characterise the air-water $CO_2$ flux for the post-monsoon season. The authors concluded "During post monsoon, the entire Hooghly-Sundarbans system acted as a source of $CO_2$ to the regional atmosphere." How can it be concluded
* * *
(even quali-tatively) from such discrete data without performing at least one complete diurnal sampling at each site within post-monsoon season, while four months (October, November, December and January) are generally considered as post monsoon season in this region?

Response: As we have stated in response to the reviewer 1, the aim of the present study is to decipher the contrast in different components of C cycle of anthropogenically affected Hooghly estuary and mangrove-dominated Sundarbans. While it would normally be ideal to have both large spatial and temporal coverage including measurements of several parameters along with their isotopic compositions to decipher the same, it is rarely possible due to severe logistics and technical limitations at different levels. We are sure, working in this region, you are aware of that. As we have said in response to reviewer 1, there is only one location in the Sundarbans so far (Ray et al., 2018) from where measurements for all components of C exists. We have strived to make it more representative by larger spatial coverage. We are also aware that four months are generally considered as postmonsoon; however, in light of the limitations mentioned above and advantage of spatial coverage, the conclusions of the present study can be considered as representative of the postmonsoon. Moreover, in the comment below, you are stating that one of the findings of the present study is similar to the one you observed, i.e., both Hooghly and Sundarbans are source of CO2 to the regional atmosphere. Although your findings on Sundarbans remains limited only to Matla estuary, which can hardly be representative of the vast Sundarbans. Compared to that, the present data set is better placed to represent Sundarbans.

Comment: The study area and sampling locations are quite similar with the recent work of Akhand et al. (2016). Moreover, the third objective and one of the conclusions of the present study is also very similar to the Akhand et al. (2016). For example, the authors stated, "The entire Hooghly-Sundarbans system acted as source of CO2 to the regional atmosphere with 17 times higher emission from the Hooghly compared to Sundarbans", whereas one of the key findings of Akhand et al. (2016) is "River

dominated Hugli Estuary emits 14 times more CO2 than the marine-dominated Matla Estuary". Surprisingly, despite of such degree of similarity between two studies, there is no comparison of data with Akhand et al. (2016) and not even mentioning of Akhand et al. (2016) in the present work.

Response: We are familiar of Akhand et al. (2016), which deals with CO2 dynamics in the Hooghly-Sundarbans, especially diurnal observation in Matla estuary. We appreciate your effort in performing 24 hours measurements in this turbulence estuary of the Indian Sundarbans. Akhand et al. (2016) covered four locations in the lower Hooghly estuary and 3 locations from Matla estuary; whereas we covered 13 locations from Hooghly and 11 locations from the Sundarbans including all major estuaries of the Indian Sundarbans (Saptamukhi, Thakuran and Matla) and their related waterways. Given the disparity in sampling designs and locations direct data comparison between these two studies will not be ideal. However, we would be happy to include the said study in the introduction section a recent work on Hooghly-Sundarbans system.

Comment: Reviewer 2 already mentioned that line no. 455 to 460 are self-contradictory. I want to add that I agree with the authors statement that in the estuarine water of Sundarban, an important source of CO2 is mangrove sediment pore-water exchange during tidal pumping. This fact is also well established from the diurnal dataset of Akhand et al. (2013) and Akhand et al. (2016) in Sundarban. But, it is not clear to me, how this phenomenon can prove the exogenous origin of CO2?

Response: We will be happy to include the above references to support our statement in the revised manuscript. "Exogenous" means outside the estuary not outside the mangrove ecosystem. We will clarify it in the revised manuscript. For Sundarbans, "Exogenous" CH4 is already established and for more details please see Dutta et al. (2015) published in Marine Chemistry.

Comment: Moreover, except Hooghly and its distributary Muriganga, all other rivers (Saptamukhi, Thakuran, Matla, Gosaba and Bidya) in the Indian part of Sundarban

have lost their original connections with the Ganga because of siltation and their estuarine character is now maintained by the monsoonal runoff only (Cole and Vaidyaraman, 1966). So, the central part of Sundarban (which comprises a major part of Indian Sundarban) experiences lack of freshwater (Chakrabarti1998; Mitra et al. 2009). Hence, the source of the exogenous nature of CO2 input in the Indian part of Sundarban needs more clarifications.

Response: It is obvious that compared to Hooghly, the estuaries of Sundarbans lack freshwater. However, it does not appear to be completely cut off from the source as can be seen from salinity range (salinity: 12.74-16.69) during the study period. However, no correlation between pCO2 and salinity ruled out significant role of freshwater contribution on CO2 of the estuary. The sources of CO2 in the Sundarbans include in situ OM respiration along with possibility of supply through pore water exchange during tidal pumping (so called exogenous with respect to estuary). Following reviewer-2 suggestion, analysis of ECO2-AOU relationship indicated CO2 production by OM respiration in the Sundarbans during the study period. Unfortunately, our dataset is not sufficient to prove exogenous supply of CO2 through pore-water. We may use Akhand et al. (2013) and Akhand et al. (2016) in the revised manuscript to support the argument.

Comment: In line no. 479 to 481 authors stated "FCO2 measured for the estuaries of Sundarbans was markedly higher than global mean FCO2 (âĄŞ63 $\mu$mol m-2 d-1) observed in mangrove creek and other similar estuaries (Call et al., 2015)". Reviewer 2 already correctly identified that the value should be 63 m mol m-2d-1. It might be a typo by the authors, but it may convey wrong message to the global audience about Sundarban's mangrove surrounding water. Because, one of the key findings of Akhand et al. (2016) is that the fCO2 (water) value of the Matla, a mangrove dominated estuary of Sundarban, is at the lower end of the reported data from other mangrove ecosystems of the world. Biswas et al. (2004) also found that the Sundarban's mangrove dominated water is acting as a sink for atmospheric CO2 for all the four post monsoon months,

while sampling in the three river-mouths. Also see Rosentreter et al. (2018), where they estimated world average flux of âĄŞ57.5 mmol m-2 d-1 of CO2 from the mangrove surrounding water, and also commented that the CO2 efflux from the estuarine water of Sundarban is much lower side than the world average even sinks for atmospheric CO2 in some cases.

Response: We are thankful to the reviewer -2 for pointing this out. After getting his comment we have rechecked the values with Call et al. (2015) and responded so in response to him. We believe that you are stretching it a bit too far for an unintentional typo in a manuscript undergoing peer-review process. As you said "It might be a typo by the authors..". It was just that.

References: Akhand, A., Chanda, A., Dutta, S., Manna, S., Sanyal, P., Hazra, S., Rao, K.H. and Dadhwal, V.K., 2013. Dual character of Sundarban estuary as a source and sink of CO2 during summer: an investigation of spatial dynamics. Environmental Monitoring and Assessment, 185(8), pp.6505-6515. Akhand, A., Chanda, A., Manna, S., Das, S., Hazra, S., Roy, R., Choudhury, S.B., Rao, K.H., Dadhwal, V.K., Chakraborty, K. and Mostofa, K.M.G., 2016. A comparison of CO2 dynamics and air-water fluxes in a river dominated estuary and a mangrove dominated marine estuary. Geophysical Research Letters, 43(22). Biswas, H., Mukhopadhyay, S.K., De, T.K., Sen, S. and Jana, T.K., 2004. Biogenic controls on the air-water carbon dioxide exchange in the Sundarban mangrove environment, northeast coast of Bay of Bengal, India. Limnology and Oceanography, 49(1), pp.95-101. Chakrabarti, P.S., 1998. Changing courses of Ganga, Ganga–Padma river system, West Bengal, India–RS data usage in user orientation, river behavior and control. Journal of River Research Institute, 25, pp.19-40. Cole, C. V., and P. P. Vaidyaraman. "Salinity distribution and effect of freshwater flows in the Hooghly River." In Proceedings of Tenth Conference on Coastal Engineering, Tokyo, Japan, September, pp. 1312-1434. 1966. Mitra, A., Banerjee, K., Sengupta, K. and Gangopadhyay, A., 2009. Pulse of Climate Change in Indian Suindarbans: A Myth or Reality?. National Academy Science Letters (India), 32(1), p.19. Rosentreter,

J.A., Maher, D.T., Erler, D.V., Murray, R. and Eyre, B.D., 2018. Seasonal and temporal $CO_2$ dynamics in three tropical mangrove creeks–A revision of global mangrove $CO_2$ emissions. Geochimica et Cosmochimica Acta, 222, pp.729-745.

Response: We will be happy to include some of the references in the revised manu

---

## Author Response (AR1)

**Editor's comment:**
**Comment:**
Two reviewers and Dr. Akhand raised a number of critical issues on your manuscript. I also agree to their primary concerns; above all the lack of clear study objectives and arguments unsupported by data (primarily those regarding the roles of phytoplankton productivity and OC mineralization in the Hooghly and exogenous C inputs in the Sundarbans). As reviewer 2 mentioned, major differences in measured parameters appear to derive simply from the dominant influence of fresh inputs to the Hooghly estuary. However, many of your arguments have been based on the assumption of anthropogenic impacts on the Hooghly C dynamics, without providing detailed descriptions and data on anthropogenic sources of C.

Response: Based on reviewer's comments, we have reanalysed our data which suggested significant impact of freshwater input on estuarine DIC and POC (line 454-461 and 577-582 of the revised manuscript) but not on DOC and $pCO_2$ (line 534-540 and 646-652 of the revised manuscript). We have discussed it in the revised manuscript. In the revised manuscript, organic carbon mineralization in the Sundarbans has been established with stronger evidence ($ECO_2$-AOU relationship) as suggested by the reviewer – 2 (line 619-633 of the revised manuscript). As we don't have any direct data, probability of $CO_2$ supply through pore-water exchange is supported based on higher low tide $CO_2$ compared to high tide at Matla estuary (of Sundarbans) as suggested by Akhand et al. (2016) (line 638-641 of the revised manuscript). For primary productivity, earlier works have been used to support our study (line 391-393 and 417-419 of the revised manuscript). Evidence for anthropogenic POC input was found from our study in the Hooghly based on stable isotope values (line 565-567 of the revised manuscript) but for DOC we have to rely only on concentration based interpretation due to lack of isotopic data (line 521-523 of the revised manuscript).

Given the large number of critical issues to address, a thorough revision beyond major revisions would be required to reconsider your manuscript for publication in Biogeosciences. The revised manuscript will be sent to the reviewers to make sure that you have adequately addressed all the raised issues and minor technical corrections. Regarding your assumption on the Hooghly as an anthropogenically modified system, I wondered if your summary of published water quality data (as shown in the supplement) would provide sufficient data supporting your arguments on the dominant role of C sources of anthropogenic origin driving the reported C dynamics in the Hooghly estuary. Please compile (and discuss) more concentration and isotope data on C species of anthropogenic origin, from your own surveys or the literature, to relate them to your interpretations.

Response: Based on reviewers as well as your comment, we have modified the entire manuscript and we think it is much better manuscript now. We have added more information (including quantification of anthropogenic discharge to the estuary on daily basis) in tabular form (Table 1 in the revised manuscript) to establish stronger anthropogenic influence on the Hooghly estuary and the same is proved from our study as well (line 564-567 of the revised manuscript). We hope now our justification and the manuscript will be well accepted to the editor.

I would like to ask you to make all the changes easily identifiable in a marked-up manuscript and a point-by-point reply to the reviewers' and my own comments to facilitate the second round of review. It would be a good idea to indicate line numbers of the revised manuscript when you respond to the reviewer comments.

Response: We are happy to do the needful during the submission procedure.

**Response to the reviewer - 1**

Review of Dutta et al., The authors made measurements of organic and inorganic carbon parameters, along with isotopes and other ancillary measurements in an attempt to determine the sources and distribution of DIC, DOC, and POC in an estuary in the Hooghly-Sundarbans system (shortly written as C biogeochemistry by the authors). Although the ms falls within the scopes of BG and covers a good data range from various sites of the estuarine system but finally it ends up in a disappointment because of poor writing and hesitations of choosing a concrete aim. Unfortunately, the manuscript reads like a data dump, with incomplete descriptions of the methods, presentation of the data, and some speculation about processes but with major processes left out; nothing seems conclusive. The manuscript is still in quite a rough stage, as detailed with a non-exhaustive list of examples below, and does not seem ready for publication.

**Response:** We are thankful to the reviewer for his constructive criticism and comments. We are now happy to include his suggestions in the revised manuscript at suitable places.

**Specific comments:**

The problem lies within the title. It seems the authors are in serous dilemma to show the data what actual basis: on C dynamics in polluted vs non-polluted system or only focus on mangroves and compare with sidechain Hooghly in a specific season or discuss on DIC mainly and less focus on DOC and POC or avoid already published articles on the same systems on same parameters on same season! (e.g. Samanta 2015, Ray 2018, 2015) Unfortunately nothing was clear due to poor writing and unclear intention.

*Response*: The main objective of the present study is to bring out contrast in different components of the carbon cycle of anthropogenically affected Hooghly estuary and mangrove-dominated estuaries of Sundarbans during postmonsoon. We have tried to focus on each component depending on the variabilities and scope of our data. We would respectfully disagree with the reviewer that we have tried to avoid the earlier works by Samanta et al., (2015) and Ray et al. (2015, 2018). We have cited their works and used the findings of these authors in our manuscript to interpret our data. We would also like to submit that whereas Samanta et al. (2015) is a nice study with comprehensive focus on only DIC in the Hooghly estuary; Ray et al. (2015, 2018) covers more number of parameters with limited spatial coverage. In a vast mangrove ecosystem as Sundarbans, Ray et al. (2015, 2018) have covered just one location during both studies. We have tried our best to cover the Hooghly-Sundarbans system on wider scale with multiple parameters to comprehensively study C dynamics in this system. We have made relevant changes in line nos: 127-155 of the revised manuscript with clearly stated intention (line no 142 – 145 of the revised manuscript).

[Figure]

[Figure]

*Ray et al. (2015)*                                    *Ray et al. (2018)*

Figure 1. Spatial coverage in the Hooghly-Sundarbans by Ray et al. (2015, 2018)

**Other major comments**

I would suggest authors to give details of the sampling stations e.g. how or what type of anthropogenic input is there in the Hooghly? From where it is more coming from (upstream?).

**Response:** Surface runoff in freshwater region such as waste water discharge from the City of Kolkata (St. H2) and jute industry (located in between many locations of St. H1 to H3) is a major source of anthropogenic inputs to the Hooghly. We have included previously published nutrients concentration as an evidence for higher anthropogenic input in the Hooghly. Relevant modification is done in line no : 120-126 along with addition of a new Table -1 showing contrast between the Hooghly and Sundarbans .

Its better to segment the study sites of Hooghly as upper/mid/lower stretch and Sundarbans as west/central and east. I anticipate the upper and mid stretches are human or industrial impacted compared to lower, so one of ideas in designing the story would be to explain variations of results within Hooghly first between e.g. H1-6 and H6-11 and then compare with S,T,M series. That would read the paper interesting otherwise its just mimicking the findings already shown by Samanta 2015, Ray 2018.

**Response:** During this postmonsoonal study, based on the present salinity range and gradient, it is difficult to divide the Hooghly estuary into upper/mid/lower stretch like other estuaries with sharp salinity gradient from fresh to marine zone. Although reviewer has suggested human and industrial influence along with lower estuarine region as a basis for such demarcation, we believe that such demarcation would be qualitative as no quantitative information are available to us to support such demarcation. Therefore, we are inclined to divide Hooghly estuary as freshwater zone (H1-H6) and mixing zone (H6 – H11) based on our salinity data. For the Sundarbans, the spatial extent is not wide enough (less than 100 km$^2$) to divide them into west, central and east zones. If we apply this criterion, we would be left with 3 data points from each region (upper/middle/lower), which is not enough for data analysis and further interpretation to understand the characteristics of individual estuaries (S, T and M). Therefore, in the revised manuscript, we discussed first the freshwater and mixing region in the Hooghly estuary and then compare it with the Sundarbans. We hope that the reviewer will agree to our suggestion.

Authors argued on C- data limitation of previous reports but it is found that Samanata'15 covered even much higher sites from Hooghly than the present report (c.a 35 vs 13 surface

water and 8 vs 8 ground water) and Ray '18 was also not far (>10 in S series vs 10 S,T,M). So this argument on data imitation does not hold true!

**Response:** We agree with the reviewer that DIC is extensively discussed by Samanta et al. (2015) for the Hooghly estuary with much better spatial and seasonal coverage compared to our study. The author also reported $\delta^{13}C_{POC}$ at some locations (n = 26). DIC and $pCO_2$ for the Hooghly and Matla estuaries have also been reported by Akhand et al. (2016). The first report for the Hooghly-Sundarbans system with different components of C cycle with their isotopic compositions were reported by Ray et al. (2015). However, this study is limited by spatial coverage (3 stations from Hooghly and one from Sundarbans). Unless reviewer is referring to paper other than Ray et al. (2018) published in The Science of Total Environment, his argument about Ray et al. (2018) having large sampling locations (>10 in S series vs 10 S,T,M as pointed out by the reviewer) appears to be not correct. The map of the sampling location of Ray et al. (2018) is shown above. In the light of the above, we would like to argue that the present study has much larger spatial coverage (13 stations from Hooghly and 11 from Indian Sundarbans, line 183-190 of the revised manuscript) with multiple parameters and is better equipped to decipher the differences in C biogeochemistry of the contrasting systems such as Hooghly and the estuaries of Sundarbans.

Result section is only meant for results and it should be avoided to define data set and add citations in Results that fully present in the paper. It is proposed to move those parts of the Result section to discussion (LN 229-234, 248-49, 257-59, 267-71).

**Response:** Thanks for the suggestion. We followed it throughout the revised manuscript.

This is over-speculative to argue on contributions of pore water on the overlying DIC concentrations based on only one measurement (Tab 3, Lothian PW).

**Response:** We agree with the reviewer that it is not enough to quantify contribution of pore water on adjoining estuarine water DIC pool based on a single measurement in this large mangrove ecosystem (Sundarbans). We are sure reviewer will appreciate that it is a logistics challenge to perform sampling in the Sundarbans. To perform sampling, permission is needed from the forest department. Also, very few islands are open for scientific investigations and some of them are tiger infested. During the present sampling, we had planned to cover at least all littoral zones of the Lothian Island. However, we were not permitted by the forest security service as conditions were not conducive to carry out investigation at mid and upper littoral zones. Therefore, we had to restrict our measurement in lower littoral zone only. Our advective DIC flux across mangrove sediment-estuary interface can be considered as first-time baseline value. We have indicated the reason for one sampling location in the manuscript as (line no 212-215 of the revised manuscript):

"Pore-water was also collected from lower littoral zone of the Lothian Island (one of the virgin island of the Indian Sundarbans) by digging a hole (~30 cm below the water table). It was not possible to collect pore-water samples from mid and upper littoral zones due to logistic problems."

LN342- 345: This is unclear why $\Delta DIC_{M2}$ is shown as micromole instead of permil.

**Response:** As you can see from the formula, the units of numerator and denominator is ($\mu$M x ‰) and ‰, respectively. The ‰' gets cancelled keeping $\Delta DIC_{M2}$ unit as '$\mu$M'.

Authors should better calculate the amount of DOC and POC added or subtracted from the system applying conservative mixing (same way they did for DIC) and explain in-depth details of their mixing pattern (same applies to DIC).

**Response:** Thanks to the reviewer for this suggestion. Using similarly calculated end members values or taken from the same references as DIC, added or removed POC and DOC in the Hooghly were calculated in the revised manuscript. For Sundarbans, mangrove derived POC and DOC addition/removal was calculated using the same expressions as DIC. Additionally, very similar to DIC, a mixing plot between $\Delta$POC and $\Delta\delta^{13}C_{POC}$ was plotted to explore influencing processes. However, for DOC it was not possible to perform this analysis due to unavailability of $\delta^{13}C_{DOC}$ data during the present study. We have used interrelationships between various parameters to justify removal or addition. We have included the above information in the revised manuscript (line no 543-614 of the revised manuscript).

**Based on the above, additions to the POC section is as following:** "To decipher processes involved in POC modification, estimated $\Delta$C for POC ($\Delta$POC) in the Hooghly indicated both net addition (n = 3) and removal (n = 3) of POC in the freshwater region ($\Delta$POC = − 0.45 to 0.48), whereas removal (n = 6) dominated over addition (n = 1) in the mixing region ($\Delta$POC = − 0.39 to 0.07). In an estuary, POC may be added through freshwater and surface runoff mediated inputs, phytoplankton productivity, and DOC flocculation. The removal of POC is likely due to settling at subtidal sediment, export to adjacent continental shelf region, modification via conversion to DOC and mineralization in case of oxygenated estuary.

The plot between $\Delta\delta^{13}C$ for POC ($\Delta\delta^{13}C_{POC}$) and $\Delta$POC (Fig. 5d) indicated different processes to be active in different regions of the Hooghly estuary. Decrease in $\Delta$POC with increase in $\Delta\delta^{13}C_{POC}$ (RR; n = 4 for mixing region and n = 1 for freshwater region) suggested modification of POC due to aerobic respiration (or mineralization). This process did not appear to significantly impact estuarine $CO_2$ pool as evident from the POC - $pCO_2$ relationship (freshwater region: p = 0.29, mixing region: p = 0.50; Fig. 5e). Decrease in both $\Delta$POC and $\Delta\delta^{13}C_{POC}$ (SD; n = 2 for mixing region and n = 2 for freshwater region) supported settling of POC to sub-tidal sediment. Despite high water residence time (~ 40 days during postmonsoon, Samanta et al., 2015), this process may not be effective in the Hooghly due to unstable estuarine condition (described earlier). Increase in $\Delta$POC with decrease in $\Delta\delta^{13}C_{POC}$ (SR, FR & PP; n = 2 for freshwater region) indicated increase of POC via surface and freshwater runoff as well as phytoplankton productivity. Increase in both $\Delta$POC and $\Delta\delta^{13}C_{POC}$ (n = 1 for mixing region and n = 1 for freshwater region) may be linked to DOC to POC conversion by flocculation.

In the Sundarbans, negative and lower $\Delta POC_{M2}$ (–209 to –28µM) compared to $\Delta POC_{M1}$ (–35 to 327µM) suggested DIC like behavior, i.e., simultaneous removal or modification along with addition of mangrove derived POC. No evidence for *in situ* POC-DOC exchange was obvious based on POC-DOC relationship; however, signal for POC mineralization was evident in the Sundarbans from POC - $pCO_2$ relationship ($r^2$ = 0.37, p = 0.05, Fig.5f). Similar to the Hooghly, despite high water residence time in mangroves (Alongi et al., 2005, Singh et al., 2016), unstable estuarine condition may not favor efficient settlement of POC at sub-tidal sediment. The export of POC from the Hooghly-Sundarbans system to the northern BOB, without significant *in situ* modification, is also a possibility. This export has been estimated to be ~0.02 - 0.07Tg and ~ 0.58Tg annually for the Hooghly and Sundarbans, respectively (Ray et al. 2018)."

Please look later for explanation related to DOC.

LN349 Are the ground and pore water discharge not being considered as 'biogeochemical' process?

**Response***:* We believe it is better to leave ground and pore water discharge from the realm of biogeochemical processes, as no biogeochemical processes are associated with them. It may be described as hydrological processes. We found the "The driving forces of pore-water and groundwater flow in permeable coastal sediments: A review" published by Santos et al. (2012) in the Estuarine Coastal and Shelf Science as a nice review work in this field.

Section 4.3. This part is weakly written and over-speculative without supporting any evidence e.g. the argument of DOC photo-oxidation or conversion of DOC to POC as removal process. While it requires suitable ambient condition for DOC photo-oxidation such as high water residence time, stable environmental condition (not expected in mangroves), the same applies to adsorption/desorption of DOC-POC. Part of that exchange is mediated by charged complexes, repulsion - attraction interactions, and therefore subject to salinity effects. So, when river water rich in DOC first mixes with saline water, at least a portion of DOC is lost from solution (removed) and incorporated into POC (Fe-oxide colloids usually are extracted at the same time). Once the salinity exceeds 2 - 3, however, the effect of salinity on coagulation behaviour is largely complete. Another point is no detailed explanation on distribution pattern with salinity was given, authors should highlight the reasons of the mild upward gradient along Hooghly and steep downward trend along the Sundarban.

**Response**: We are thankful to the reviewer for insightful comment on the DOC study. We have included these points in the revised manuscript (line no 463-540 of the revised manuscript).

[revised manuscript text omitted]

Section 4.4 LN410 only freshwater runoff, no surface run off that adds POC too in upstream?

**Response:** We have included possibility of surface runoff mediated POC addition in the revised manuscript (line no 546-557 of the revised manuscript).

**The section is looking like:** No significant SPM-salinity or POC-salinity relationship was observed during the present study (Fig. 5a & 5b), except for a moderate negative correlation between POC and salinity ($r^2 = 0.62$, p = 0.06) in the freshwater region of the Hooghly. This inverse relationship may be linked to freshwater mediated POC addition. Also, as described earlier, contribution of POC via surface-runoff is also a possibility in this region due to presence of several industries and large urban population (St: H2: Megacity Kolkata) that discharge industrial effluent and municipal wastewater to the estuary on regular basis (Table 1). Primary signal for surface runoff mediated POC addition was evident in the freshwater zone where ~ 61% and ~ 43% higher POC at 'H3' and 'H4' compared to an upstream location (St. H2) was observed. However, based on the present data, it is not possible to decouple freshwater and surface runoff mediated POC input to the Hooghly estuary."

LN440-446 this part is totally redundant as there was not an iota of signal of CH4 from the observed d13 POC (13CH4 is ~ 55-60 permil)

**Response:** We have removed the section from the revised manuscript.

Does the author have Chl-a or nutrient data (even from literature) to support higher marine input in POC in Sundarban and 13C values of mangrove leaf, and soil from Hooghly to denote higher terrigenous contribution to the POC pool? Authors are suggested to read carefully the works of Samanta'15 and Ray'18 and use their values to support some of the arguments.

**Response:** We are thankful to the reviewer for this suggestion. The modification is looking as following (line no – 561 – 577 of the revised manuscript):

"In general, wide range for $\delta^{13}C$ (rivers ~ –25 to –28‰; marine plankton ~ –18 to –22‰; $C_3$ plant ~ –23 to –34‰; $C_4$ plants ~ – 9 to –17‰) have been reported by different researchers in different ecosystems (Smith and Epstein, 1971, Hedges et al., 1997, Zhang et al., 1997, Dehairs et al., 2000, Bouillon et al., 2002). In the Hooghly, our measured $\delta^{13}C_{POC}$ suggested influx of POC via freshwater runoff as well as terrestrial $C_3$ plants. Additionally, the estuary was also anthropogenically stressed during postmonsoon with measured $\delta^{13}C_{POC}$ within the range reported for sewage ($\delta^{13}C$ ~ –28 to –14 ‰, Andrews et al., 1998). In the mixing zone of the Hooghly, significantly lower $\delta^{13}C_{POC}$ at 'H11' and 'H12' compared to other sampling locations may be linked to localized $^{13}C$ depleted organic C influx to the estuary from

adjacent mangrove and anthropogenic discharge, respectively. In the estuaries of Sundarbans, isotopic signatures of POC showed similarity with terrestrial $C_3$ plants. Interestingly, despite being mangrove-dominated estuary (salinity: 12.74 - 16.55) no clear signature of either freshwater or mangrove ($\delta^{13}C$: mangrove leaf ~ –28.4‰, soil ~ –24.3‰, Ray et al., 2015, 2018) borne POC was evident from $\delta^{13}C_{POC}$ values, suggesting towards the possibility of significant POC modification within the system. Modification of POC within the estuaries of Indian sub-continent have been reported earlier (Sarma et al., 2014)."

**points of concern**

terminology > I counted 'biogeochemistry' was used over 25 times in the 16 pages ms! too much. Additionally, this is not clear to me what does it actually mean by C biogeochemistry?

**Response:** We have taken care of it throughout the revised manuscript.

Is it C-components distributions in different phases (solid suspended and dissolved) under varying biogeochemical processes? If so please specify at least once

**Response:** We have included it in line no 143 of the revised manuscript.

> d13C values are not 'depleted' or 'enriched' (LN256, 428..). When referring to d13C values, they can be described as higher or lower when comparing different samples, or one could describe differences as e.g. a certain C pool is enriched or depleted in 13C versus another C pool or sample.

**Response:** We have taken care of it throughout the revised manuscript.

> r2 not R2

**Response:** We have taken care of it throughout the revised manuscript.

Inconsistent use of [POC] in the discussion, if the bracket is used for POC then it should also appear for DIC and DOC

**Response:** Brackets have been removed for all cases in the revised manuscript.

unit Random use of units: DOC in mg/L, DIC in mM, POC in uM. These should be harmonized. Use DOC in uM for better compare with other studies

**Response:** To maintain uniformity, all dissolved and particulate C parameters are presented as 'µM' in the revised version.

Sampling Define sampling strategy neatly, Its written postmonsoon was chosen due to high litterfall, but there is no account of litter source identified for DOC or POC or any impact positive or negative on estuarine C biogeochemistry authors assumed. That is to be addressed in the discussion. Mention the H, S, T, M series in the text Mention general tidal nature while sampling (height, HT/LT, depth).

**Response:** The leaf litter fall is the main source of organic carbon in mangrove sediment, which peaks during postmonsson (Ray et al., 2011). It is expected that high litter fall might influence C components in the Sundarbans (line no – 145 – 147 of the revised manuscript). The signal for influence of litter fall on DOC was evident from the DOC:POC ratio (as leaching) in the Sundarbans (line no – 498 – 502 of the revised manuscript), but no direct signature for mangrove leaf litter on POC was found (modification is also a possibility, see POC section for more details) (line no – 571 – 575 of the revised manuscript). We have included these points in the revised manuscript. Details on 'H, S, T and M' are included in

the revised manuscript (line no – 184 – 186 and 190 of the revised manuscript). All samples were collected during the low tide phase as intertidal mangrove sediment - water interaction through groundwater discharge is maximum during low tide phase. Therefore, low tide is ideal sampling time to understand impact of mangroves on adjoining estuarine systems. To assess contrasting features between the Sundarbans and Hooghly, sampling was also conducted during low tide in the Hooghly estuary (line no – 186 – 191 of the revised manuscript).

**Methods**

Comment – 1: specify> pore size of filters used for DOC, SPM relative uncertainty in POC methods;

**Response:** Pre-combusted (500°C for 6 hours) Whatman GF/F (pore size: 0.7µm) was used for DOC filtration and SPM collection. Uncertainty for POC was < 10%. Related information are included in line no – 221 – 222 and 243 of the revised manuscript.

Comment – 2: technique of pore water collection; ground water (from tube pump?)

**Response:** We have included collection techniques for pore-water and groundwater in the revised manuscript on following lines (line no – 212 – 218 of the revised manuscript):

"Pore-water was also collected from lower littoral zone of the Lothian Island (one of the virgin island of the Indian Sundarbans) by digging a hole (~30 cm below the water table). It was not possible to collect pore-water samples from mid and upper littoral zones due to logistic problems. After purging water at least twice in the bore, sample was collected from the bottom of the bore through syringe and transferred to the glass vial (Maher et al., 2013). Twelve groundwater samples were collected from the nearby locations of the Hooghly-Sundarbans system via tube pump."

**Figures**

Again weak representation: font sizes of x, y axis digits (and titles) to be increased much (too much stress to eyes now!); use box to cover legends, its confusing with data points and legends, remove break in y axis in Fug 3e and 4a), black star coding was used both for sundarban and observed d13DIC and grey round coding was used for Hooghly and observed DIC, these symbols must be changed to give separate identity of them in all figs <overall IMPROVE CLARITY of ALL FIGURES>

**Response:** We have presented high resolution figures in the revised manuscript to present each region and each ecosystem.

Data use a consistent number of decimals (1) to report d13C data, and Salinity considering the analytical error on the measurements.

**Response:** Both salinity and $\delta^{13}$C data will be presented up to two decimals in the revised manuscript.

**Minor comments**

Comment – 1: First sentence of abstract is redundant

**Response:** We have removed it from the revised manuscript.

Comment – 2: LN65 Use current reference for the riverine export flux (works of Pete Raymond, Huang)

**Response:** We are thankful to the reviewer for suggestion. We have included Huang et al. (2012) in the revised manuscript (line no – 65-69 of the revised manuscript).

Comment – 3: Many references are out of place e.g. the comparison of present data with Khura (LN 231, 249 Miyajima paper) was unlikely as two environments are totally different even if compared authors should mention conservative data like S in Khura estuary for better comparison.

**Response:** Salinity of Khura estuary is presented in the revised manuscript (line no – 399-405 of the revised manuscript)

Comment – 4: LN234: Pro-vide values of Samanta et al 2015

Response: We have included postmonsoon DIC (1.70 – 2.25mM) and $\delta^{13}C_{DIC}$ (–11.4 to – 4.0‰) values of Hooghly estuary as reported by Samanta et al. (2015) in the revised manuscript (line no – 369 of the revised manuscript).

Finally, I think it is necessary to stand back and consider how to best weave the entire story together in the discussion more efficiently and succinctly

Thanks for the valuable suggestions. We have almost entirely rewritten the manuscript and hope it will be well accepted to the reviewer.

**Response to Reviewer 2**

Review of Dutta et al "The postmonsoon carbon biogeochemistry of estuaries under different levels of anthropogenic impact". Submitted to Biogeosciences. This study presents data from a single cruise in 2 Indian estuaries to try and decipher differences in carbon cycling between a 2 Indian estuaries with differing levels of anthropogenic influence. After reading and rereading this paper several times, it is unclear what the purpose of this study is. There is no defined hypothesis to be tested, and while the title suggests there will be some kind of comparative analysis to look at anthropogenic impact on carbon cycling (an interesting and important topic), I am left a little underwhelmed with the analysis undertaken. The entire manuscript is based single sampling campaigns, which while not ideal is not the main issue. The main area of concern is the lack of any direction in the paper, and the somewhat descriptive and qualitative nature. I suggest that the authors define their hypothesis more clearly, and use the data to test this hypothesis.

Thanks to reviewer for going through our manuscript and providing valuable suggestions which will help to improve the quality of the revised version. We understand the concern he has raised and we have tried to improve the manuscript accordingly. As we have said in the response to reviewer 1, the main objective of the present study is to bring out contrast in different components of the carbon cycle of anthropogenically affected Hooghly estuary and mangrove-dominated estuaries of the Sundarbans during postmonsoon (line 142-145 of the revised manuscript). As suggested by the reviewer later in comments, we have introduced a table (Table 1) bringing out the differences in basic characteristics of these two systems, which will help the readers to appreciate the differences in anthropogenically affected and mangrove-dominated system. As suggested by the reviewer, in the revised version, given the contrasting nature of the estuaries, we also propose to bring out a central hypothesis. The central hypothesis of this study is: "Considering different nature and quantity of supplied OM within these two contrasting system, we hypothesize C metabolism between these two estuaries to be very different with higher $CO_2$ exchange flux from anthropogenically influenced estuary compared to mangrove-dominated estuary (line 147-151 of the revised manuscript)." Given the larger spatial coverage of the mangrove-dominated estuary during the present (so far only one estuary in this system has been studied), there is a need for this hypothesis to be tested on wider spatial level.

I have a series of comments below, some minor some major that may help.

Abstract I am not convinced that the data as presented can be used to draw such strong conclusions as to the drivers of carbon dynamics in the studied estuaries. For example, Ln 35-38 The evidence supporting these processes is weak at best – no measurements of production, carbonate dissolution nor pore-water exchange were measured, and the spatial trends in concentrations and isotopes (and relationships between carbon variables and DO etc.) were not strong enough to draw any distinct conclusions on the importance of these mechanisms.

Same goes for lines 45-47.

**Response:** Based on the specific comments of the reviewers, we have reanalysed the data and reassessed the role of processes he is referring to in sentences mentioned above. In the response to comments below, he will find that we have either discarded the descriptive part or backed the processes active with reanalysis of the data during the present study.

Line 49 – 52. I am unconvinced that the observed trends are shown to be directly linked to anthropogenic influence. Yes, the estuaries appear to differ, but what else might be driving this. For example, looking at salinity and $p$CO$_2$ in the 2 different estuaries – the highest

salinity in the "anthropogenically" impacted estuary is lower than the lowest salinity in the "undisturbed" estuaries. Could the observed differences simply be related to freshwater input? What are the nutrient concentrations in the 2 estuaries? How different are they in hydrodynamics (looks like the geomorphology is distinctly different between the 2 estuary types from Fig 1). These are just a few of the alternative reasons to look at for explaining the differences observed

**Response:** Based on the comments from both reviewers, we have provided a table in the response below (Table 1 in the revised manuscript), which will help readers to understand the basic differences between the two estuaries. The present study was carried out during postmonsoon season, which brings significant amount of freshwater inputs to the region. Moreover, the Hooghly undergoes sever anthropogenic stress as it passes through industrial areas as well as one of the most densely populated region in India (included in table). We revisited the data in light of the comments from both reviewers and in responses we discuss the changes and processes active in the two estuaries, which led to observed difference.

**Introduction**

Ln 59 – 60 What is meant by "record biogeochemical and hydrological processes"?

**Response:** We meant physical/hydrological processes such as mixing between marine and freshwater, tide and wave action, sediment transport etc. and biogeochemical processes such as primary productivity, organic matter decomposition etc. We believe the reviewer was concerned with 'record'. We have modified the sentence in the revised manuscript as follows (line 60-61 of the revised manuscript):

"Situated at the interface of land and sea, estuaries are highly susceptible to anthropogenic inputs and undergo intricate biogeochemical and hydrological processes."

Ln 67 – Richey is not correct ref for this statement (Richey paper is on Amazon)

**Response:** Thanks to point this out. We have modified the section as follows (line 65-69 of the revised manuscript):

"Tropical rivers, which constitute ~ 66% of global river water discharge, deliver ~ 0.53Pg C to the estuaries annually (Huang et al., 2012). The majority of this exported C is in dissolved form [dissolved inorganic C (DIC): $0.21PgCyr^{-1}$ and dissolved organic C (DOC): $0.14PgCyr^{-1}$] with some contribution as particulate [particulate organic C (POC): $0.13PgCyr^{-1}$ and particulate inorganic C (PIC): $0.05PgCyr^{-1}$] (Huang et al., 2012)."

Ln 68 – 70 – Still large uncertainties on estuarine $CO_2$ flux – look at error bars on Cai, 2011 estimate

**Response:** We have pointed out this issue in the revised manuscript (lines 70- 72 of the revised manuscript).

Ln 76 What is meant by "biogeochemical characteristic"?

**Response:** We meant with regards to cycling of bio-available elements, such as C, N and P. We have changed this sentence to more specific (line 102-105 of the revised manuscript).

Ln 78 – 79 Not always – see Cotovicz Jr, L. C., Knoppers, B. A., Brandini, N., Costa Santos, S. J., & Abril, G. (2015). A strong $co_2$ sink enhanced by eutrophication in a tropical coastal embayment (guanabara bay, rio de janeiro, brazil). Biogeosciences, 12(20), 6125-6146. doi:10.5194/bg-12-6125-2015

**Response:** Thanks for this reference. In the revised version, the section may look like (line 105-110 of the revised manuscript):

"In anthropogenically affected estuarine systems, heterotrophy generally dominates over autotrophy (Heip et al., 1995; Gattuso et al., 1998) and a substantial fraction of biologically reactive OM gets mineralized within the system (Servais et al., 1987; Ittekkot, 1988; Hopkinson et al., 1997; Moran et al., 1999). However, this is not always the case as observed in Guanabara Bay, Brazil, which acts as a strong $CO_2$ sink enhanced by eutrophication (Cotovicz Jr. et al., 2015)."

**Among others**

Ln 81 – 84 There has been a lot of work on mangrove carbon cycling work done since Dittmar and Larra's work in the early 2000's. Might be worth looking at more recent papers to see how far our understanding has come since then.

**Response:** We have modified this section in the revised manuscript. Following information may be added (line 76-96 of the revised manuscript).:

"Mangroves covering 137,760 $km^2$ along tropical and sub-tropical estuaries and coastlines (Giri et al. 2011) are among the most productive natural ecosystems in the world with net primary productivity of 218 ± 72Tg C $yr^{-1}$ (Bouillon et al. 2008). Fine root production coupled with litter fall and wood production are primary sources of mangrove derived C to intertidal sediment (Bouillon et al., 2008). The fate of this mangrove derived C remains poorly understood. Despite taking C burial and $CO_2$ emission flux across mangrove sediment-atmosphere interface into account, estimates of global mangrove C budget revealed a significant imbalance (~72%) between mangrove net primary productivity and its sinks (Bouillon et al., 2008). Earlier studies reported mangroves to be responsible for ~10% of the global terrestrial derived POC and DOC export to the coastal zones (Jennerjahn and Ittekkot, 2002; Dittmar et al. 2006). However, recent studies proposed DIC exchange as major C export pathway from mangrove forests, which was ~70% of the total mineralized C transport from mangrove forests to coastal waters (Maher et al., 2013; Alongi, 2014; Alongi and Mukhopadhyay, 2014). Another study reported groundwater advection from mangrove to be responsible for 93-99% of total DIC export and 89-92% of total DOC export to the coastal ocean (Maher et al., 2013). Upon extrapolating these C export fluxes to the global mangrove area, it was found that the calculated C exports were similar to the missing mangrove C sink (Sippo et al., 2016). The remaining C that escapes export gets buried in sub-surface sediment layers and participates in anaerobic processes (linked to production of biogenic trace gases like $CH_4$) or undergoes long-term sequestration (Jennerjhan and Ittekkot 2002; Barnes et al., 2006; Kristensen and Alongi, 2006; Donato et al., 2011; Linto et al., 2014)".
.

Ln 104- 106 Give some quantitative data to support your "anthropogenically influenced" argument. What are nutrient concentrations like? Population density? Land use? Freshwater inflow? Etc etc. A table compiling this data would give the reader an instant understanding of the differences.

**Response:** Thanks to the reviewer for bringing this point. Reviewer 1 has also asked to include some information in this context from literature. Texts or a table comparing the Hooghly and Sundarbans during postmonsoon based on nutrients concentration, Chla, population density and freshwater inflow will be introduced in the revised manuscript. The information is presented in Table-1 of the revised manuscript as follows:

| Parameters | Hooghly | Sundarbans |
|---|---|---|
| Nutrients (postmonsoon) | DIN: $14.72 \pm 1.77$ to $27.20 \pm 2.05\mu M$
DIP: $1.64 \pm 0.23$ to $2.11 \pm 0.46\mu M$
DSi: $77.75 \pm 6.57$ to $117.38 \pm 11.54\mu M$
(Mukhopadhyay et al., 2006) | DIN: $11.70 \pm 7.65\mu M$
DIP: $1.01 \pm 0.52\mu M$
DSi: $75.9 \pm 36.9\mu M$
(Biswas et al., 2004) |
| Chl*a* (postmonsoon) | Chl-a: $2.35 - 2.79$ mgm$^{-3}$
(Mukhopadhyay et al., 2006) | Chl*a*: $7.88 \pm 1.90$ mgm$^{-3}$
(Dutta et al., 2015) |
| Population density | North 24 Parganas and Hooghly: 2500 km$^{-2}$, Kolkata: 22000 km$^{-2}$, Howrah: 3300km$^{-2}$, South 24 Parganas: 820 km$^{-2}$ | No major Cities and town |
| Freshwater discharge (postmonsoon) | 3070 - 7301 million m$^3$
(Rudra et al., 2014) | No information available |
| Catchment area | $6 \times 10^4$km$^2$
(Sarkar et al., 2017) | No information available |
| Industrial and municipal wastewater discharge | 1153.8Million L d$^{-1}$
(Ghosh, 1973; Khan, 1995) | No information available |
| Dissolved metal flux | Increased from $230 - 1770\%$ annually
(Samanta and Dalai, 2018) | No information available |

Line 117 What is meant by positive and negative feedback here? These terms are not really applicable to biogeochemistry as a whole, but may be related to specific mechanisms/cycles.

**Response:** In the revised manuscript we have changed this statement as follows (line 145-147 of the revised manuscript):

"The postmonsoon sampling was chosen because of relatively stable estuarine condition for wider spatial coverage and peak mangrove leaf litter fall during this season (Ray et al., 2011), which may have influence on estuarine C dynamics."

Ln 137-140 Clearly there is freshwater input – the salinities are very low. In fact, my thoughts are that these freshwater inputs are a main driver of the observed differences.

**Response:** The freshwater input in the estuaries of Sundarbans is evident from the salinity values (12.64-16.69) during the study period. However, if you see the salinity values in the Hooghly estuary during the same season (0.04-10.37), the extent of freshwater input in Hooghly is far greater. This difference gets further widened during premonsoon. Because of this reason, we stated 'no perennial source of freshwater and limited anthropogenic input during monsoon". We have changed the sentence as (line 183-186 of the revised manuscript):

"Covering upper, middle, and lower estuarine regions, the present study was carried out during low tide condition in three major estuaries of the Indian Sundarbans [Saptamukhi (S1-S3), Thakuran (T1-T3), and Matla (M1-M3); Fig. 1a] along with its related waterways (S4 & M4)."

Ln 159 Assume the filters were GF/F filters – add these details.

**Response:** Yes, as reviewer stated it was Whatman GF/F filters. We have included it in the revised manuscript (line 207 of the revised manuscript).

Ln 161 Accuracy of TAlk measurements. Were CRMs measured (hope so!). Also add accuracy/precision etc of all other parameters.

**Response:** Uncertainties were as follows (line 201-244 of the revised manuscript):

Water temperature: $\pm0.1^\circ$C, Salinity: $\pm0.1$, DO: $\pm0.1$ mgL$^{-1}$, DIC: <1%, $\delta^{13}$C$_{DIC}$: < $\pm0.10$‰, DOC: $\pm52$ µgL$^{-1}$, POC: <10%, $\delta^{13}$C$_{POC}$: < $\pm0.10$‰, $p$CO$_2$: $\pm$ 1%. Yes, accuracy of TAlk was tested using Dickson standard (CRM: Bottle – 131) and uncertainty was found to be $\pm1$µmolkg$^{-1}$.

Ln 196 – 198 What were the input parameters for measuring pCO2? What disassociation constants were used etc?

**Response:** The $p$CO$_2$ was calculated using TAlk, pH, water temperature and salinity and the dissociation constants were calculated following Millero, (2013). We have included it in the revised manuscript (line 247-248 of the revised manuscript).

Ln 205 – 208 Why use L&M relationship? Need some kind of justification here other than saying it is conservative.

**Response:** Unfortunately, we don't have data on estuarine current velocity which along with wind speed is used for flux calculation as it is believed that turbulence of estuary might have an effect on air-water trace gas flux calculation. Based on only wind velocity, the L&M relationship is one of the most reliable and tested methods for flux calculations, which has been used in previous studies in the region as well (Biswas et al., 2004) We have included it in line 257-258 of the revised manuscript).

**Results**

Do not compare and contrast your data with previous studies in the results. Just report your data.

**Response:** We have removed the comparison part from the result to the discussion.

**Discussion**

Ln 289 – 293 What are the implications for these findings? Need to dig deeper or remove.

**Response:** Our intension was to present influence of salinity on pH and provide the information at the beginning that this region is a bicarbonate dominated system. We have removed the sentences in the revised version.

Ln 306-311 (and Fig 3b) How was the conservative d13C-DIC mixing line calculated? Looks like you have simply added a linear relationship between the 2 endmembers, the relationship is generally not linear (See Fry, B. (2002). Conservative mixing of stable isotopes across estuarine salinity gradients: A conceptual framework for monitoring watershed influences on downstream fisheries production. Estuaries, 25(2), 264-271. Also as you do not have any mineralogy of carbonates – I would avoid using the term "calcite" precipitation, change to "carbonate" precipitation.

**Response:** Concentrations and stable isotopic compositions of dissolved or particulate C (presented as C) during conservative mixing (C$_{CM}$ and $\delta^{13}$C$_{CM}$) were computed as follows (Carpenter et al., 1975, Mook and Tan, 1991):

$$C_{CM} = C_F F_F + C_M F_M$$

$$\delta^{13}C_{CM} = \frac{S_S [C_F \delta^{13}C_F - C_M \delta^{13}C_M] + S_F C_M \delta^{13}C_M - S_M C_F \delta^{13}C_F}{S_S (C_F - C_M) + S_F C_M - S_M C_F}$$

Here, 'S' denotes salinity, the suffixes CM, F, M and S denote conservative mixing, freshwater end member, marine end member and sample, respectively. $F_F$ = freshwater fraction = $1 - (S_S / S_M)$ and $F_M$ = marine water fraction = $(1 - F_F)$. This is a commonly used expression for such studies and has been followed by many other workers (Samanta et al. (2015); Bouillon et al. (2003)). The following expressions have been included in line 273-282 of the revised manuscript)

We have changed 'calcite precipitation' as 'carbonate precipitation' in the revised manuscript (line 385-386 and 1063-1064 of the revised manuscript).

Ln 323-325 – What does DO tell you about primary production? Looks like DO is generally under-saturated?

**Response:** The influence of primary productivity (PP) and/or $CO_2$ outgassing on DIC at the mixing zone was evident from mixing plot between $\Delta DIC$ and $\Delta\delta^{13}C_{DIC}$. We tried to go further and decouple these two processes based on TAlk - DIC relationship. However, as suggested by the reviewer, due to lack of PP measurements and level of DO indicate that it may not be a stretch. We have removed this part from the revised manuscript.

Ln 335-338 Describe all the terms in this equation in the following text

**Response:** We have described all terms in the revised manuscript (line no 273-306 of the revised manuscript) Additionally, in the revised manuscript, $\delta^{13}C_{Mangrove}$ will be changed as -28.4‰ as reported by Ray et al. (2015) for the Sundarbans system (line no 304 of the revised manuscript).

Ln 359 Where do the TAlk/DIC numbers come from? The stoichiometric relationship should be based on the slope of the line over the whole estuary, rather than individual data points – therefore not sure how you have a range here.

**Response:** Thanks to the reviewer for this suggestion. Based on his advice, we have made the changes in the revised manuscript (line 425-437 of the revised manuscript):

"High $p$CO$_2$ and DIC along with low pH and TAlk/DIC are general characteristics of groundwater, specially within carbonate aquifer region (Cai et al., 2003). Although all the parameters of groundwater inorganic C system (like pH, TAlk and $p$CO$_2$) were not measured during the present study, groundwater DIC were ~5.57 and ~3.61 times higher compared to mean surface water DIC in the Sundarbans and Hooghly, respectively. The markedly higher DIC in groundwater as well as similarity in its isotopic composition with estuarine DIC may stand as a signal for influence of groundwater on estuarine DIC, with possibly higher influence at the Sundarbans than Hooghly as evident from the slope of the TAlk - DIC relationships (Hooghly: 0.98, Sundarbans: 0.03). In the Sundarbans, to the best of our knowledge, no report exists regarding groundwater discharge. Contradictory reports exist for the Hooghly, where Samanta et al. (2015) indicated groundwater contribution at low salinity regime (salinity < 10, same as our salinity range) based on 'Ca' measurement, which was not observed based on 'Ra' isotope measurement in an earlier study (Somayajulu et al., 2002)."

Ln 364 – 368 Give details on this calculation. Just using the discharge rate and pore water DIC concentration I get a different value.

*Response:* Advective DIC flux from intertidal mangrove sediment to estuarine water column ($F_{ISW}$) was computed using the relation (Reay et al., 1995); $F_{ISW} = \Phi.v.C$; where, $\Phi$ = porosity of sediment = 0.58 (Dutta et al., 2013), $v$ = average linear velocity = $d\Phi^{-1}$ (d = specific discharge), C = DIC concentration in intertidal sediment pore water.

So ultimately: $F_{ISW}$ = d.C. During postmonsoon, d = 0.008 cm min$^{-1}$ (Dutta et al., 2015a). Therefore, $F_{ISW}$ = (0.008 cm min$^{-1}$ x 13.43 mmolL$^{-1}$) = 0.107mmol.cm.min$^{-1}$/1000cm$^3$= 0.000107 mmol cm$^{-2}$ min$^{-1}$ = 1.07 mmol m$^{-2}$ min$^{-1}$.

In Sundarbans, tides are semidiurnal in nature, so depending upon changes in hypsometric gradient discharge of pore water will be effective during low period only (i.e. 12 hours). So, $F_{ISW}$ =1.07 mmolm$^{-2}$min$^{-1}$ = (1.07 x 60 x 12 mmolm$^{-2}$d$^{-1}$) =770.4 mmolm$^{-2}$d$^{-1}$. There is a marginal difference in the manuscript, which will be corrected. We have included all details in line 441-449 of the revised manuscript.

Ln 383-390 – Not sure that looking at $p$CO$_2$ VS DOC gives any indication as to the importance of pore-water exchange! Could also simply be freshwater input from upstream, surface water runoff, or simply leaching/respiration.

**Response:** We have suggested to modify the DOC section which does not include the above argument. Please see response to reviewer 1 which deals with DOC (section 4.3, line 463-540 of the revised manuscript).

Ln 412 Give details about the "jute" industry.

 **Response:** This is an industry based on fiber of Corchorus plants, which is used in fabrics for packaging a wide range of agricultural and industrial commodities that require bags, sacks, packs, and wrappings. Locally this is known as *Jute* industry. We included some information on jute industry in line 124-126 of the revised manuscript.

Ln 424-426 The POC isotopes could simply be related to the relative amount of freshwater inputs in each system (this can also be applied to most of the other differences observed)

*Response:* Related to this following sections have included in the revised manuscript (line 565-575 of the revised manuscript):

"In the Hooghly, our measured $\delta^{13}C_{POC}$ suggested influx of POC via freshwater runoff as well as terrestrial C$_3$ plants. Additionally, the estuary was also anthropogenically stressed during postmonsoon with measured $\delta^{13}C_{POC}$ within the range reported for sewage ($\delta^{13}C \sim -28$ to $-14$ ‰, Andrews et al., 1998). In the mixing zone of the Hooghly, significantly lower $\delta^{13}C_{POC}$ at 'H11' and 'H12' compared to other sampling locations may be linked to localized $^{13}C$ depleted organic C influx to the estuary from adjacent mangrove and anthropogenic discharge, respectively.
In the estuaries of Sundarbans, isotopic signatures of POC showed similarity with terrestrial C$_3$ plants. Interestingly, despite being mangrove-dominated estuary (salinity: 12.74 - 16.55) no clear signature of either freshwater or mangrove ($\delta^{13}C$: mangrove leaf $\sim -28.4$‰, soil $\sim -24.3$‰, Ray et al., 2015, 2018) borne POC was evident from $\delta^{13}C_{POC}$ values, suggesting towards the possibility of significant POC modification within the system."

Ln 431-446 I am unsure why anaerobic respiration (which is energetically les favourable than aerobic respiration) would be more important in a well oxygenated estuary. The authors should expand this to explain things more clearly or remove.

**Response:** We have removed anaerobic respiration part from the revised manuscript.

Ln 447-451 What is the importance/implications of this – expand or remove.

*Response:* The intension was to quantitatively explore dominant OC form (DOC or POC) in total OC pool and dominant dissolved C form (DIC or DOC) in total dissolved C pool in the estuary. We have removed it from the revised manuscript.

Ln 455 – 460 These sections seem to contradict each other. Initially it is stated mangrove inputs are insignificant – then pore-water exchange of mangrove derived $CO_2$ is highlighted as important?

**Response:** For the revised manuscript, $ECO_2$ - AOU relationship (as suggested by the reviewer) was investigated (please see response to a later comment). The significant positive relationship between the two ($ECO_2 = 0.154AOU + 1.22$, $r^2 = 0.76$, $p = 0.005$, $n = 8$) suggested influence of OM respiration on pCO2 in the Sundarbans. Although, the calculated slope (0.154) was markedly lower compared to the slope for Redfield respiration in $HCO_3^-$ rich environment [$\Delta CO_2$: $(-\Delta O_2)$ = 124/138 = 0.90, Zhai et al., 2005] indicating effect of OM mineralization in controlling $pCO_2$ to be not so potent. Therefore, possibility of pore-water mediated $CO_2$ influx cannot be totally neglected in mangroves. Although based on present dataset (only low tide phase sampling) it is not possible to justify the argument, a signal for it was also observed from 24 hours $pCO_2$ observation in the Matla estuary (Sundarbans) by Akhand et al. (2016). We have added it in the revised version line 619-641 of the revised manuscript.

references for this process, might be more appropriate to use some of those here

*Response:* We have included some other mangrove references in the revised manuscript, such as Call et al. (2015), Bouillon et al. (2007) (line 639-640 of the revised manuscript).

Ln 463 – 466 How about plotting ECO2 vs AOU (in molar units). Look at the slope of the line. This will give a better indication of the importance or aerobic vs anaerobic R.

*Response:* Regarding this following modification is done in the revised manuscript (line 620-652 of the revised manuscript)

"In the Sundarbans, barring three locations (S3, T3 and M2), a significant negative correlation between $pCO_2$ and %DO ($r^2 = 0.76$, $p = 0.005$; Figure not given) suggested presence of processes, such as OM mineralization, responsible for controlling both $CO_2$ production and $O_2$ consumption in the surface estuarine water. Furthermore, significant positive correlation between $ECO_2$ and AOU ($ECO_2 = 0.057AOU + 1.22$, $r^2 = 0.76$, $p = 0.005$, $n = 8$; Fig.6a) confirmed the effect of aerobic OM mineralization on $CO_2$ distribution, particularly in the upper region of the Sundarbans. Our observations were in agreement with a previous study in the Sundarbans (Akhand et al., 2016) as well as another sub-tropical estuary, Pearl River estuary, China (Zhai et al., 2005). However, relatively lower slope for $ECO_2$ - AOU relationship (0.057) compared to the slope for Redfield respiration in $HCO_3^-$ rich environment [$(CH_2O)_{106}(NH_3)_{16}H_3PO_4 + 138O_2 + 18HCO_3^{2-} \rightarrow 124CO_2 + 140H_2O + 16NO_3^- + HPO_4^{2-}$; $\Delta CO_2$: $(-\Delta O_2)$ = 124/138 = 0.90, Zhai et al., 2005] suggested lower production of $CO_2$ than expected from Redfield respiration. This may be linked to formation of low molecular weight OM instead of the final product ($CO_2$) during aerobic OM respiration (Zhai et al., 2005). Moreover, $pCO_2$ - salinity relationship ($p = 0.18$, Fig.6b) confirmed no significant effect of fresh and marine water contribution on variability of $pCO_2$ in the Sundarbans. Other potential source of $CO_2$ to mangrove-dominated Sundarbans could be groundwater (or pore water) exchange across intertidal mangrove sediment-water interface. Although based on our own dataset, it is not possible to confirm the same. However, relatively higher $pCO_2$ levels during low-tide compared to high-tide at Matla estuary in the Sundarbans (Akhand et al. 2016) as well as in other mangrove systems worldwide (Rosentreter et al., 2018, Call et al., 2015, Bouillon et al., 2007) suggested groundwater (or pore water) exchange to be a potential $CO_2$ source in such systems.

Unlike Sundarbans, $ECO_2$ - AOU relationship did not confirm significant impact of OM respiration on $CO_2$ in either freshwater (p = 0.50) or mixing regions (p = 0.75) of the Hooghly (Fig. 6c). Overall, $pCO_2$ in the freshwater region of the Hooghly was significantly higher compared to the mixing zone (Table 3), which may be linked to $CO_2$ supply in the freshwater region through freshwater or surface runoff from adjoining areas (Table - 1). Inter-estuary comparison of $pCO_2$ also revealed ~1291 µatm higher $pCO_2$ in the Hooghly compared to the Sundarbans, which was largely due to significantly higher $pCO_2$ in freshwater region of the Hooghly (Table 2 & 3). Lack of negative correlation between $pCO_2$ - salinity in freshwater region (Fig. 6d) of the Hooghly suggested limited contribution of $CO_2$ due to freshwater inputs. Therefore, $CO_2$ supply via surface runoff may be primary reason for higher $pCO_2$ in the Hooghly estuary."

Ln 470 – 473 How was gas exchange and the differences between CO2 and O2 coupled into this calculation? Also how does this value compare to your air-water CO2 fluxes (you will need to normalize your volumetric rates to surface area for comparison)

**Response:** In both freshwater and mixing zone of the Hooghly estuary, no evidence for significant impact of aerobic OM respiration on $pCO_2$ was found. Therefore, we have remove this section from the revised version.

Ln 480 – I think your global value for mangrove systems (63 umol/m2/d) should be 63 mmol/m2/d – which is much higher than the fluxes measured in this study.

**Response:** We are thankful to the reviewer for pointing this out. We have rechecked the value from Call et al. (2015). The actual value (range) is ~ 43-59 mmol C $m^{-2}d^{-1}$. We have corrected it in the revised manuscript (line 659-661 of the revised manuscript).

**Conclusions:**

**Comment:** Point 1 – this variability is likely simply linked to the variability in salinity (and therefore freshwater inputs) between the studied estuaries.

**Response:** Freshwater inputs definitely has a role to play in the variabilities of DIC and POC as observed. However, these variabilities are also linked to *in situ* processes in the estuaries as described in our responses to both the reviewers.

**Comment:** Point 2 – Unconvinced that primary production has been shown to be the main controlling factor on DIC. Without any measurements of PP or some more thorough analysis of other potential mechanisms, this statement is far too strong

**Response:** We have changed the conclusion as (line 682-686 of the revised manuscript):

"Coupled with freshwater contribution, inorganic and organic C metabolism appeared to be dominant processes affecting DIC in the Hooghly. However, in the Sundarbans, significant DIC removal over addition was noticed. Influence of groundwater on estuarine DIC biogeochemistry was also observed with relatively higher influence at the Sundarbans."
.

**Comment:** Point 3 – I see no strong conclusive evidence of either of these points. Again statement is too strong without measurements of DOC flocculation or porewater exchange of DOC

**Response:** We have changed the conclusion as (line 687-689 of the revised manuscript):

"Higher DOC level in the Hooghly appeared to be regulated by coupled interactions among anthropogenic inputs, biogeochemical processes and groundwater contribution rather than freshwater mediated inputs"

**Comment:** Point 4 Assume this is based on isotopes? Again this could simply be related to the marked differences in freshwater content within each of the estuaries.

**Response:** We have changed the conclusion as (line 690-692 of the revised manuscript):

"Signatures of freshwater runoff, terrestrial $C_3$ plants, and anthropogenic discharge were found in POC of the Hooghly, whereas evidence for only $C_3$ plants were noticed at the Sundarbans with possible POC modification."

**Short Comment (Akhand)**

In the present study, large spatial extent has been covered which includes Hooghly River and other rivers of Indian part of Sundarban. My comments regarding the present study are as follows: 1. From the sampling strategy (line no. 150 to 153), it is apparent that only one-time discrete sampling has been done in all the sites in duplicate, whereas from the third objective of the study it is clear that the authors had the aim to quantify and characterise the air-water $CO_2$ flux for the post-monsoon season. The authors concluded "During post monsoon, the entire Hooghly-Sundarbans system acted as a source of $CO_2$ to the regional atmosphere." How can it be concluded (even quali-tatively) from such discrete data without performing at least one complete diurnal sampling at each site within post-monsoon season, while four months (October, November, December and January) are generally considered as post monsoon season in this region?

**Response:** As we have stated in response to the reviewer 1, the aim of the present study is to decipher the contrast in different components of C cycle of anthropogenically affected Hooghly estuary and mangrove-dominated Sundarbans (line 142-145 of the revised manuscript). While it would normally be ideal to have both large spatial and temporal coverage including measurements of several parameters along with their isotopic compositions to decipher the same, it is rarely possible due to severe logistics and technical limitations at different levels. We are sure, working in this region, you are aware of that. As we have said in response to reviewer 1, there is only one location in the Sundarbans so far (Ray et al., 2018) from where measurements for all components of C exists. We have strived to make it more representative by larger spatial coverage. We are also aware that four months are generally considered as postmonsoon; however, in light of the limitations mentioned above and advantage of spatial coverage, the conclusions of the present study can be considered as representative of the postmonsoon. Moreover, in the comment below, you are stating that one of the findings of the present study is similar to the one you observed, i.e., both Hooghly and Sundarbans are source of $CO_2$ to the regional atmosphere. Although your findings on Sundarbans remains limited only to Matla estuary, which can hardly be representative of the vast Sundarbans. Compared to that, the present data set is better placed to represent Sundarbans.

2. The study area and sampling locations are quite similar with the recent work of Akhand et al. (2016). Moreover, the third objective and one of the conclusions of the present study is also very similar to the Akhand et al. (2016). For example, the authors stated, "The entire Hooghly-Sundarbans system acted as source of $CO_2$ to the regional atmosphere with 17 times higher emission from the Hooghly compared to Sundarbans", whereas one of the key findings of Akhand et al. (2016) is "River dominated Hugli Estuary emits 14 times more $CO_2$ than the marine-dominated Matla Estuary". Surprisingly, despite of such degree of similarity between two studies, there is no comparison of data with Akhand et al. (2016) and not even mentioning of Akhand et al. (2016) in the present work.

**Response:** We are familiar of Akhand et al. (2016), which deals with $CO_2$ dynamics in the Hooghly-Sundarbans, especially diurnal observation in Matla estuary. We appreciate your effort in performing 24 hours measurements in this turbulence estuary of the Indian Sundarbans. Akhand et al. (2016) covered four locations in the lower Hooghly estuary and 3 locations from Matla estuary; whereas we covered 13 locations from Hooghly and 11 locations from the Sundarbans including all major estuaries of the Indian Sundarbans (Saptamukhi, Thakuran and Matla) and their related waterways. Given the disparity in

sampling designs and locations direct data comparison between these two studies will not be ideal. However, we have included the said study in the introduction section a recent work on Hooghly-Sundarbans system (line 133-134 of the revised manuscript).

3. Reviewer 2 already mentioned that line no. 455 to 460 are self-contradictory. I want to add that I agree with the authors statement that in the estuarine water of Sundarban, an important source of $CO_2$ is mangrove sediment pore-water exchange during tidal pumping. This fact is also well established from the diurnal dataset of Akhand et al. (2013) and Akhand et al. (2016) in Sundarban. But, it is not clear to me, how this phenomenon can prove the exogenous origin of $CO_2$?

**Response:** We have included Akhand et al. (2016) in the revised manuscript to support our statement in the revised manuscript (line 637-641 of the revised manuscript). "Exogenous" means outside the estuary not outside the mangrove ecosystem. We will clarify it in the revised manuscript. For Sundarbans, "Exogenous" $CH_4$ is already established and for more details please see Dutta et al. (2015) published in Marine Chemistry.

Moreover, except Hooghly and its distributary Muriganga, all other rivers (Saptamukhi, Thakuran, Matla, Gosaba and Bidya) in the Indian part of Sundarban have lost their original connections with the Ganga because of siltation and their estuarine character is now maintained by the monsoonal runoff only (Cole and Vaidyaraman, 1966). So, the central part of Sundarban (which comprises a major part of Indian Sundarban) experiences lack of freshwater (Chakrabarti1998; Mitra et al. 2009). Hence, the source of the exogenous nature of $CO_2$ input in the Indian part of Sundarban needs more clarifications.

**Response:** It is obvious that compared to Hooghly, the estuaries of Sundarbans lack freshwater. However, it does not appear to be completely cut off from the source as can be seen from salinity range (salinity: 12.74-16.69) during the study period. However, no correlation between $pCO_2$ and salinity ruled out significant role of freshwater contribution on $CO_2$ of the estuary. The sources of $CO_2$ in the Sundarbans include *in situ* OM respiration along with possibility of supply through pore water exchange during tidal pumping. Following reviewer-2 suggestion, analysis of $ECO_2$-AOU relationship indicated $CO_2$ production by OM respiration in the Sundarbans during the study period. Unfortunately, our dataset is not sufficient to prove supply of $CO_2$ through pore-water exchange. We have used Akhand et al. (2016) in the revised manuscript to support the argument (line 637-641 of the revised manuscript).

4. In line no. 479 to 481 authors stated "FCO$_2$ measured for the estuaries of Sundarbans was markedly higher than global mean FCO$_2$ ($\Box$63 µmol m$^{-2}$ d$^{-1}$) observed in mangrove creek and other similar estuaries (Call et al., 2015)". Reviewer 2 already correctly identified that the value should be 63 m mol m$^{-2}$d$^{-1}$. It might be a typo by the authors, but it may convey wrong message to the global audience about Sundarban's mangrove surrounding water. Because, one of the key findings of Akhand et al. (2016) is that the fCO$_2$ (water) value of the Matla, a mangrove dominated estuary of Sundarban, is at the lower end of the reported data from other mangrove ecosystems of the world. Biswas et al. (2004) also found that the Sundarban's mangrove dominated water is acting as a sink for atmospheric $CO_2$ for all the four post monsoon months, while sampling in the three river-mouths. Also see Rosentreter et al. (2018), where they estimated world average flux of $\Box$57.5 mmol m$^{-2}$ d$^{-1}$ of $CO_2$ from the mangrove surrounding water, and also commented that the $CO_2$ efflux from the estuarine water of

Sundarban is much lower side than the world average even sinks for atmospheric $CO_2$ in some cases.

**Response:** We are thankful to the reviewer -2 for pointing this out. After getting his comment we have rechecked the values with Call et al. (2015) and responded so in response to him. We believe that you are stretching it a bit too far for an unintentional typo in a manuscript undergoing peer-review process. As you said "It might be a typo by the authors.". It was just that. The above value is clarified in line 660 of the revised manuscript.

References: Akhand, A., Chanda, A., Dutta, S., Manna, S., Sanyal, P., Hazra, S., Rao, K.H. and Dadhwal, V.K., 2013. Dual character of Sundarban estuary as a source and sink of $CO_2$ during summer: an investigation of spatial dynamics. Environmental Monitoring and Assessment, 185(8), pp.6505-6515.

Akhand, A., Chanda, A., Manna, S., Das, S., Hazra, S., Roy, R., Choudhury, S.B., Rao, K.H., Dadhwal, V.K., Chakraborty, K. and Mostofa, K.M.G., 2016. A comparison of $CO_2$ dynamics and air-water fluxes in a river dominated estuary and a mangrove dominated marine estuary. Geophysical Research Letters, 43(22).

Biswas, H., Mukhopadhyay, S.K., De, T.K., Sen, S. and Jana, T.K., 2004. Biogenic controls on the air-water carbon dioxide exchange in the Sundarban mangrove environment, northeast coast of Bay of Bengal, India. Limnology and Oceanography, 49(1), pp.95-101.

Chakrabarti, P.S., 1998. Changing courses of Ganga, Ganga–Padma river system, West Bengal, India–RS data usage in user orientation, river behavior and control. Journal of River Research Institute, 25, pp.19-40.

Cole, C. V., and P. P. Vaidyaraman. "Salinity distribution and effect of freshwater flows in the Hooghly River." In Proceedings of Tenth Conference on Coastal Engineering, Tokyo, Japan, September, pp. 1312-1434. 1966.

Mitra, A., Banerjee, K., Sengupta, K. and Gangopadhyay, A., 2009. Pulse of Climate Change in Indian Suindarbans: A Myth or Reality?. National Academy Science Letters (India), 32(1), p.19.

Rosentreter, J.A., Maher, D.T., Erler, D.V., Murray, R. and Eyre, B.D., 2018. Seasonal and temporal $CO_2$ dynamics in three tropical mangrove creeks–A revision of global mangrove $CO_2$ emissions. Geochimica et Cosmochimica Acta, 222, pp.729-745.

**Response:** We have included Akhand et al. (2016) and Rosentreter et al. (2018) in the revised manuscript (line 713-716 and 970-973 of the revised manuscript).

**The postmonsoon carbon biogeochemistry of estuaries under different levels of anthropogenic impacts**

**Manab Kumar Dutta[1], Sanjeev Kumar[1]\*, Rupa Mukherjee[1], Prasun Sanyal[2], Sandip Kumar Mukhopadhyay[2]**

[1]Geosciences Division, Physical Research Laboratory, Ahmedabad - 380009, Gujarat, India
[2]Department of Marine Science, University of Calcutta, Kolkata - 700019, West Bengal, India

\***Correspondence**: Sanjeev Kumar (sanjeev@prl.res.in)

**Abstract**

The present study focused on understanding  differences in postmonsoon carbon (C) biogeochemistry of two adjacent estuaries undergoing different levels of anthropogenic stress by investigating anthropogenically influenced Hooghly estuary and mangrove-dominated estuaries of the Sundarbans in the north-eastern India. The salinity of well oxygenated (%DO: 91 - 104%) estuaries of the Sundarbans varied over a narrow range (12.74 - 16.69)  relative to the Hooghly (0.04 - 10.37).  Apart from freshwater contribution, mixing model suggested carbonate precipitation and dissolution  to be major processes controlling DIC  in the in the freshwater region of the Hooghly, whereas  phytoplankton productivity and $CO_2$ outgassing dominated mixing zone. The signatures of significant DIC removal over addition through mangrove derived organic C mineralization was observed in the Sundarbans. The DOC in the Hooghly was ~ 40% higher compared to the Sundarbans, which was largely due to cumulative effect of anthropogenic inputs, biogeochemical processes and groundwater contribution rather than ~~Sundarbans. In both estuarine systems, DOC behaved non-conservatively with ~ 40% higher DOC levelcompared to the Sundarbans. No significantof phytoplankton production on DOC level was found in these estuaries, however signalDOC input through pore-water exchange at thewas observed. Relatively lower $\delta^{13}C_{POC}$ at the Hooghly~~ suggested significant POC modifications. The average $pCO_2$ in the Hooghly was ~ 1291 µatm higher compared to the Sundarbans with surface run-off and organic matter respiration as dominant factors controlling $pCO_2$ in the Hooghly and Sundarbans, respectively. The entire Hooghly-Sundarbans

system acted as source of $CO_2$ to the regional atmosphere with ~17 times higher emission from the Hooghly compared to Sundarbans. The present studyTaken together, the cycling of C in estuaries with different levels of anthropogenic influences are clearly establishes thedifferent with dominance of anthropogenically influenced estuary over relatively pristine mangrove-dominated one in as $CO_2$ source to the regional greenhouse gas budget and climate change perspective.atmosphere.

**1 Introduction**

Estuaries connecting terrestrialSituated at the interface of land and marine ecosystems recordsea, estuaries are highly susceptible to anthropogenic inputs and undergo intricate biogeochemical and hydrological processes operating between these two environments. Estuaries play an important role in modulating global carbon (C) cycle and anthropogenic carbon dioxide ($CO_2$) budget (Bauer et al., 2013; Regnier et al., 2013; LeQuéré et al., 2016). Atmospheric $CO_2$ is sequestered into terrestrial systems through photosynthesis and weathering reactions and is transported to the ocean via rivers and estuaries. About $1 \times 10^{15}$ gTropical rivers, which constitute ~ 66% of C is discharged annually from the land global river water discharge, deliver ~ 0.53Pg C to the ocean through rivers and estuaries (Degensannually (Huang et al., 1991). Around 40 %2012). The majority of this exported C is discharged as in dissolved form [dissolved inorganic carbonC (DIC)): 0.21PgCyr$^{-1}$ and the rest as dissolved organic carbonC (DOC) and ): 0.
[revised manuscript text omitted]

DIP: $1.64 \pm 0.23$ to $2.11 \pm 0.46\mu M$
DSi: $77.75 \pm 6.57$ to $117.38 \pm 11.54\mu M$
(Mukhopadhyay et al., 2006) | DIN: $11.70 \pm 7.65\mu M$
DIP: $1.01 \pm 0.52\mu M$
DSi: $75.9 \pm 36.9\mu M$
(Biswas et al., 2004) |
| Chl$a$ (postmonsoon) | Chl-a: $2.35 - 2.79$ mgm$^{-3}$
(Mukhopadhyay et al., 2006) | Chl$a$: $7.88 \pm 1.90$ mgm$^{-3}$
(Dutta et al., 2015) |
| Population density | North 24 Parganas and Hooghly: 2500 km$^{-2}$, Kolkata: 22000 km$^{-2}$, Howrah: 3300km$^{-2}$, South 24 Parganas: 820 km$^{-2}$ | No major Cities and town |
| Freshwater discharge (postmonsoon) | 3070 - 7301 million m$^3$
(Rudra et al., 2014) | No information available |
| Catchment area | $6 \times 10^4$km$^2$
(Sarkar et al., 2017) | No information available |
| Industrial and municipal wastewater discharge | 1153.8Million L d$^{-1}$
(Ghosh, 1973; Khan, 1995) | No information available |
| Dissolved metal flux | Increased from $230 - 1770\%$ annually
(Samanta and Dalai, 2018) | No information available |

Table - 2: Physicochemical parameters, inorganic and organic C related parameters, and CO$_2$ exchange fluxes across water-atmosphere at the estuaries of Sundarbans. Here, water temperature (W$_T$), DO, isotopic compositions, DIC, DOC, POC, $p$CO$_2$ and FCO$_2$ are presented in '°C' 'mgL$^{-1}$', '‰', 'μM', 'μM', 'μM', 'μatm' and 'μmol m$^{-2}$ hr$^{1}$', respectively.

| Station | W$_T$ | Salinity | DO | pH | DIC | δ$^{13}$C$_{DIC}$ | DOC | POC | δ$^{13}$C$_{POC}$ | $p$CO$_2$ | FCO$_2$ |
|---------|-------|----------|------|------|-----------|-------|----------|-----|---------|-----|------|
| S1 | 28.50 | 12.74 | 6.65 | 8.02 | 1780 | − 5.59 | 278 | 154 | − 22.85 | 536 | 26.5 |
| S2 | 28.00 | 16.02 | 6.65 | 8.02 | 1703 | − 4.33 | 267 | 124 | − 23.54 | 561 | 30.3 |
| S3 | 28.00 | 16.69 | 6.61 | 8.12 | 1700 | − 4.29 | 197 | 114 | − 23.43 | 395 | 0.9 |
| S4 | 29.00 | 15.25 | 6.46 | 8.01 | 1861 | − 5.27 | 315 | 93 | − 23.68 | 543 | 27.6 |
| T1 | 29.00 | 14.30 | 6.56 | 8.05 | 1757 | − 5.57 | 259 | 80 | − 23.62 | 490 | 18.1 |
| T2 | 29.00 | 15.51 | 6.74 | 8.07 | 1727 | − 4.79 | 182 | 106 | − 23.21 | 456 | 11.9 |
| T3 | 28.50 | 16.55 | 6.46 | 8.11 | 1683 | − 4.39 | 154 | 154 | − 22.97 | 403 | 2.4 |
| M1 | 28.00 | 15.14 | 6.99 | 8.07 | 1711 | − 5.93 | 282 | 264 | − 23.07 | 443 | 9.4 |
| M2 | 28.00 | 15.14 | 6.91 | 8.12 | 1735 | − 4.63 | 219 | 436 | − 23.15 | 376 | -2.6 |
| M3 | 28.00 | 15.23 | 7.46 | 8.13 | 1736 | − 5.30 | 222 | 287 | − 23.62 | 401 | 1.9 |
| M4 | 28.50 | 14.78 | 6.84 | 8.04 | 1920 | − 5.38 | 215 | 96 | − 23.82 | 503 | 20.3 |

Table - 3: Physicochemical parameters, inorganic and organic C related parameters, and $CO_2$ exchange fluxes across water-atmosphere at the Hooghly estuary. Here, water temperature ($W_T$), DO, all isotopic compositions, DIC, DOC, POC, $pCO_2$ and $FCO_2$ are presented in '°C' 'mgL$^{-1}$', '‰', ' µM', 'µM', 'µM', 'µatm' and 'µmol m$^{-2}$ hr$^1$', respectively.

| Station | $W_T$ | Salinity | DO | pH | DIC | $\delta^{13}C_{DIC}$ | DOC | POC | $\delta^{13}C_{POC}$ | $pCO_2$ | $FCO_2$ |
|---|---|---|---|---|---|---|---|---|---|---|---|
| H1 | 32.0 | 0.04 | 6.29 | 7.92 |  2700 | − 6.98 |  244 | 313 | − 25.34 | 2036 | 285.2 |
| H2 | 33.0 | 0.07 | 6.11 | 7.71 |  1678 | − 8.38 |  304 | 177 | − 25.19 | 2316 | 343.8 |
| H3 | 31.0 | 0.08 | 6.45 | 7.83 |  2498 | − 6.70 |  235 | 286 | − 25.95 | 2490 | 355.4 |
| H4 | 31.0 | 0.13 | 5.24 | 7.73 |  2446 | − 7.38 |  243 | 254 | − 25.40 | 2691 | 389.2 |
| H5 | 31.0 | 0.19 | 5.38 | 7.77 |  2355 | − 7.56 |  340 | 130 | − 25.67 | 2123 | 293.1 |
| H6 | 30.5 | 0.32 | 5.66 | 7.31 |  2157 | − 8.61 |  308 | 116 | − 24.07 | 4678 | 717.5 |
| H7 | 31.5 | 5.83 | 6.71 | 7.68 |  1829 | − 6.79 |  662 | 145 | − 24.70 | 1184 | 132.0 |
| H8 | 31.0 | 5.19 | 7.14 | 7.31 |  2023 | − 6.78 |  354 | 139 | − 23.47 | 3153 | 455.8 |
| H9 | 31.5 | 9.08 | 6.62 | 7.90 |  1915 | − 6.08 |  332 | 161 | − 23.53 | 665 | 44.9 |
| H10 | 31.5 | 9.72 | 6.17 | 8.08 |  1787 | − 5.78 |  249 | 95 | − 24.06 | 452 | 10.1 |
| H11 | 31.0 | 8.43 | 6.37 | 8.07 |  1977 | − 7.21 |  358 | 95 | − 25.94 | 486 | 15.6 |
| H12 | 31.5 | 5.83 | 7.40 | 8.29 |  1871 | − 6.60 |  260 | 133 | − 26.28 | 274 | -19.3 |
| H13 | 31.0 | 10.37 | 7.00 | 8.24 |  1843 | − 5.57 |  394 | 129 | − 24.72 | 267 | -19.8 |

Table - 4:  The  concentrations and $\delta^{13}C_{DIC}$ of groundwater (GW) and pore-water (PW) samples collected around Hooghly-Sundarbans system.

| Ecosystem | Station |  (μM) | $\delta^{13}C_{DIC}$ (‰) |
|---|---|---|---|
| **Hooghly** | H3GW | 11756 | − 12.66 |
| | H4GW | 6230 | − 7.85 |
| | H5GW | 6327 | − 8.96 |
| | H6GW | 7026 | − 11.27 |
| | H7GW | 5655 | − 6.91 |
| | H11GW | 9115 | − 7.67 |
| | H12GW | 6858 | − 7.49 |
| | H13GW | 7258 | − 7.21 |
| | Gangasagar GW | 7246 | − 6.67 |
| **Sundarbans** | Lothian GW | 7524 | − 6.84 |
| | Lothian PW | 13425 | − 18.05 |
| | Kalash GW | 13599 | − 6.69 |
| | Virat Bazar GW | 8300 | − 10.56 |

[Figure]

[Figure]

[Figure]

[Figure]

**Fig.-1**

[Figure]

[Figure]

**Fig.-2**

[Figure]

[Figure]

**Fig. 3**

[Figure]

**Fig.-4**

[Figure]

[Figure]

[Figure]

**Fig.-5**

[Figure]

**Fig.6**

---

## Author Response (AR3)

**Response to the technical corrections**

**Comment:** Thank you for your thorough revision. I am pleased to let you that your manuscript can be published after some technical correction as noted below:

**Response:** Thanks to the editor for appreciating our effort and recommending it for publication. As suggested, we have done all technical corrections in the abstract as well as in the main text.

**Comment:** -L32: "DO: 91-104%"

**Response:** We have corrected it throughout.

**Comment:** -L37-38: Please specify "removal of DIC" by what process? Did you mean "through mineralization of mangrove-derived organic C"?

**Response:** We have specified the processes in the abstract as well as corresponding parts in the main text of the revised manuscript.

**Comment:** -L38: Did you mean "The concentration of DOC in the Hooghly was ~ 40% higher than in the Sundarbans?"

**Response:** Yes. We have rewritten it accordingly in the abstract as well as corresponding parts in the main text of the revised manuscript.

**Comment:** -L39: What "biogeochemical processes"?

**Response:** In the revised manuscript we have specified "DOC-POC interconversion" in the abstract as well as corresponding parts in the main text of the revised manuscript.

**Comment:** -L45: Please specify "where" or "by what processes" following "modifications"

**Response:** We have included the responsible process of POC modification in the Sundarbans (i.e., degradation of POC by respiration) in the revised manuscript.

**Comment:** -L46: Did you mean "decomposition" or "degradation" by respiration?

**Response:** Yes, we meant that. Following your suggestion, we have revised the sentences.